# Transient Attracting Profiles in the Great Pacific Garbage Patch

Luca Kunz[1], Alexa Griesel[1], Carsten Eden[1], Rodrigo Duran[2,3], and Bruno Sainte-Rose[4]

[1]Institute of Oceanography, Universität Hamburg, Hamburg, Germany
[2]National Energy Technology Laboratory, U.S. Department of Energy, Albany OR, USA
[3]Planetary Science Institute, 1700 East Fort Lowell, Tucson, AZ 85719, USA
[4]The Ocean Cleanup, Rotterdam, The Netherlands

**Correspondence:** Luca Kunz (luca.kunz@orac.earth)

**Abstract.** A major challenge for cleanup operations in the Great Pacific Garbage Patch is the daily prediction of plastic concentrations that allows to identify hotspots of marine debris. Lagrangian simulations of large particle ensembles are the method in use and effectively reproduce observed particle distributions at synoptic scales $\mathcal{O}(1000\,\text{km})$. However, they lose accuracy at operational scales $\mathcal{O}(1-10\,\text{km})$ and operators regularly experience differences between predicted and encountered debris accumulations within the garbage patch. Instead of asking *Where do objects go as they follow the current?* as in Lagrangian methods, here, we take a different approach and question *Which locations attract material?*. The recently developed concept of Transient Attracting Profiles (TRAPs) provides answers to this because TRAPs uncover the most attracting regions of the flow. TRAPs are the attracting form of hyperbolic Objective Eulerian Coherent Structures and are computable from the instantaneous strain field on the ocean surface. They describe flow features which attract drifting objects and could facilitate offshore cleanups that are currently taking place in the Great Pacific Garbage Patch. However, the concept remains unapplied since little is known about the persistence and attraction of these features, specifically within the Pacific. Therefore, we compute a 20-year dataset of daily TRAP detections from satellite-derived mesoscale velocities within the North Pacific subtropical gyre. We are the first to track these instantaneous flow features as they propagate through space and time. It allows us to study the life cycle of TRAPs, which can range from days to seasons and lasts an average of six days. We show how long-living TRAPs with lifetimes beyond 30 days intensify and weaken over their life cycle, and we demonstrate that the evolution stage of TRAPs affects the motion of nearby surface drifters. Our findings indicate that, at the mesoscale, operators in the Great Pacific Garbage Patch should search for long-living TRAPs that are at an advanced stage of their life cycle. These TRAPs are the most likely to induce a large-scale confluence of drifting objects and their streamlining into hyperbolic pathways. Such a streamlined bypass takes, on average, five days and creates an opportunity to filter the flow around TRAPs. But we also find TRAPs that retain material over multiple weeks where we suspect material clustering at the submesoscale. Prospective research could investigate this further by applying our algorithms to soon-available high-resolution observations of the flow. Our analysis contributes to a better understanding of TRAPs, which can benefit other offshore operations besides ocean cleanups, such as optimal drifter deployment, oil spill containment or humanitarian search and rescue.

## Short summary

TRansient Attracting Profiles (TRAPs) indicate the most attracting regions of the flow and have the potential to facilitate offshore cleanup operations in the Great Pacific Garbage Patch. We study the characteristics of TRAPs and the prospects for predicting debris transport from a mesoscale permitting dataset. Our findings provide an advanced understanding of TRAPs in this specific region and demonstrate the importance of TRAP lifetime estimations to an operational application. Our TRAPs tracking algorithm complements the recently published TRAPs concept and prepares its use with high-resolution observations from the SWOT mission. Our findings may also benefit research in other fields like optimal drifter deployment, sargassum removal, oil spill containment or search and rescue.

# 1 Introduction

The horizontal long-term flow at the ocean surface is understood to be the main forcing that transports floating material over large distances (van Sebille et al., 2020) and can be well-described by the combination of geostrophic and Ekman currents (Röhrs et al., 2021). Floating marine debris follows the large-scale convergence within each of the five subtropical gyres and forms basin-scale accumulation zones (van Sebille et al., 2020), which exhibit elevated levels of plastic concentration. In this context, the North Pacific subtropical gyre is, to date, the area of highest scientific and public concern. First initiatives to clean up ocean plastic pollution at a global scale are taking place in this particular gyre (Slat, 2022), and a variety of experiments have been dedicated to estimating the limits of this accumulation zone, which is colloquially termed the *Great Pacific Garbage Patch* (Onink et al., 2019; Lebreton et al., 2018; Law et al., 2014). Figure 1 highlights this large-scale convergence zone and the horizontal long-term flow at the surface of the northeast Pacific Ocean.

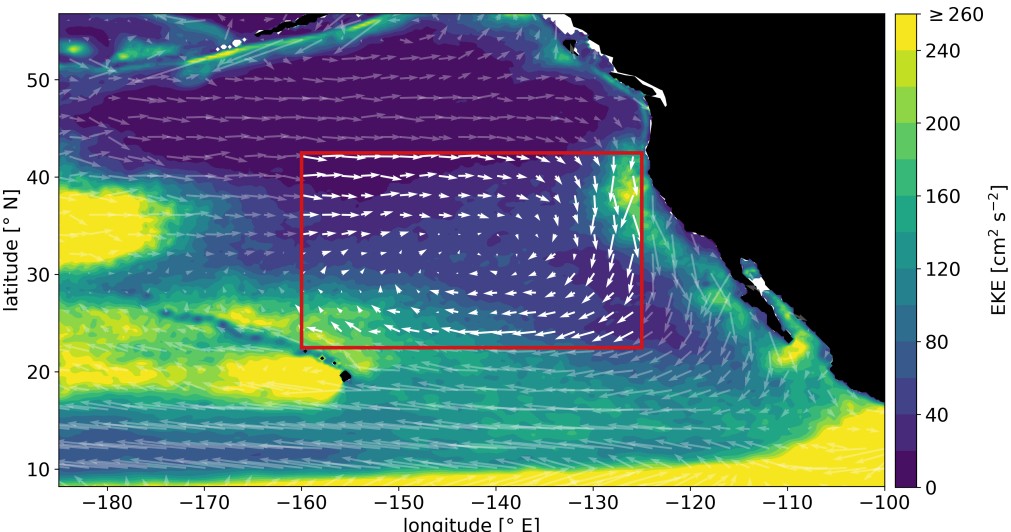

**Figure 1.** Mean circulation and EKE in the eastern North Pacific. Vectors indicate the direction and magnitude of the 20-year average combined near-surface geostrophic and Ekman current velocities for the period 2000-2019 (CMEMS, 2022a). Arrow size scales with increasing velocity magnitude. The red box defines our study domain and highlights the area of large-scale convergence due to Ekman transport. The boundaries of the study domain approximate the estimated limits of the Great Pacific Garbage Patch (Onink et al., 2019; Lebreton et al., 2018; Law et al., 2014). The ocean surface is coloured by the magnitude of eddy kinetic energy (EKE) w.r.t. to the same period, derived from geostrophic currents only (CMEMS, 2022b).

A constant challenge for cleanup operations in this region is to predict day-to-day variabilities of plastic concentration and to identify hotspots of marine debris. The common method is to release large ensembles of virtual particles at the ocean surface and to derive their trajectories and distribution as they follow the time-evolving surface flow (Duran et al., 2021; van Sebille et al., 2020; van Sebille et al., 2018). Measurements from altimetry or estimations from numerical ocean models typically

provide the velocity fields that drive these Lagrangian particle simulations, from which particle concentrations at the ocean surface can be derived. At synoptic scales $\mathcal{O}(1000 \text{ km})$, Lagrangian simulations succeed in predicting the limits of the Great Pacific Garbage Patch (Onink et al., 2019; Lebreton et al., 2018; Law et al., 2014). However, within the garbage patch and at operational scales $\mathcal{O}(1-10 \text{ km})$, The Ocean Cleanup reports that Lagrangian simulations often fail in accurately predicting the particle distributions they observe at sea. This deficiency may result from the combination of Lagrangian methods with submesoscale velocity estimations from numerical models. At operational scales and within this region, numerical models currently represent the only source for now- and forecast estimations of the surface flow, but these simulations can only approximate the true dynamics at sea. On the other hand, Lagrangian simulations can produce significant trajectory errors if the underlying velocity data or the trajectory modelling itself are missing important physics, specifically because errors can accumulate quickly during the integration process (Duran et al., 2021; Serra et al., 2020).

Instead of asking *Where do objects go as they follow the current?* as in Lagrangian methods, here, we take a fundamentally different approach and question *Which locations attract material?*. The recently published concept of Transient Attracting Profiles (TRAPs, Serra et al. (2020)) can answer this since it allows to detect the most attracting regions of the flow. TRAPs are the attracting form of hyperbolic Objective Eulerian Coherent Structures (OECSs, Serra and Haller (2016)) and are computable from the symmetric part of the velocity gradient. They indicate regions of maximal compression and stretching on a two-dimensional surface, such as the ocean surface, that translate into the attraction and hyperbolic transport of nearby floating objects.

Serra et al. (2020) and Duran et al. (2021) provide experiments that show the capability of TRAPs to attract drifting objects. They demonstrate how TRAPs uncover the stretching of tracer patterns that remains undetected by conventional diagnostics like streamlines or divergence. Their experiments also indicate that TRAPs are insensitive to the shape, submergence level, release time and release position of drifting objects. These parameters are generally uncertain in applications but must be considered in Lagrangian simulations. Moreover, Serra and Haller (2016), Serra et al. (2020) and Duran et al. (2021) argue that TRAPs are robust to moderate errors in the underlying velocity field while trajectory-based methods are susceptible to error accumulation during the velocity integration. We list more benefits of the TRAPs method in Table B1, but here, we highlight one essential aspect of the concept: TRAPs can predict material aggregation. The time scale of prediction will generally depend on the temporal and spatial scales of the underlying velocity data. At the mesoscale and daily frequency, TRAPs computed from nowcast or near-real-time observations of the flow should inherently indicate where drifting objects will aggregate within a few days. Lagrangian simulations can only provide such predictions by extrapolating nowcast observations or using model forecasts, which, as reported, leads to inaccurate predictions of debris hotspots.

So, it stands to reason that TRAPs could facilitate offshore cleanups that are currently taking place in the Great Pacific Garbage Patch. However, the concept remains unapplied since the persistence and attractive properties of TRAPs have not been characterised in this particular region, neither at the mesoscale $\mathcal{O}(10-100 \text{ km})$ nor at the submesoscale $\mathcal{O}(1-10 \text{ km})$.

Therefore, we create a 20-year dataset of daily mesoscale TRAP detections and provide a first analysis of these features within the North Pacific subtropical gyre.

We compute TRAPs from combined near-surface geostrophic and Ekman current velocities since geostrophic velocity from altimetry is the only large-scale observation that resolves flow features at the mesoscale (Abernathey and Haller, 2018). Many studies have established that altimetry-derived velocity products are accurate for Lagrangian transport applications, see Sect. 3.1.3 of Duran et al. (2021) and references therein, and we focus the analysis on TRAPs derived from observations of sea

surface height. With this, we advance our knowledge about the natural occurrence of large-scale TRAPs and test their ability to predict debris aggregation from only mesoscale-permitting flows, which are available in near-real-time. Our analysis provides an essential first step to applying the concept during offshore cleanups. A second important step would be to characterise submesoscale TRAPs and their relation to mesoscale TRAPs. While we leave the submesoscale analysis for future research, our mesoscale analysis provides a blueprint that can be applied to higher-resolution observations, which will be available soon

from the current SWOT mission (International Altimetry Team, 2021).

Since conventional altimetry measurements of the ocean surface filter out all small-scale, short-term features of the flow, our study will focus on the low-frequency circulation. Therefore, we will locate TRAPs within the mesoscale eddy field by comparing our dataset to corresponding records of mesoscale eddy detections. We investigate the relation between TRAPs and

mesoscale eddies to advance our understanding of strain between eddies and its potential for predicting debris transport.

Serra et al. (2020) mention that TRAPs "necessarily persist over short times" with examples of TRAPs existing for several hours and attracting nearby objects within two to three hours. These time scales for persistence and impact derive from TRAPs computed upon submesoscale velocity fields with a high tempo-spatial resolution. However, the concept is scale-invariant and

can be applied to velocity data of any resolution. TRAP characteristics will depend on the velocity data used for their computation, and the lifetime and impact of TRAPs will be relative to the time scales of the oceanic structures that give rise to the hyperbolic-type Lagrangian motion that TRAPs identify. For this reason, Duran et al. (2021) find persistent TRAPs that predict transport patterns eight days in advance. They compute TRAPs from mesoscale surface velocities with daily frequency and consequently study these structures at larger scales than Serra et al. (2020). With our choice of altimetry data, we follow

Duran et al. (2021) and expect mesoscale TRAPs to exist and impact on time scales comparable to those of mesoscale flow features. Indeed, some examples from altimetry data in Serra and Haller (2016) show that different types of OECSs, including the attracting hyperbolic type studied here, can last for at least six days.

TRAPs that persist over several days will then highlight permanent flow features where we might find large-scale conflu-

ence of material. Therefore, identifying persistent TRAPs can help to point cleanup operations in the right direction, which motivates us to design a tracking algorithm that follows TRAPs through space and time. We are the first to track these Eulerian flow features, determine their lifetimes and describe their propagation through the domain. We further combine these methods

with observations of surface drifters to investigate the TRAP properties relevant for an offshore cleanup in the Great Pacific Garbage Patch. Eventually, the findings we make have the potential to facilitate even more maritime search operations that are taking place in other contexts and regions.

The paper is organised as follows. In Section 2, we review the theoretical aspects of Transient Attracting Profiles, outline the design of the experiment and go through the methods we use for our analysis of mesoscale TRAPs. Section 3 presents our results in four parts - the spatial distribution of TRAPs, their life cycle and propagation, vorticity patterns around TRAPs and the impact of TRAPs on nearby drifters. In Section 4, we discuss our findings and the directions they offer for future research.

## 2 Methodology

### 2.1 Transient Attracting Profiles

Serra et al. (2020) derive TRAPs from the instantaneous strain field on the ocean surface using snapshots of the two-dimensional surface velocity field $\boldsymbol{u}(\boldsymbol{x},t)$, with $\boldsymbol{u}$ being dependent on position $\boldsymbol{x}$ and time $t$. The symmetric part of the velocity gradient represents the time-dependent strain tensor $\mathbf{S}(\boldsymbol{x},t) = \frac{1}{2}(\boldsymbol{\nabla}\boldsymbol{u}(\boldsymbol{x},t) + [\boldsymbol{\nabla}\boldsymbol{u}(\boldsymbol{x},t)]^{\top})$ with the eigenvalue fields $s_i(\boldsymbol{x},t)$ and eigenvector fields $\boldsymbol{e}_i(\boldsymbol{x},t)$. $\mathbf{S}$, $s_i$ and $\boldsymbol{e}_i$ denote the respective quantities at a fixed position $\boldsymbol{x}_0$ and time $t_0$ and we apply the notation for the diagonal form of $\mathbf{S}$ from Serra and Haller (2016):

$$\mathbf{S}\boldsymbol{e}_i = s_i\boldsymbol{e}_i, \quad |\boldsymbol{e}_i| = 1, \quad i = 1,2; \quad s_1 \leq s_2, \quad \boldsymbol{e}_2 = \mathbf{R}\boldsymbol{e}_1, \quad \mathbf{R} := \begin{pmatrix} 0 & -1 \\ 1 & 0 \end{pmatrix} \tag{1}$$

The deformation of any fluid's surface element $A$ is determined by the local strain rates $s_i$, which specify the rates of stretching ($s_i > 0$) or compression ($s_i < 0$) of $A$ along the principle axes indicated by the local eigenvectors $\boldsymbol{e}_i$, see Olbers et al. (2012) for details. Due to the condition $s_1 \leq s_2$, the local eigenvector $\boldsymbol{e}_1$ identifies the direction of compression and $\boldsymbol{e}_2$ the direction of stretching for a non-uniform deformation with $s_1 < 0$ and $s_2 > 0$. The compression and stretching of surface elements translate into the attraction and repulsion of material. Negative local minima of $s_1(\boldsymbol{x},t_0)$ at snapshot time $t_0$ describe the most attracting regions of the flow, maximising attraction normal to $\boldsymbol{e}_2$ at the respective position. For incompressible conditions, $s_1 = -s_2$ further holds and local minima of $s_1(\boldsymbol{x},t_0)$ simultaneously indicate local maxima of $s_2(\boldsymbol{x},t_0)$. Then, the strongest attraction and strongest repulsion occur at the same position and in orthogonal directions.

TRAPs indicate the most attracting regions of the flow as they start at negative local minima of the $s_1(\boldsymbol{x},t_0)$ strain field and extend tangent to the local eigenvectors $\boldsymbol{e}_2$ until the strain rate $s_1$ along the tangent ceases to be monotonically increasing. Consequently, TRAPs contain one minimum value of $s_1(\boldsymbol{x},t_0)$, i.e. the point of strongest attraction perpendicular towards the TRAP. The position of this local minimum is called the TRAP *core*, which represents an objective saddle-type stagnation point of the unsteady flow (Serra and Haller, 2016). A TRAP describes the direction of maximal stretching because it is tangent at each point to the unit eigenvectors $\boldsymbol{e}_2$, and in the following, we will also call it a TRAP *curve*.

Figure 2 presents an example of a Transient Attracting Profile. The TRAP is displayed as a red curve with the TRAP core as a red dot in the middle. The image shows this structure upon a colourmap of the underlying $s_1(\boldsymbol{x},t_0)$ strain field, superimposed by velocity vectors of the surrounding flow, all derived from the same snapshot of combined surface geostrophic and Ekman current velocities at time $t_0$. The TRAP core coincides with the point of highest attraction, i.e. a negative local minimum of the $s_1(\boldsymbol{x},t_0)$ field, and the velocity indicates water motion normal towards the TRAP and subsequent transport to both ends of the structure. The figure also displays the positions of this TRAP after 10 and 20 days.

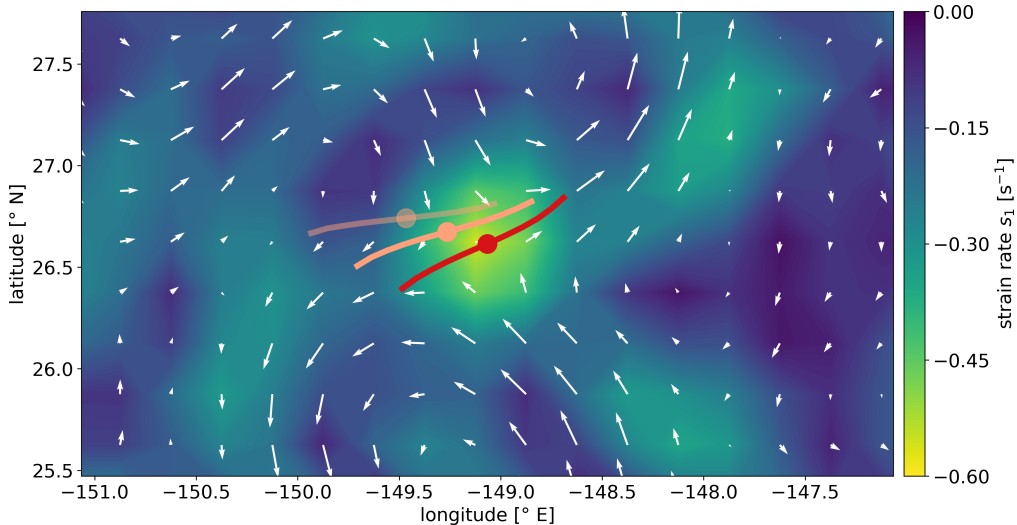

**Figure 2.** Example Transient Attracting Profile. The red structure represents one TRAP from our 20-year record of daily TRAPs, computed from snapshots of combined surface geostrophic and Ekman current velocities (CMEMS, 2022a). The red dot in the middle indicates the TRAP core. The red line represents the TRAP curve. Vectors illustrate the surrounding flow, and the colourmap indicates the $s_1(\boldsymbol{x}, t_0)$ strain field derived from the same velocity snapshot at time $t_0$. The geometries in salmon colour indicate, with increasing transparency, the position of the same TRAP after 10 and 20 days.

## 2.2 TRAPs computation

Serra et al. (2020) have published a programme to compute TRAPs from two-dimensional snapshots of an Eulerian velocity field $\boldsymbol{u}(\boldsymbol{x}, t)$ (Serra, 2020), see Table B2 for details of the algorithm and Kunz (2024a), Kunz (2024b) for our post-processing of the output. We call the points that discretise the TRAP curve *curve points*, and the algorithm gives the positions of all curve points and corresponding TRAP cores in the domain. It also outputs the normal attraction rate $s_1$ at every TRAP core, which we name *core attraction*.

We truncate TRAP curves wherever the attraction rate along the curve falls below 30 % of the attraction at the respective core. Such a cutoff criterion makes sense physically because the attraction of nearby parcels becomes negligible as distance increases away from the core. Without cutoff, TRAPs can become indefinitely long and merge with nearby structures, which makes them hard to distinguish. In addition, their converging ends put wrong emphasis on regions between TRAP cores where the attraction rate is comparably low. Moving away from a TRAP core, the local eigenvectors $\boldsymbol{e}_2$ also start pointing in arbitrary directions, and they stop being representative of the TRAP. The attraction strength criterion does not necessarily prevent such an excessive integration of the eigenvectors. To obtain an accurate TRAP that indicates hyperbolic flow, one has to define an upper limit for TRAP length, in addition to the cutoff by attraction strength, see Fig. S1 in the Supplementary Material for details.

The TRAPs algorithm (Serra, 2020) provides default values of 1° for the maximum arclength of a TRAP and 30 % for the attraction strength cutoff. Together, both parameters determine the length of a TRAP curve and must be set thoughtfully before computation. With the mesoscale velocity data we use, the preset values provide a clear saddle-type representation of TRAPs. Also, a maximal arclength of 1° limits each TRAP branch to a maximal arclength of 0.5°, which approximately equals the average radius $\bar{r}_e$ of mesoscale eddies in our domain. We consult an eddy census product by AVISO+ et al. (2022) and

derive an average eddy radius and its standard deviation of $\bar{r}_e \approx (53 \pm 20)$ km. We expect TRAPs to highlight strain between mesoscale eddies, and therefore, it is helpful to study TRAPs and eddies on comparable length scales. For these reasons, we keep the preset parameter values. However, this choice does not affect our main diagnostics, and future studies should adjust these settings according to the applied velocity data. Modified TRAP lengths do not change our analysis since our statistics refer to the position and attraction of the TRAP *core*.

    We set the boundaries of our study domain to [22.5° N, 42.5° N] in latitude and [-160° E, -125° E] in longitude and compute TRAPs within this domain. We choose these limits, highlighted by the red box in Fig. 1, to enclose the North Pacific accumulation zone between Hawaii and California as defined by Onink et al. (2019). These boundaries also warrant no intersection with any land mass.

    We compute TRAPs from daily snapshots of combined near-surface geostrophic and Ekman currents and therefore extract the velocity fields *uo* and *vo* from the product *Global Total Surface and 15m Current (COPERNICUS-GLOBCURRENT) from Altimetric Geostrophic Current and Modeled Ekman Current Reprocessing* that is provided by the E.U. Copernicus Marine Service (CMEMS, 2022a). The velocity fields are available at three-hourly instantaneous time steps, from which we select data

at UTC midnight to sample snapshots with a daily frequency. The latitude-longitude grid of the velocity field has a resolution of 0.25°.

## 2.3   Tracking algorithm

Our tracking algorithm is applied to the full TRAPs record and finds spatially proximate detections at consecutive timestamps,

which can be identified as one single feature of the flow. The only free parameter $\epsilon$ defines the size of the search area around a given TRAP to look for a detection in the following snapshot, and we set it to $\epsilon = 0.25°$. For a higher value, the algorithm creates "jumps" from a current to an unrealistically far future TRAP detection and overestimates trajectory lengths, see Sect. S2 in the Supplementary Material for a detailed explanation and motivation for this choice. The algorithm assigns a unique label to each TRAP trajectory and its associated instances. It derives metrics like the lifetime $\Lambda$ of TRAPs and their age $\tau$ at

a particular snapshot. The algorithm only captures the time spent *inside* the study domain and period. Therefore, it gives rise to potential bias in the lifetime estimation of TRAPs that reach beyond the tempo-spatial limits of the domain. However, we find that only 5.4 % of all TRAP trajectories are adjacent to these limits and might not entirely occur within the study domain. Our conclusions and the distribution of TRAP lifetimes don't change if we exclude those biased trajectories, see Sect. S3 in

the Supplementary Material, where we analyse this in detail. With the trajectory estimation, we can now derive the zonal and meridional translation speeds $c_x$ and $c_y$ for every instance of a TRAP trajectory. Therefore, we choose all TRAPs that persist for at least three days and compute propagation speeds using centred differences. We derive no velocities at the start and end of a trajectory where the TRAP forms and decays. At these stages, $c_x$ and $c_y$ could be estimated by following the underlying oceanic structure that creates the TRAP.

## 2.4 Mesoscale eddy data

We compare TRAPs against eddy detections from *The altimetric Mesoscale Eddy Trajectories Atlas (META3.2 DT)* that is produced by SSALTO/DUACS and distributed by AVISO+ with support from CNES, in collaboration with IMEDEA (AVISO+ et al., 2022). This dataset provides Eulerian detections of eddies derived from sea surface height (SSH) contours. It is at frequency with our TRAPs record and includes estimations of the eddy contour speed $U$. The eddy contour speed $U$ is the average geostrophic speed of the contour of maximum circum-average geostrophic speed for the detected eddy. We filter the dataset for eddy detections within the study domain and period and retrieve 28,645 cyclonic and 24,193 anticyclonic eddy trajectories from 689,460 cyclonic and 686,720 anticyclonic eddy detections. Since we cut off eddy trajectories beyond the domain boundaries, trajectories can show discontinuities if they leave and return to the domain. This effect occurs for 4.2 % of all eddy trajectories. We do not correct for this cutoff since the impact on our aggregate statistics should be negligible. We estimate the eddy lifetime by taking the time difference between the first and the last occurrence within our domain, added by one day. Accordingly, the estimated eddy lifetime can also include times spent outside the domain, given that the eddy returns to it again. We further derive eddy propagation speeds as we do for TRAPs.

## 2.5 Vorticity curve

We use the relative vorticity field to characterise the flow around TRAPs. Many TRAPs seem to be surrounded by four vortices of alternating polarity. In this case, two vortices on each side of a TRAP curve exhibit perpendicular flow towards the TRAP core and tangential flow away from it with respect to the TRAP curve. We call this vorticity pattern a *quadrupole* and classify variations of it. To detect vorticity patterns without a coordinate transformation, we draw a circle in the horizontal plane around every TRAP core with a radius equal to the distance between the core and the furthest curve point. Starting from the position of the furthest curve point, we parameterise the circle, with angles increasing counter-clockwise, and bilinearly interpolate the vorticity field to the points on this circle. The *vorticity curve* $\zeta(\alpha)$ then describes the vorticity along this circle with respect to the angle of parameterisation, i.e. the phase $\alpha$.

By visualising the vorticity curve, we can see polarity changes in the surrounding vorticity field and their spatial orientation towards the TRAP. Figure 3a illustrates an example of such a vorticity measurement around a TRAP. For a quadrupole in the northern hemisphere, the polarity pattern along the vorticity curve results in cyclonic ($\oplus$, positive vorticity), anticyclonic ($\ominus$, negative vorticity), cyclonic, anticyclonic and the vorticity curve reveals four zero crossings.

Since the ensemble mean of all vorticity curves $\zeta(\alpha)$ will indicate a quadrupole pattern, we filter the ensemble for specific combinations of four vortices with either cyclonic or anticyclonic rotation. We do not explicitly resolve patterns with less than four vortices because we can identify them from this classification if needed. There are in total $2^4$ possible vortex combinations, 10 of which exert distinct dynamics, i.e. are unique under rotations of $180°$ around a TRAP core. To isolate each pattern, we first remove a constant average background vorticity from every vorticity curve $\zeta(\alpha)$ and then filter the ensemble for these 10 vortex combinations. To detect the vortex configuration within a given vorticity curve, we divide the curve into four phase intervals of $\alpha_{\mathrm{I}} = [0, \pi/2)$, $\alpha_{\mathrm{II}} = [\pi/2, \pi)$, $\alpha_{\mathrm{III}} = [\pi, 3\pi/2)$, $\alpha_{\mathrm{IV}} = [3\pi/2, 2\pi)$ and determine the sign of the average vorticity within each interval, see Fig. S5 for more details on all 10 vorticity patterns.

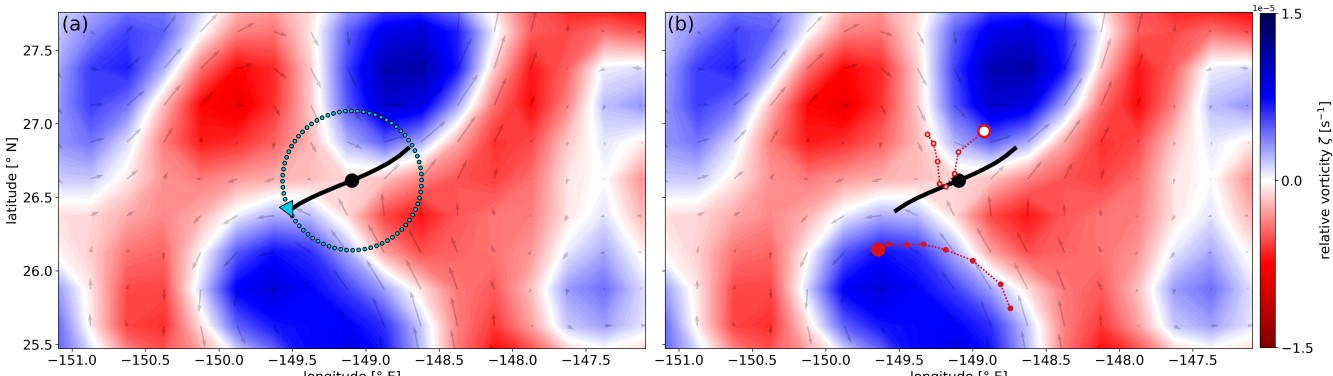

**Figure 3.** Vorticity circle and observed drifter transport around a TRAP. (a) Blue dots indicate the interpolation points of the vorticity curve $\zeta(\alpha)$. The blue triangle highlights the starting point of the parameterisation at $\alpha = 0°$. $\alpha$ increases counter-clockwise as we sample the vorticity $\zeta(\alpha)$ along the circle. (b) Big red circle markers indicate the current position of two drifters around the same TRAP. Red tails with small markers show the respective drifter positions throughout the preceding seven days. One drifter is drogued and indicated by filled markers. The other drifter is undrogued and represented by empty markers. We consult these drifter positions from the Global Drifter Program Lumpkin and Centurioni (2019). In both panels, vectors illustrate the surrounding flow, and the colourmap indicates the relative vorticity field $\zeta(\boldsymbol{x}, t_0)$ at snapshot time $t_0$.

## 2.6 Impact on drifters

Since TRAPs highlight the most attracting regions and their cores represent objective saddle points of the unsteady surface flow, we expect them to attract and disperse drifting objects in a hyperbolic pattern. We study their transient impact on drifting objects by looking at surface drifter trajectories around TRAPs. To compare drifters and TRAPs at simultaneous timestamps, we consult 24-hourly drifter positions at UTC midnight from the Global Drifter Program (Lumpkin and Centurioni, 2019). For our study domain and period, this dataset provides 842, i.e. 328 drogued and 514 undrogued, drifter trajectories distributed over 221,979, i.e. 67,885 drogued and 154,094 undrogued, positions. We call these daily drifter positions *drifter days*. See Fig. S2 in the Supplementary Material for an overview of all drifter positions. Figure 3b illustrates an example of hyperbolic drifter motion from this dataset - drifters are first attracted perpendicular towards the TRAP and then transported along one of its

branches towards the end.

We want to see how drifters behave in the surroundings of a TRAP and detect pairs of drifters and nearby TRAPs. In Kunz (2024a), we provide a comprehensive description of our pair algorithm that works from a drifter's perspective and searches for the closest TRAP within a search radius $r_s$. This approach is equivalent to, but more efficient than, searching drifters within

a radius $r_s$ from a TRAP and discarding drifter positions related to another neighbouring but closer TRAP. For every drifter-TRAP pair, the algorithm records the drifter's current distance to the TRAP. It saves attributes like the TRAP age $\tau$ at first encounter, the TRAP lifetime $\Lambda$ and its attraction rate $s_1$ at every instance of the pair. Moreover, we know the daily vorticity pattern in the surroundings of a drifter-TRAP pair, and we measure the pair's duration, i.e. the retention time $\varphi$ of a drifter around its closest TRAP.

Since we frequently observe TRAPs within groups of four vortices, we assume that the position and size of surrounding mesoscale eddies determine the limit to which we can observe hyperbolic drifter motion around a TRAP. Considering an idealised group of four eddies with radii $\overline{r}_e$, we can capture the eddy regions that constitute the hyperbolic flow within a search radius of $r_s = \sqrt{2}\overline{r}_e$ from the TRAP core, see Fig. S4 for details. AVISO+ et al. (2022) find an average radius of $\overline{r}_e \approx 53$ km for mesoscale eddies in our domain, and we use it to set $r_s = 75$ km for the search radius of our algorithm. Smaller radii $r_s$ will

also allow the detection of hyperbolic drifter motion. However, they will not provide a complete picture up to the centre of an eddy, and they may not suffice for larger eddies, for eddies that are less adjacent than we assume, or for TRAPs that are up to 25 km off their estimated position due to the coarse resolution of our velocity data. Yet, we want to maximise the search zone to capture as many hyperbolic drifter trajectories as possible since they will occur within an abundance of motion patterns, and

we need a large dataset to develop robust statistics. Therefore, we apply $r_s = 75$ km.

Another motivation for this choice is the distance distribution of drifters around TRAPs. $r_s = 75$ km represents the average distance $\overline{d}$ plus 1 standard deviation between a drifter and its closest TRAP core, i.e. $\overline{d} \approx (51 \pm 25)$ km for drogued and $\overline{d} \approx (49 \pm 24)$ km for undrogued drifters. 86 % of drifter days occur within a 75 km distance to their closest TRAP core, see

Fig. S3 in the Supplementary Material for the respective distribution. The 14 % drifter days beyond this limit do not provide additional insights for our analysis but significantly increase the number of drifter-TRAP pairs that last for one day. Such one-day pairings also occur, but less frequently, within the search radius, due to ephemeral TRAPs with lifetimes of $\Lambda = 1$ day, due to drifters passing by in the periphery of a TRAP or due to a drifter meeting another structure in the way. We exclude these one-day pairs from our analysis since we cannot infer any useful motion statistics from them.

Because our pair algorithm searches for the closest TRAP around a drifter, its detection is insensitive to the individual attraction strength or impact range of surrounding TRAPs. The definition of a dynamic impact range would be a valuable contribution to the TRAPs concept, which we propose for future research. However, our approach here will allow us to show the aggregate effect of TRAPs on drifters and to provide a first estimate of the retention times that drifters can spend around a

TRAP.

## 3 Results

### 3.1 Spatial distribution of TRAPs

We first look at important circulation features of the study area. Figure 1 shows the distribution of eddy kinetic energy (EKE) over the domain with respect to the period 2000-2019. The lowest values of EKE occur in the northwest corner of the domain. This subregion is part of an *eddy desert* in the northeast Pacific, where mesoscale eddies are low in amplitude and short in lifetime, if present at all (Chelton et al., 2011). We find two distinct regions of high EKE in the northeast and southwest of the domain, indicating frequent turbulence and mesoscale eddy activity. These two regions neighbour the California Upwelling System (CALUS) and the North Hawaiian Ridge Current (NHRC), respectively, which are known for the production of energetic mesoscale eddies (Lindo-Atichati et al., 2020; Pegliasco et al., 2015).

Within our domain and period, we detect 4,076,065 TRAP instances from which we identify 720,391 TRAP trajectories. TRAPs occur everywhere but with distinct patterns in quantity, persistence and attraction strength. In panel (a) of Fig. 4, we separate the domain into bins of $0.25° \times 0.25°$ and show the 20-year bin averages of instantaneous TRAP core attraction rates $s_1$. A comparison with Fig. 1 reveals that TRAPs are particularly strong in regions of high EKE close to the CALUS and the NHRC. In the central-north, middle and southeast of the domain, moderate to low EKE prevails, and TRAPs are, on average, moderately to weakly attracting. The eddy desert in the northwest corner remains with a clear preference for weak TRAPs. We derive a mean attraction rate and standard deviation of $\overline{s}_1 \approx (-0.23 \pm 0.11) \text{ s}^{-1}$ over all TRAP instances. The most attracting TRAP that we find is located within the eastern hotspot of Fig. 4a and exhibits an attraction rate of $s_1 = -1.73 \text{ s}^{-1}$. We correlate the average attraction rates $s_1$ from Fig. 4a with the EKE field given in Fig. 1 and find a Pearson correlation coefficient of $r = -0.93$ with a p-value of $p < 0.001$. It indicates a strong and significant negative correlation between both variables. We infer from this that weak TRAPs, i.e. with a less negative attraction rate $s_1$, occur in regions of low EKE, while strong TRAPs occur in regions of high EKE.

In panel (b), we visualise our tracking results and show the trajectories of TRAPs with lifetimes $\Lambda > 30$ days. We call this subset *long-living* TRAPs, and we expect that these trajectories indicate westward propagation with a tendency towards the equator, considering the movements we observe in time-lapse animations of the TRAPs field, see Videos S1, S2 and S3 in Kunz (2024c). Pegliasco et al. (2015) find similar propagation characteristics for anticyclonic mesoscale eddies originating in the California Upwelling System east of our domain. A comparison to panel (a) suggests that attraction strength may vary along the trajectories of long-living TRAPs, which, however, will exclude very weak attraction rates. Indeed, long-living TRAPs are stronger than TRAPs with lifetimes $\Lambda \leq 30$ days. We find mean attraction rates of $\overline{s}_{1,\Lambda>30} \approx (-0.28 \pm 0.11) \text{ s}^{-1}$ and $\overline{s}_{1,\Lambda\leq30} \approx (-0.20 \pm 0.09) \text{ s}^{-1}$ for TRAP instances associated with these groups, respectively.

In panel (c), we separate the domain into bins of $1° \times 1°$ and count the number of TRAP trajectories that pass through each histogram bin. We find more TRAPs passing through the northwest than through the southeast half of the domain. TRAP trajec-

tories are especially abundant around the eddy desert, i.e. in the northwest and central-north of the study region. We complete the picture by deriving the average lifetime $\overline{\Lambda}$ of all trajectories that pass through a histogram bin in panel (d). We find a clear preference for ephemeral TRAPs in the northwest corner and for persistent TRAPs within the southeast half of the domain. We summarise that TRAP trajectories are very abundant but only remain for a few days around the eddy desert while they

become less abundant but more persistent towards the equator and the eastern boundary. It suggests that the underlying oceanic structures that create TRAPs show different characteristics in these two regions. Our observations are therefore consistent with the sparse occurrence of primarily weak mesoscale eddies in the eddy desert around the northern domain boundary (Chelton et al., 2011), and with the generation of energetic mesoscale eddies around the CALUS and the NHRC (Lindo-Atichati et al., 2020; Pegliasco et al., 2015), which eventually propagate through the southeastern part of the domain.

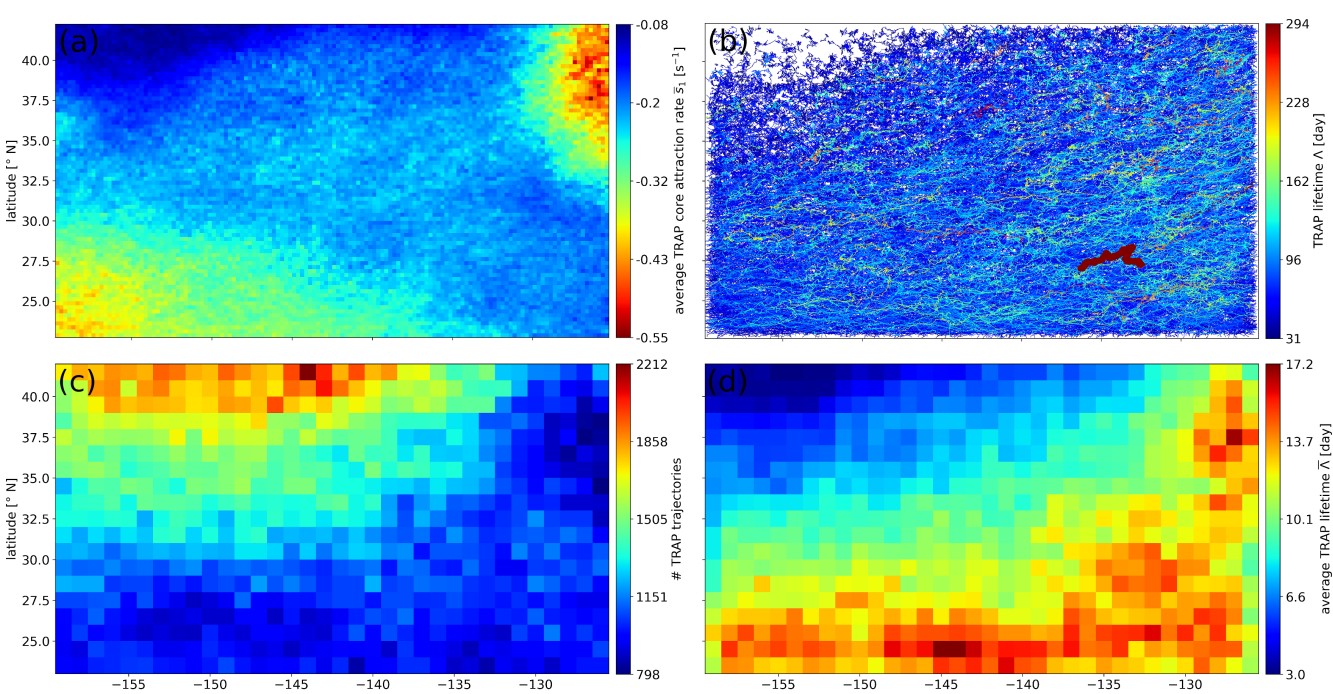

**Figure 4.** Distribution of TRAP characteristics over the domain within the period 2000-2019. (a) TRAP core attraction strength $s_1$ averaged over 20 years and all instances within bins of 0.25°×0.25° size. Red colours indicate high attraction. (b) Trajectories of long-living TRAPs with lifetimes $\Lambda > 30\,\text{days}$. Trajectories are coloured by the associated TRAP lifetime $\Lambda$, using the respective colour scale on the right. The most persistent TRAP is indicated by a thick line. (c) Number of identified TRAP trajectories that pass through histogram bins of 1°×1° size. (d) Average lifetime $\overline{\Lambda}$ of TRAP trajectories passing through each bin.

## 3.2 Life cycle and propagation

We find that TRAPs typically persist for a few days. However, the life cycle of some profiles can also span several seasons. Figure 5a presents the distribution of TRAP lifetimes $\Lambda$ over all TRAP trajectories. We find a mean lifetime of $\overline{\Lambda} \approx (6 \pm 12)\,\text{days}$.

The most persistent TRAP counts a lifetime of $\Lambda = 294 \text{ days}$ and is indicated by the thick red-brown trajectory in Fig. 4b.
Only 4 % of TRAP trajectories exhibit lifetimes of $\Lambda > 30 \text{ days}$, but they include around 41 % of all instantaneous detections.

In Fig. 5a, we also present the lifetime distribution for all mesoscale eddy detections that AVISO+ et al. (2022) find in our domain. We consult the eddy census product by AVISO+ et al. (2022) to provide a first overview of three comparable features between TRAPs and mesoscale eddies. AVISO+ et al. (2022) identify 28,645 cyclonic and 24,193 anticyclonic mesoscale eddy trajectories within our study domain and period. On average, these cyclonic eddies persist for $\overline{\Lambda}_{\oplus} \approx (24 \pm 40) \text{ days}$ and anticyclonic ones for $\overline{\Lambda}_{\ominus} \approx (29 \pm 56) \text{ days}$. The distributions in panel (a) resemble one another, but there are considerably more TRAP than eddy trajectories in our domain, and their lifetimes shift towards smaller values. Over their lifetime, approximately 25 % of TRAPs find no eddy detection, 54 % find one and 21 % find multiple eddy detections within a radius of 0.5° arclength. This cumulative number of encountered eddy detections depends on TRAP lifetime with a correlation coefficient of $r = 0.65$ and a p-value of $p < 0.001$. Long-living TRAPs encounter, on average, four eddy detections over their life cycle, while some long-living TRAPs can be related to up to 18 different eddy detections. From these statistics, we understand that the detections from AVISO+ et al. (2022) are not suitable for explaining individual TRAP detections.

However, we expect high rotation speeds of mesoscale eddies to create high strain between them, which should reflect in a relation between $s_1$ and the eddy contour speed $U$. We investigate this relation by recasting, without showing, panel (a) from Fig. 4 but for the mean eddy contour speeds $U$. We find a correlation coefficient between both histograms of $r = -0.94$ with a p-value of $p < 0.001$. It confirms that, on average, TRAP attraction strength scales with eddy contour speed. Panel (b) of Fig. 5 then views the evolution of $s_1$ and $U$ over the lifetime of long-living TRAPs and eddies. We adopt this approach from Pegliasco et al. (2015), who study the evolution of eddy radii and amplitudes. On average, both TRAP attraction and eddy contour speed intensify in the first half and decrease in the second half of a respective life cycle. We conclude that the life cycles of TRAPs and eddies relate.

Panel (c) presents the latitudinal dependence of the zonal translation speed $c_x$ for TRAPs and eddies. We find that $c_x$ is primarily negative and thus westward for both features. The mean values $\overline{c}_x$ vary with latitude and decrease towards the equator, which indicates that both phenomena propagate faster towards the west at lower latitudes. The average westward propagation ranges between $0$ and $4 \text{ cm s}^{-1}$. The close agreement between the latitudinal means of $c_x$ provides evidence that, on average, TRAPs move along with mesoscale eddies. We find similar coincidence for the latitudinal and longitudinal distribution of zonal and meridional propagation speeds $c_x$ and $c_y$, see Fig. A1. However, mesoscale eddies exhibit more extreme propagation speeds, suggesting that eddies might only create persistent strain within a specific dynamic range or around a certain combination of eddies.

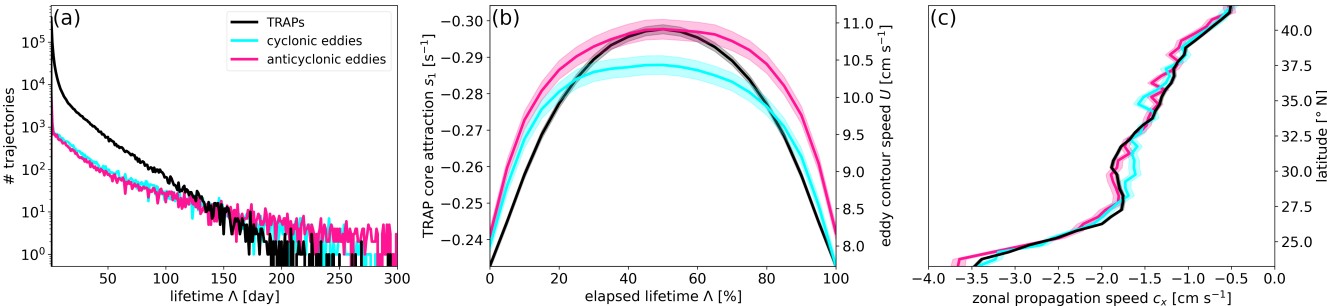

**Figure 5.** Comparisons between TRAPs and mesoscale eddies. (a) Distribution of lifetime $\Lambda$ over all TRAP trajectories, cyclonic and anticyclonic eddy trajectories in the domain. For clarity, we cut off the distributions of eddy lifetimes at 300 days. (b) Evolution of attraction rate $s_1$ over TRAP lifetime and evolution of contour speed $U$ over eddy lifetime for phenomena with lifetimes $\Lambda > 30\,\mathrm{days}$. (c) Latitudinal dependence of zonal propagation speed $c_x$ for 2,951,028 TRAP, 604,296 cyclonic and 608,650 anticyclonic eddy instances. Lines in (b) and (c) indicate bin means, shaded bands their errors w.r.t. a confidence level of 95 %. Mesoscale eddies as detected by AVISO+ et al. (2022).

Our results indicate that TRAPs are located within the mesoscale eddy field, but we wonder where exactly. While TRAPs exert maximum normal attraction and induce significant deformation, i.e. strain-dominated regions, coherent eddies are characterised by water parcels rotating about a common axis within closed transport barriers. Hence, coherent eddies represent
vorticity-dominated regions which should exclude TRAPs. As a consequence, TRAPs should emerge at the eddy periphery. However, the eddy detections from AVISO+ et al. (2022) cannot explain all TRAP occurrences. We often observe TRAPs at eddy boundaries. But we also frequently find TRAPs in regions with no eddy detections or even inside eddy contours, see Fig. A2 or Video S3 for details. The latter occurs for 15 % of all TRAP detections and suggests the presence of multiple eddies within one eddy detection. Even though the eddy dataset allows us to reveal average relations between TRAPs and mesoscale
eddies, it is not suited for describing the actual dynamics around individual TRAPs. Instead, we use the relative vorticity field to characterise the flow around TRAPs.

### 3.3 Vorticity patterns around TRAPs

A considerable number of TRAPs seem to be surrounded by four vortices of alternating polarity, which exhibit perpendicular flow towards the TRAP core and tangential flow away from it w.r.t. the TRAP curve. We call this pattern a *quadrupole*.
Here, we unravel this quadrupole and its variations to demonstrate the driving mechanisms behind the formation of mesoscale TRAPs. We compute the vorticity curves $\zeta(\alpha)$ around all available TRAPs, and because we first want to show the raw signal, we do not remove background vorticities at this stage. For visualisation purposes, we normalise each curve $\zeta(\alpha)$ by its maximum absolute value to obtain $\hat{\zeta}(\alpha)$. We take this large ensemble of normalised curves $\hat{\zeta}(\alpha)$ and show its mean and standard deviation in Fig. 6. It reveals the mean pattern in the vorticity field around a TRAP.

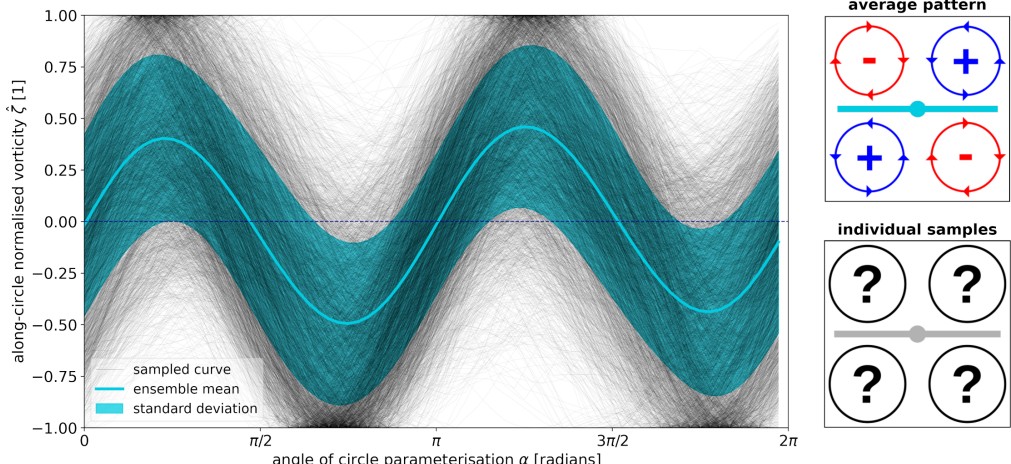

**Figure 6.** Ensemble of 3,568,850 normalised vorticity curves $\hat{\zeta}(\alpha)$ around TRAPs. Every raw vorticity curve $\zeta(\alpha)$ is normalised by its maximum absolute value to obtain $\hat{\zeta}(\alpha)$. $\hat{\zeta}(\alpha) > 0$ indicates cyclonic, $\hat{\zeta}(\alpha) < 0$ anticyclonic rotation in the surrounding flow field of a TRAP. The blue line indicates the ensemble mean, the shaded band renders its standard deviation. Black lines represent an arbitrary 0.1 % subset from this ensemble. The panels on the right illustrate two TRAPs, one surrounded by a quadrupole as suggested by the ensemble mean, the other surrounded by vortices of unknown polarity: What vorticity patterns will appear around individual samples?

The ensemble of all vorticity curves clearly resembles a sine wave. The similarity between the ensemble mean and a harmonic function, as well as its smoothness, is remarkable. Even though individual curves might not follow this shape, the entirety of the approximately 3.5 million curves generates a robust signal. Since a vorticity curve with four zero crossings and a polarity pattern of cyclonic-anticyclonic-cyclonic-anticyclonic sequence indicates a quadrupole, the above signal gives reason to believe that the mean pattern in the vorticity field around a TRAP is a quadrupole. Figure 3 illustrates such a typical quadrupole situation.

Now, we remove the background vorticity from each vorticity curve $\zeta(\alpha)$ and filter the ensemble of all vorticity curves $\zeta(\alpha)$ for the 10 distinct vortex combinations. The quadrupole pattern in Fig. 6 serves as our reference pattern, and we define the other nine patterns in terms of variation from it. Therefore, we introduce the *quadrupole order* $q$. It describes the number of vortices in a given pattern that need to change polarity in order to obtain the reference quadrupole pattern. We group the vorticity patterns of the previous 3,568,850 TRAP instances by $q$ and illustrate the most frequent groups in Fig. 7. See Fig. S5 for details on all 10 vorticity patterns and their attributed quadrupole order.

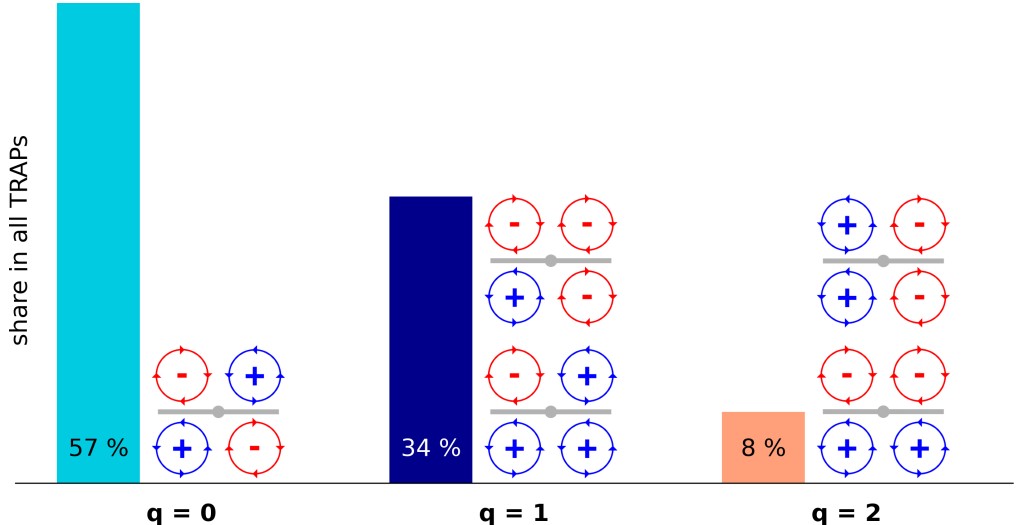

**Figure 7.** Quadrupole orders and their frequency. The quadrupole order $q$ describes the number of vortices in a given pattern that need to change polarity in order to obtain the reference quadrupole pattern. After removing the background vorticity from every vorticity curve $\zeta(\alpha)$, 99.97 % of the ensemble can be explained by the five patterns illustrated here. Figure S5 provides an overview of all vorticity patterns.

We shorthand name TRAP instances within a specific quadrupole surrounding *q-th-order quadrupoles*. Zero-, first- and second-order quadrupoles constitute 99.97 % of the signal in Fig. 6. We identify approximately 57 % of all TRAPs as zero-order quadrupoles, 34 % as first-order quadrupoles and 8 % as second-order quadrupoles. As we expect, the most prevalent group is the reference quadrupole indicated by the ensemble mean. However, a significant share of TRAPs are surrounded by a first-order quadrupole. A quick look into the ensemble means in Fig. S5 reveals that first-order quadrupoles can also include dipoles with one dominating polarity, and some second-order quadrupoles might represent symmetric dipoles. Higher-order quadrupoles also exist. They involve the lowest attraction rates, but we neglect them since they rarely occur.

Next, we want to know how quadrupole patterns evolve over the TRAP lifetime $\Lambda$ and how they relate to the attraction rate $s_1$. Panel (a) in Fig. 8 illustrates the shares of quadrupole orders zero, one and two at different evolution stages of long-living TRAPs. We observe a gradual change in the vorticity field around long-living TRAPs towards the reference quadrupole state during the first half of a life cycle and away from the reference quadrupole state during the second half. Zero-order quadrupoles are the most important vortex group throughout the entire lifetime, and they are especially abundant during the mature phase of TRAPs, i.e. between 20 % and 80 % of lifetime. However, the first-order quadrupole is a comparably probable state at the formation and decay phase of this cycle.

A comparison between Figs. 8a and 5b suggests that the evolutions of quadrupole order $q$ and TRAP attraction rate $s_1$ go along with each other, i.e. TRAPs intensify towards their mid-life while their vorticity surroundings approach a lower-order quadrupole state. From this, we expect that zero-, first- and second-order quadrupoles create different kinds of strain. Panel (b)

in Fig. 8 displays the distribution of the instantaneous attraction rates $s_1$ w.r.t. these quadrupole orders. The weakest TRAPs, i.e. with the highest strain rates $s_1$, are surrounded only by first-order quadrupoles. Zero- and second-order quadrupole environments appear for slightly higher attractions, i.e. smaller strain rates $s_1$. All three distributions peak within the same niche, but with a decreasing attraction rate, the probability densities for second- and first-order quadrupoles decline faster than for the reference quadrupole. It shows that the probability of finding strongly attracting TRAPs is higher among quadrupoles of order zero. The average attraction rates $\overline{s}_1$ confirm this tendency, and the p-values for the respective differences in mean values result in $p < 0.001$. However, the strongest TRAPs are attributed to both the zero- and first-order quadrupole environments, and we note that in panel (b), we are looking at slight differences at the extremes. We conclude that there is a trend towards the reference quadrupole environment for increasingly attractive TRAPs, while first-order quadrupoles are likewise able to induce high strain. We now investigate which of these groups organise tracer patterns particularly well.

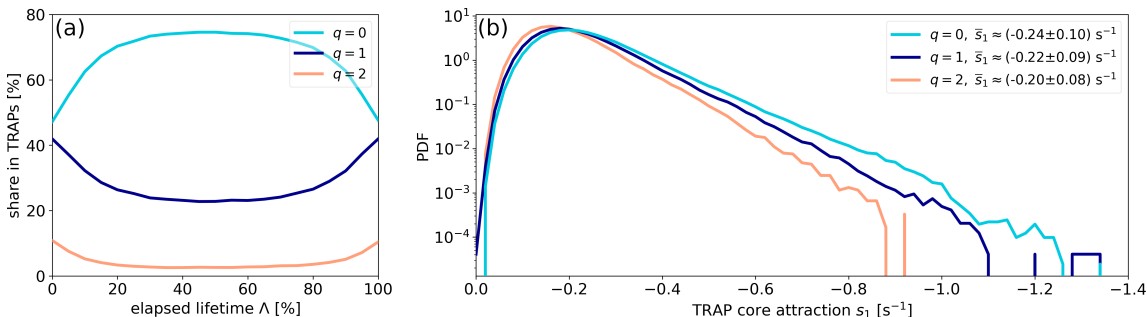

**Figure 8.** Distribution of quadrupole orders over TRAP lifetime and attraction. (a) Shares of zero-, first- and second-order quadrupoles over the lifetime of 26,675 long-living TRAPs with $\Lambda > 30$ days. (b) Distribution of TRAP attraction rate $s_1$ for all zero-, first- and second-order quadrupoles within the ensemble of vorticity curves $\zeta(\alpha)$.

## 3.4 Impact on drifters

We identify 33,878 drifter-TRAP pairs with retention times of $\varphi > 1$ day. These pairs cover 73 % of all drifter days and exhibit a mean retention time of $\overline{\varphi} \approx (4.8 \pm 3.7)$ days which reflects the transient impact of TRAPs, i.e. drifters are attracted and dispersed again within a few days. However, we also find a few drifters that spend multiple weeks around a TRAP. The highest retention time we measure is $\varphi = 46$ days. Pairs with retention times $\varphi > 7$ days represent only 9 % of all pairs but cover 28 % of all drifter days. We generally observe similar behaviours of drogued and undrogued drifters around TRAPs. But we emphasise a subtle difference in retention times to explain why we find 2.6 times more undrogued than drogued drifter-TRAP pairs while there are only 1.6 times more undrogued than drogued drifters in our domain. Figure 9 presents the distribution of retention times $\varphi$ over these drifter-TRAP pairs w.r.t. a pair's drogue state. The probability density function illustrates, irrespective of sample size, which drogue type is more likely to enter into a long retention. With increasing retention time $\varphi$, the probability density for undrogued drifters declines faster than for drogued ones. In general, drogued drifters show a higher susceptibility to stay around TRAPs for eight days and longer. Eventually, a longer retention of drogued drifters makes them

less available for transient pairings.

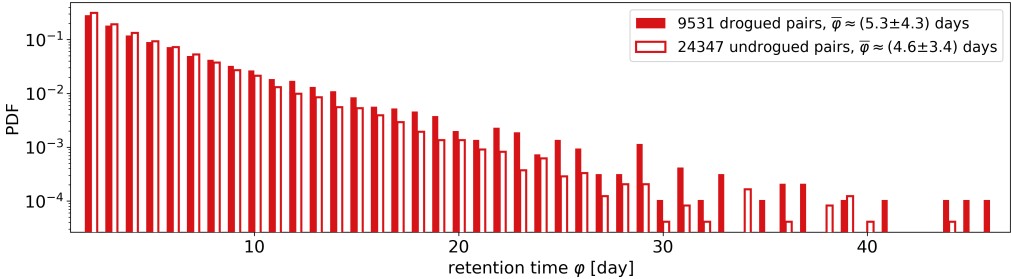

**Figure 9.** Retention times of drifters around TRAPs. Filled bars indicate the probability density of retention times $\varphi$ for drogued drifters. Empty bars present the probability density for undrogued drifters. Drogued drifters remain on average $\overline{\varphi} \approx (5.3 \pm 4.3)$ days, undrogued ones $\overline{\varphi} \approx (4.6 \pm 3.4)$ days around a TRAP. The p-value for the difference in means results in p < 0.001. Pairs with retention times of one day are excluded.

We study the average drifter motion around TRAPs and find that transport patterns depend on the evolution stage of the respective TRAP. Thus, we divide all drifter-TRAP pairs into four groups. The first group includes pairs for which the respective TRAP forms and decays during the drifter visit. Technically, this means that a pair covers the first and the last detection of a TRAP. The second group consists of pairs which start at the formation of a TRAP but end before its decay. The third group represents pairs which begin after the formation of a TRAP but end with its decay, and the fourth group defines pairs that exist throughout the lifetime of a TRAP but exclude its formation and decay. Table 1 summarises the average retention times, attraction rates and TRAP lifetimes related to these groups. We find the longest average retention time for pairs that occur throughout the life cycle of a TRAP without formation or decay. These pairs involve TRAPs of significantly stronger attraction and higher persistency compared to pairs that experience TRAP formation or decay.

**Table 1.** Characteristics of drifter-TRAP pairs at specific TRAP evolution stages. $\overline{\varphi}$ indicates the average retention time over all respective pairs, $\overline{\Lambda}$ the average lifetime over all associated TRAPs and $\overline{s}_1$ the average attraction rate over all associated TRAP instances. TRAP formation is given if a pair includes the first detection of a TRAP, and TRAP decay if a pair includes the last detection.

|  | TRAP formation | TRAP decay | $\overline{\varphi}$ [days] | $\overline{\Lambda}$ [days] | $\overline{s}_1$ [s$^{-1}$] | # pairs | share in drifter days |
|---|---|---|---|---|---|---|---|
| Group I | yes | yes | $(3.5 \pm 2.6)$ | $(3.5 \pm 2.6)$ | $(-0.17 \pm 0.07)$ | 5,456 | 8.5 % |
| Group II | yes | no | $(4.8 \pm 3.8)$ | $(19.6 \pm 20.8)$ | $(-0.21 \pm 0.08)$ | 5,521 | 11.8 % |
| Group III | no | yes | $(4.7 \pm 3.7)$ | $(19.6 \pm 20.3)$ | $(-0.21 \pm 0.08)$ | 5,609 | 11.8 % |
| Group IV | no | no | $(5.3 \pm 3.8)$ | $(47.9 \pm 35.4)$ | $(-0.27 \pm 0.11)$ | 17,292 | 40.9 % |

In Fig. 10, we illustrate the average drifter motion around TRAPs with respect to this grouping. We rotate drifter tracks around TRAPs towards the zonal axis and allocate their respective drifter positions to hexagonal bins. For every bin, we av-

erage the elapsed retention time $\varphi$, the eastward and the northward velocity components over all binned drifter positions. Bin colours indicate the average elapsed retention time $\varphi$ per bin, vectors the average velocity components.

Panel (a) highlights the average drifter motion around TRAPs that form and decay. The situation appears somewhat disor-
470 ganised, with no specific motion pattern to detect. Velocity vectors hardly show any preferred direction of drifter transport, and green colours dominate the picture. The colour map indicates that the retention times of the underlying drifter-TRAP pairs begin and end anywhere throughout the search circle. Panel (b) gives insight into drifter motion during the formation of TRAPs that persist beyond the retention time. We find that drifters mostly move away from the TRAP and in parallel to the TRAP curve. Panel (c) describes the opposite situation in which decaying TRAPs attract rather than disperse drifters. This attraction
is directed mainly towards the TRAP core. Panels (a) to (c) reveal that drifters around TRAPs that form or decay do not tend to follow a hyperbolic pattern. However, TRAPs that are at the final stage of their life cycle can at least indicate the confluence of material.

In contrast, the velocity vectors in panel (d), where TRAP formation and decay are excluded, describe hyperbolic transport
as expected. Drifters flow perpendicularly towards the TRAP core and tangentially away from it, with respect to the TRAP curve. We also observe distinct regions of blue and red bin colours, indicating that many drifters enter the zone perpendicular towards the TRAP and leave it at one end of the structure. A comparison with Figs. 5b, 8a and Table 1 clarifies that hyperbolic transport primarily occurs throughout the mature phase of a long-living TRAP because, at this stage, the surrounding flow organises into a quadrupole pattern that generates high strain.

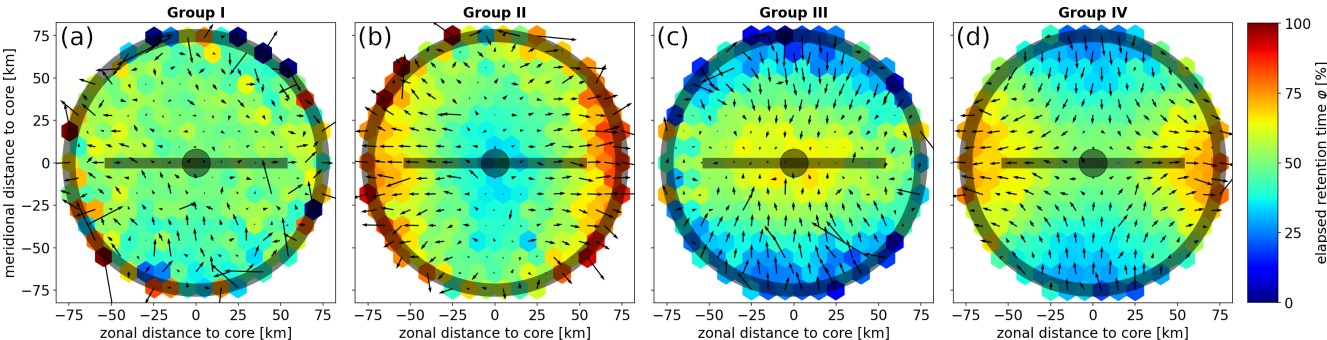

**Figure 10.** Drifter motion around TRAPs. For the first TRAP detection of every drifter-TRAP pair, we determine the angle between the vector pointing from the core to the furthest curve point and the zonal axis, with 0° pointing eastward and angles increasing counterclockwise. We use this angle to rotate all drifter positions towards the zonal axis. We allocate all rotated drifter positions to hexagonal bins. For every bin, we average the elapsed retention time $\varphi$ as well as the eastward and northward velocity components over all binned drifter instances. The colour mapping indicates bin averages of $\varphi$, vectors the average velocity components. A transparent TRAP in the middle represents a generic profile, the black circle draws the limits of the drifter search zone around it. (a) TRAP forms and decays, (b) TRAP only forms, (c) TRAP only decays and (d) TRAP neither forms nor decays during the drifter visit.

## 4    Summary, discussion and conclusion

We studied the characteristics of TRAPs and the prospects for predicting debris transport from satellite-derived mesoscale-permitting datasets. Our findings provide an advanced understanding of TRAPs in the Great Pacific Garbage Patch and demonstrate the importance of TRAP lifetime estimations to an operational application. We find that the life cycle of TRAPs can range from days to seasons with an average lifetime of $\overline{\Lambda} \approx (6 \pm 12)$ days. However, 41 % of TRAP detections relate to profiles with lifetimes of $\Lambda > 30$ days. Such long-living TRAPs exhibit a distinct evolution of attraction strength. They intensify during the first and weaken during the second half of their life cycle. At the same time, the vorticity field around TRAPs gradually changes towards and away from a specifically ordered state, i.e. a group of four vortices with alternating polarity. Therefore, the life cycle of TRAPs can explain why we observe hyperbolic drifter transport primarily throughout the mature phase of long-living TRAPs. At this stage, the surrounding flow creates the optimal vorticity pattern for generating high, hyperbolic strain. We find that hyperbolic transport around TRAPs takes on average $\overline{\varphi} \approx (5.3 \pm 3.8)$ days. In general, retention times can be very short, and strong TRAPs quickly attract and disperse material. However, we also detect a few drifters that spend multiple weeks around a TRAP. The highest retention time we measure is $\varphi = 46$ days.

We identify the evolution stage of TRAPs as the most significant predictor for drifter motion. However, the coherence of surrounding vortices might further explain our observations of hyperbolic transport and long retentions. We propose to investigate this with higher-resolution measurements of the flow, which can resolve the small-scale vortices we see around many TRAPs in Video S3. The mesoscale eddy detections by AVISO+ et al. (2022) do not include such vortices, which explains the high number of TRAPs without close-by eddy detections. Moreover, we find 15 % of TRAPs inside these eddy detections, even though the centres of the eddies are not expected to be strain-dominated regions. It is known that eddy detection algorithms based on closed contours of sea surface height often yield false-positive identifications (Andrade-Canto and Beron-Vera, 2022; Beron-Vera et al., 2013). Liu and Abernathey (2023) present an alternative algorithm which detects eddies from Lagrangian-averaged vorticity deviations (LAVD, Haller et al. (2016)). They claim the absence of false-positive detections. However, we still find TRAPs in the centres of these eddy detections, possibly due to the discrepancy between the time scale over which Lagrangian averaging takes place and the instantaneous nature of TRAPs, see Fig. A2 for details. We expect true coherence and no inclusion of TRAPs within eddy centres when we use a detection framework that descends from the deformation tensor $\mathbf{S}(\boldsymbol{x}, t)$. Serra and Haller (2016) introduce elliptic OECSs, which can be derived from singularities of $\mathbf{S}(\boldsymbol{x}, t)$ and build a complement to the strain-dominated regions uncovered by TRAPs, with the additional benefit of both methods being applicable to Eulerian snapshots of velocity. A computational implementation of elliptic OECSs would be the subject of future research.

There is an ongoing debate about whether coherent mesoscale eddies accumulate and transport floating material. van Sebille et al. (2020) discuss confirming examples such as Brach et al. (2018), Budyansky et al. (2015) and Dong et al. (2014). Early et al. (2011) use the zero contour of relative vorticity to define eddy boundaries in an idealised flow. They show how an anticyclonic eddy core transports floats and tracers over large distances, but they also explain why fluid from the outside cannot be

entrained by the core. The authors illustrate that the ring of fluid around an eddy core simultaneously entrains and sheds fluid from and into the environment, dispersing material across different scales. Abernathey and Haller (2018) argue that transport by coherent eddies is negligible and that material transport is caused by stirring and filamentation at the periphery of strictly coherent eddies rather than by the coherent motion within eddy cores. They emphasise the need for objective methods to identify such peripheral regions. Froyland et al. (2015) demonstrate an objective approach using finite-time coherent sets from a transfer operator that minimises mass loss from eddy boundaries. They track the long-term decay of an observed Agulhas ring and estimate the proportion of surface water leaking from this coherent eddy. Using their theory, Denes et al. (2022) derive finite-time coherent sets from a dynamic Laplace operator and estimate the material transport provided by the periphery of a modelled Agulhas ring. They show that the quasi-coherent outer ring of this eddy significantly contributes to the entrainment and retention of fluid. TRAPs are intrinsic to these peripheral regions, and the concept should facilitate further understanding of these processes.

In our study domain, the average surface area within a mesoscale eddy is $\overline{A}_e \approx 8361 \text{ km}^2$, as derived from eddy detections by AVISO+ et al. (2022). Even if mesoscale eddies accumulate floating material in their interior, there is no preferential region within that area expediting debris collection. For a similar area, however, TRAPs maximise normal attraction of nearby trajectories, which then move tangentially to a TRAP. Surrounding material will move towards the TRAP, aggregate and then move along the TRAP towards its ends. Because the circulation aggregates material from *both* sides of a TRAP into a smaller subarea, we expect the density of debris along a TRAP, i.e. the number of debris per unit area, to be considerably greater than in its periphery. Eventually, the hyperbolic flow would convey this aggregated debris into a strategically placed cleanup system. For these reasons, navigating a cleanup system along mesoscale TRAPs could be more productive than navigating it through mesoscale eddies.

We demonstrate this aggregation with Fig. 10d, where the hyperbolic flow transports drifters into a smaller subarea. As illustrated in these rotated scenes, there are two pathways for searching debris, i.e. the western and eastern branches of the TRAP, each supplied with material from the north and south. However, we can only show this effect using a composite of many drifter trajectories in Fig. 10d. Individual examples of a TRAP attracting multiple drifters are rare due to the low number of drifters in the domain. Fig. 3b presents one of the few observations with two drifters. Although the number of drifters allows us to show the impact of TRAPs on individual, nearby drifters, it is insufficient to quantify the likelihood of drifters aggregating around mesoscale TRAPs and other regions of the flow, such as mesoscale eddies. We illustrate this deficiency with a time series for the number of daily drifter positions spent around TRAPs and within mesoscale eddies in Fig. S6 of the Supplementary Material. The high standard deviations for respective drifter counts result from the low number of drifters and prevent an accurate comparison. We leave the time series as a motivation for future studies. Prospective research could investigate the likelihood of aggregation using a significantly higher number of drifters or appropriate measurements of debris concentrations, the latter being available soon from current missions.

There are obvious limits to the application of this mesoscale permitting dataset. The effective resolution in space and time should be coarser than the 0.25° latitude-longitude grid and the daily frequency. We observe the effects of this in animations where TRAPs reappear after a one-day gap. At such a gap, our tracking algorithm defines a new trajectory, and therefore, we might underestimate TRAP lifetimes. Similarly, detection gaps affect the identification of drifter-TRAP pairs, leading to an underestimation of retention times. However, we find a remarkable consistency between Duran et al. (2021) and our study.

They find persistent mesoscale TRAPs that predict the spread of surface oil at least eight days in advance, in agreement with our finding of retention times of $\overline{\varphi} = (5.3 \pm 3.8)$ days for hyperbolic drifter motion. These comparable time scales and the overall similar behaviour we observe for drogued and undrogued drifters further underline the concept's robustness against differences in tracer properties.

Our analyses of drifter-TRAP pairs reveal the hyperbolic-type Lagrangian motion induced by TRAPs. These observations confirm the persistence of TRAPs over periods considerably longer than a few hours, which is the lifetime of a TRAP that is mathematically guaranteed to exist. We know from our drifter-TRAP statistics that, for about a week, drifters are attracted normally to a TRAP, then accelerate and finally leave the TRAP in a tangential direction. Given that the drifter and altimetry datasets are independent oceanic observations, these statistics show that TRAPs often persist for at least a week. Importantly,

we note that the behaviour of drifters in the vicinity of forming or decaying TRAPs is distinct from the hyperbolic behaviour observed near TRAPs that are neither forming nor decaying. This result shows that we are following the same TRAP and not following different TRAPs that quickly form and decay at locations that coincide with the path of propagating eddies. Due to the relatively short transport time of a drifter in the vicinity of a TRAP, we would expect that a TRAP can persist for considerably longer than a week. Indeed, the persistent relation that we find between TRAPs and mesoscale vorticity structures, including the similarity in their propagation speed, suggests that the lifetimes of mesoscale TRAPs are often related to the

lifetime of long-lived mesoscale structures. Duran et al. (2021) present another example of the temporal continuity of TRAPs and their influence on hyperbolic tracer deformation, again from independent observations. The hyperbolic nature of this latter deformation pattern is established in Olascoaga and Haller (2012) and Duran et al. (2018).

Even though our study cannot resolve important submesoscale processes like filamentation, Langmuir cells or submesoscale vortices, it demonstrates the capability of TRAPs to predict material transport from mesoscale observations. We observed the effect of TRAPs on drifters, both drogued and undrogued, which sample any oceanic structures found along their path, including submesoscales. Thus, our results include the aggregate effect of such flow components on the daily drifter positions that we use in the analysis, even though these flow components are not resolved by the satellite altimetry itself.

Computing TRAPs and their statistics from submesoscale velocity observations is an essential second step for introducing the concept to offshore cleanups. Submesoscale TRAPs will indicate material aggregation at the scales of a cleanup system, and they will likely show different characteristics than their mesoscale counterparts. Investigating these differences and the interaction between mesoscale and submesoscale TRAPs will advance our understanding of material aggregation across mul-

tiple scales, which bears the potential to use nowcast observations for cleanup navigation. Our algorithms offer a great chance to reapply the TRAPs concept to future high-resolution observations that will be provided by the current SWOT mission (International Altimetry Team, 2021), with a resolution about 1 order of magnitude higher than the data used here. Since the geostrophic assumption is needed to obtain a sea-surface velocity from SWOT measurements, such exploration can be complemented with additional observations in coastal regions from high-frequency radar, which gives the full and not only the geostrophic sea-surface velocity.

An interesting approach would be to study the flow around drifter-TRAP pairs with long retention times. These long retention times might be due to drifter trapping within submesoscale vortices and filaments that result from instabilities at mesoscale fronts (van Sebille et al., 2020; Zhang et al., 2019; Gula et al., 2014). Mesoscale TRAPs indicate these mesoscale fronts and might provide a window to enhanced material clustering at the submesoscale, where we also expect higher numbers of small-scale TRAPs. The large-scale confluence we observe around mesoscale TRAPs could further supply these submesoscales with material. More research is needed to clarify whether mesoscale TRAPs from large-scale observations of sea surface height $\mathcal{O}(10-100 \text{ km})$ can be used as a proxy for material accumulation at operational scales $\mathcal{O}(1-10 \text{ km})$.

Our results can support cleanup operations in the Great Pacific Garbage Patch since they reveal which TRAPs indicate a large-scale confluence of drifting objects. Mesoscale-permitting flow observations, like the satellite data applied here, are available in near-real-time, and operators should use them to search for long-living TRAPs that are at an advanced stage of their life cycle. These TRAPs can predict material aggregation since they streamline floating objects into hyperbolic pathways. Such a streamlined bypass involves a short but strong attraction, which can be exploited to *filter* the flow around a TRAP. The state-of-the-art cleanup system consists of a two-kilometres-long surface barrier, towed by two vessels (The Ocean Cleanup, 2023), and it could move along TRAPs to act like a filter on the through-flowing water. The large-scale navigation along mesoscale TRAPs could then be complemented by vessel-based methods that enable the detection of debris and attracting flow features on the small scales, for instance, through automated object recognition using cameras (de Vries et al., 2021) or shipboard marine X-band radar (Lund et al., 2018). Prospective research about meso- and submesoscale TRAPs will further contribute to bridging observational gaps between these scales.

Eventually, this research is not limited to the subject of marine debris, and various offshore applications can benefit from the detection and tracking of these hyperbolic structures. For instance, authorities might use TRAPs to mitigate sargassum transport towards ports and coastal areas where beaching events cause limited accessibility. In case TRAPs indicate enhanced small-scale clustering of organic compounds, biologists could use them to monitor and protect the foraging of marine species. Oceanographers can apply the TRAPs concept to optimise drifter deployments whenever drifters are supposed to separate fast or remain in a specific region. Furthermore, TRAPs are suitable for estimating oil transport at the ocean surface, making them a considerable tool for the emergency response to oil spills. And finally, a better understanding of TRAPs will help establish their use in the essential search and rescue operations that are ever-saving lives at sea.

 **Appendix A: Figures**

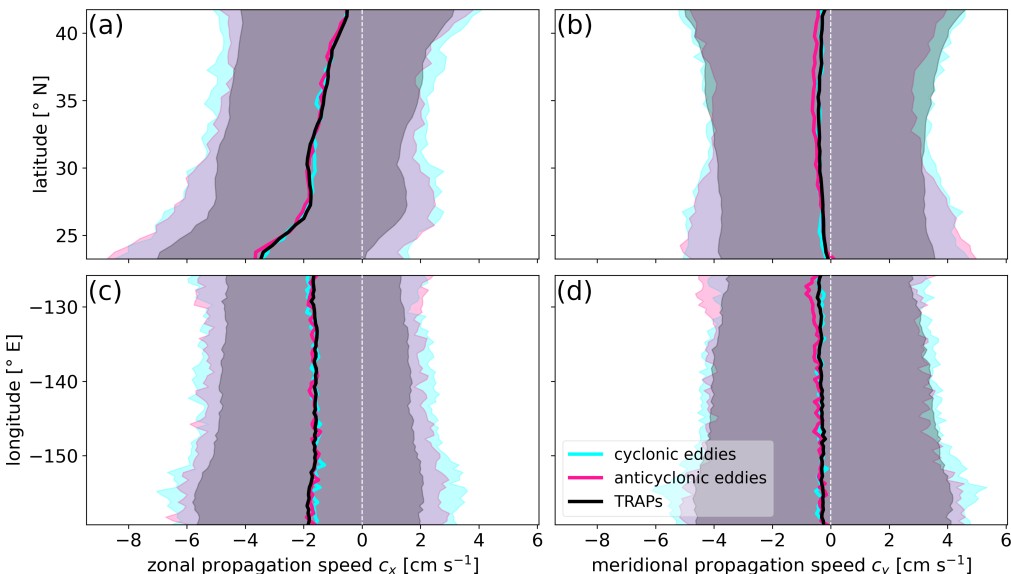

**Figure A1.** Comparison of propagation speeds between TRAPs and mesoscale eddies. Latitudinal dependence of the (a) zonal propagation speed $c_x$ and (b) meridional propagation speed $c_y$. Longitudinal dependence of the (c) zonal propagation speed $c_x$ and (d) meridional propagation speed $c_y$. Lines indicate bin means, shaded bands the respective standard deviations from 2,951,028 TRAP, 604,296 cyclonic and 608,650 anticyclonic eddy instances. Mesoscale eddies as detected by AVISO+ et al. (2022).

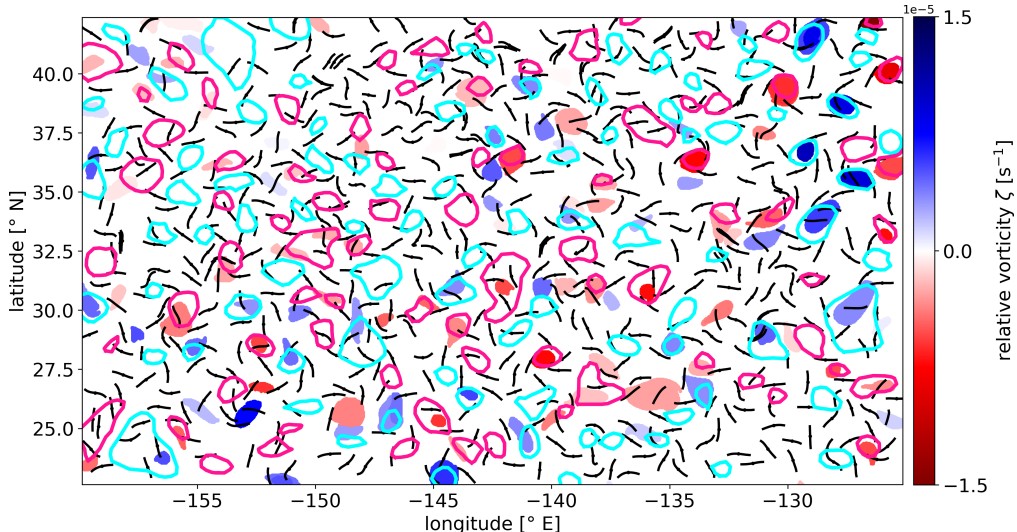

**Figure A2.** Snapshot of mesoscale TRAPs and mesoscale eddy detections within the study domain. Black lines represent TRAPs. Cyan/pink lines depict the speed contours of cyclonic/anticyclonic mesoscale eddies as detected by AVISO+ et al. (2022). Blue/red patches depict Lagrangian particles trapped by cyclonic/anticyclonic coherent eddy boundaries as detected by Liu and Abernathey (2022) for the same snapshot. The colour scale on the right indicates the relative vorticity $\zeta(\boldsymbol{x}_\ell, t_0)$ of Lagrangian particles at position $\boldsymbol{x}_\ell$, with $\ell \in [1, n]$ being the index for each of the $n$ Lagrangian particles at snapshot time $t_0$.

## Appendix B: Tables

**Table B1.** Benefits of TRAPs. Adopted from Serra and Haller (2016), Serra et al. (2020) and Duran et al. (2021).

| | |
|---|---|
| **predict material aggregation** | Since TRAPs attract close-by drifting objects, TRAPs computed from nowcast observations of the flow can indicate where material aggregates in the near future. |
| **use velocity snapshots** | TRAPs can be directly inferred from snapshots of the surface velocity field and thus are light in computation. They allow for input data with time gaps, and predictions can be made from a nowcast of the velocity field. |
| **avoid pitfalls of trajectory integration** | TRAPs do not require costly trajectory integrations, tracer release times, locations or the length of the observation period. Velocity errors do not accumulate, which should make TRAPs less sensitive to uncertainties in the underlying velocity field. TRAPs also provide full domain coverage. Lagrangian methods, on the other hand, become computationally demanding with increasing resolution of the initial particle field where particles may eventually leave a finite-size domain. |
| **indicate direction of transport** | What makes TRAPs special is that they indicate the directions of transport while other structures like $s_1(\boldsymbol{x},t)$ minima, divergence minima, or ridges of the Finite-Time Lyapunov Exponent (FTLE) field don't. FTLE ridges do not necessarily indicate the direction of largest stretching (Haller, 2015). |
| **easy to interpret** | TRAPs describe localised, one-dimensional curves of attraction in contrast to large open sets of, e.g., horizontal divergence or particle density. |
| **robust** | TRAPs are robust to different inertia and windage effects and thus are insensitive to the varying shape or submergence level of drifting objects. |
| **uncover hidden flow structures** | TRAPs can be perpendicular to streamlines and exist in divergence-free flows. |
| **observer-independent** | The objective nature of TRAPs leads to the same conclusions on different platforms. Classic Eulerian quantities like streamlines, velocity magnitude, velocity gradient, energy, or relative vorticity are not objective and will lead to different results in different frames of reference. |
| **scale-invariant** | TRAPs can be computed for velocity fields of any temporal and spatial resolution. The concept can be applied to the scales of the confluence phenomena of interest. |

**Table B2.** Algorithm to compute Transient Attracting Profiles in a two-dimensional flow. Adopted from Serra and Haller (2016) and Serra et al. (2020).

---

**Input**: A two-dimensional Eulerian velocity field $\boldsymbol{u}(\boldsymbol{x},t)$ at the current time $t = t_0$, defined on a rectangular grid over the $\boldsymbol{x} = (x_1, x_2)$ coordinates.

**Compute**

1. the instantaneous rate-of-strain tensor $\mathbf{S}(\boldsymbol{x},t_0) = \frac{1}{2}(\boldsymbol{\nabla}\boldsymbol{u}(\boldsymbol{x},t_0) + [\boldsymbol{\nabla}\boldsymbol{u}(\boldsymbol{x},t_0)]^\top)$

2. the eigenvalue fields $s_1(\boldsymbol{x},t_0) \leq s_2(\boldsymbol{x},t_0)$ and the associated unit eigenvector fields $\boldsymbol{e}_i(\boldsymbol{x},t_0)$ of $\mathbf{S}(\boldsymbol{x},t_0)$ for $i = 1,2$.

3. the set $S_{1m}(t_0)$ of negative local minima of $s_1(\boldsymbol{x},t_0)$.

4. TRAPs ($\boldsymbol{e}_2$-lines) as solutions of the ODE

$$\begin{cases} \dfrac{d\boldsymbol{x}}{ds} = \mathrm{sign}\langle \boldsymbol{e}_2(\boldsymbol{x}(s),t_0), \boldsymbol{e}_2(\boldsymbol{x}(s-\Delta),t_0)\rangle \cdot \boldsymbol{e}_2(\boldsymbol{x}(s),t_0) \\ \boldsymbol{x}(0) \in S_{1m}(t_0). \end{cases}$$

where $s$ is the independent variable of the TRAP curve parameterisation $\boldsymbol{x}(s)$ at time $t_0$, $\Delta$ denotes the integration step in the arclength parameter $s$ and $\boldsymbol{x}(0)$ indicates the position of a TRAP core at time $t_0$. The sign term in step 4 guarantees the local smoothness of the direction field $\boldsymbol{e}_2(\boldsymbol{x},t_0)$. Stop integration when $s_1(\boldsymbol{x}(s),t_0)$ ceases to be montone increasing or when $s_1(\boldsymbol{x}(s),t_0) > 0.3 \cdot s_1(\boldsymbol{x}(0),t_0)$. TRAP curves are shortened such that the attraction rate along the curve is everywhere at least 30 % of the attraction at the respective core in order to ensure that TRAPs indicate a distinguished attraction relative to nearby structures (Serra et al., 2020).

**Output**: Transient Attracting Profiles at time $t_0$ and their normal attraction rate field $s_1(\boldsymbol{x},t_0)$.

---

*Code and data availability.* We provide source code at Kunz (2024a) to post-process TRAP computations, track TRAPs through the domain and identify drifter-TRAP pairs. We also provide a 20-year dataset of TRAP detections with lifetime estimations, vorticity pattern detections and our dataset of drifter-TRAP pair detections at Kunz (2024b).

*Video supplement.* We provide animations of the TRAPs tracking algorithm in Video S1, of the evolution of TRAPs, drifter positions and the relative vorticity field in Video S2, and of the evolution of TRAPs, mesoscale eddy detections and the relative vorticity field in Video S3. Videos S1, S2 and S3 are available at Kunz (2024c), and Fig. S7 summarises the details.

*Author contributions.* LK programmed the algorithms, did the analyses and wrote much of the text. AG supervised the writing process, contributed to interpreting the results and wrote text. AG and CE acquired the funding. CE contributed to editing the paper, interpreting 635 the results and wrote text. RD contributed to developing the methodology, editing the paper, interpreting the results and wrote text. BSR contributed to developing the methodology, interpreting the results and acquired computational resources. All authors contributed to the design of the experiment. LK, AG and RD prepared the revised version of the manuscript.

*Competing interests.* The authors declare that they have no conflict of interest.

*Acknowledgements.* This paper is a contribution to the project L3 (Meso- to submesoscale turbulence in the ocean) of the Collaborative 640 Research Centre TRR 181 "Energy Transfer in Atmosphere and Ocean", Projektnummer 274762653, funded by the German Research Foundation (DFG), and it has been written in collaboration with The Ocean Cleanup. The authors thank the anonymous referees and the handling editor Erik van Sebille for their careful review and helpful comments on the initial manuscript. LK has consulted the free versions of the AI tools *DeepL* and *Grammarly* for assistance in English writing.

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
