# Peer review of "Transient Attracting Profiles in the Great Pacific Garbage Patch"

_EGUsphere, 2024_

## Referee Comment (RC2)

**Review of:** *Transient Attracting Profiles in the Great Pacific Garbage Patch*, by Kunz et al.

This manuscript applies the theory of *Transient Attracting Profiles* (*TRAP*s, defined as regions of strong attraction identified from the instantaneous velocity field) to identify regions of attraction in the Great Pacific Garbage Patch. Using a 20-year long dataset, the authors track TRAPs through time, identifying regions in the garbage patch that exhibit large numbers of TRAP trajectories, regions that exhibit the longest-lived TRAP trajectories, and regions with the highest average attraction rates. They correlate the location of TRAPs to the edges of mesoscale eddies, identifying a typical quadrupole pattern of eddies around a given TRAP. They also show that drifters are typically attracted to TRAPs, with shorter retention times on average compared to TRAP lifetimes.

Overall, the manuscript provides a novel analysis and is a nice contribution to the field. Below I provide some major and minor comments that I think would help to improve the manuscript.

**Major comments:**
  1. Temporal continuity of TRAPs
The most important issue to address is the temporal continuity of TRAPs. TRAPs by definition, are features that arise in the instantaneous velocity field, and the Serra et al. (2020) paper describes TRAPs as '*short-term attractors*', and that '*TRAPs necessarily persist over short times*', with an example of a TRAP existing for several hours. They used a high-spatial resolution HF Radar dataset, along with a high-resolution MIT-MSEAS forecast model with hourly output. Their focus was, of course, on the timescale of hours due to the search and rescue nature of their paper.

Lifetimes of TRAPs in this manuscript are in the timescale of days to almost a year long, and it's not clear to me how spatially proximate detections of TRAPs at consecutive timesteps necessarily determine that these TRAPs are the same object. TRAPs are, by definition, instantaneous features that '*necessarily persist over short times*' (Serra et al. (2020)). They could emerge, persist for hours, and later die, all within a day.

Can the authors provide more evidence on why TRAPs can be tracked on timescales of days (and months), when they may not exist for more than, as I understand, a few hours? Could successive TRAP identifications simply be older TRAPs decaying and newer TRAPs emerging? The comparison with drifter-TRAP pairs shows typical retention times of just a few days, with the largest retention time being 46 days, far shorter than the longest lifetime of a tracked TRAP. As it stands, I don't think there is enough in the manuscript to make that connection, and additional justification is needed.

  2. Additional mathematical rigour
**Section 2.1 Transient Attracting Profiles**. This section would benefit from a more thorough description of the theory of TRAPs. In particular, additional rigour in the mathematics is required to make the method more readable to users. As I understand, $s_i = s_i(\mathbf{x}, t)$ are, in fact, eigenvalue **fields**, and $\mathbf{e_i} = \mathbf{e_i}(\mathbf{x}, t)$ are eigenvector **fields**. The manuscript then describes $\mathbf{e_1}$-lines and $\mathbf{e_2}$-lines, along with local minima of $s_1$ and local maxima of $s_2$, which from the current description of $s_i$ and $\mathbf{e_i}$ don't make sense. This section (and later sections) would benefit from more careful notation and rigour.

  3. Spatial analysis of TRAP trajectories
**Section 3.1 Spatial distribution of TRAPs**. I like the spatial analysis, however I think it is hampered by the same problem that spatial analyses using Lagrangian approaches have. Specifically, that trajectories of TRAPs (like Lagrangian particles) that start outside of the domain

and later enter the domain (or TRAPs that start in the domain and shortly exit the domain), will be undersampled throughout their true lifetimes, and necessarily have shorter lifetimes (on average) than those that start and remain in the domain throughout their entire lifetimes. Given the size of the domain, the timescales of the largest TRAP lifetimes, and the 20-year duration of the dataset, can the authors comment on any bias this might have on the analysis?

**Minor comments:**
1. Overall, the manuscript would benefit from additional editing, for language and grammar, as some parts of the manuscript are a little hard to follow.
2. The abstract contains notation ($\overline{\Lambda}$, and $\overline{\varphi}$) which are somewhat confusing if not explained, I would stick to more plain language.
3. Throughout the manuscript, the authors describe 'attractive regions' in the flow. To be consistent with typical dynamical systems literature, these should be described as 'attracting regions', 'attractors', or 'regions of high attraction'.
4. Line 73, the authors mention 'inevitable errors'. Can they expand on what these errors are? Inevitable in the sense of those that Serra et al. (2020) mention with numerical integration schemes, or other errors?
5. In Section 2.2-2.6, the authors use a maximal arclength of 1°, a search area of $\epsilon = 0.25°$ (which corresponds to the model resolution), and a maximal drifter-TRAP pair distance of 75km. These choices seem a little arbitrary, can the authors comment on why they chose these parameters? Could one, for instance, choose a drifter-TRAP pair distance that is related to the TRAP attraction rate? Weak TRAPs may not influence debris 75km away, but strong TRAPs can?
6. The authors use a 0.25° spatial resolution velocity dataset, which is quite coarse for operational purposes. Would the authors expect similar results (and similar statistics) when using a higher resolution velocity field (e.g. 0.1° eddy-resolving, or even higher submesoscale resolving velocity fields more commonly used for operational purposes)?
7. In the discussion around Figure 4, can the authors give further explanation for why the locations of the strongest average attraction rate, number of TRAP trajectories, and largest average TRAP lifetimes don't correlate well? Could this be hampered by the major point above (point 3)?
8. On lines 425-426, the authors say the computations of OECSs and TRAPs are 'instantaneous'. Do the authors mean these computations are on 'instantaneous datasets'?
9. The paragraph on line 428 describes a debate in the community on whether mesoscale eddies accumulate and transport material, whether the transport by an eddy is largely outside of the eddy core, and whether objective methods exist that identify the periphery of an eddy. This discussion point is missing some references, and would be further enhanced with comments on the following articles which describe the transport by both the eddy core and the periphery of an eddy core:
      Early et al. (2011) (using relative vorticity in an idealised flow),
      Froyland et al. (2015) (using finite-time coherent sets from a transfer operator),
      Denes et al. (2022) (using finite-time coherent sets from a dynamic Laplace operator).
10. The manuscript suggests that the TRAP approach is useful to marine debris cleanup operators, but the analysis is mostly statistical, analysing a large set of TRAP trajectories. A nice-to-have would be a description of how operators may use the TRAP approach in their cleanup operations.
11. The manuscript would benefit from more discussion around the potential applications of the TRAP-tracking approach, mentioned in the very last line of the manuscript (lines 470-

471). The current main application mentioned is marine pollution cleanup, but a broader description of the applications (by expanding the very last line of the conclusion) may benefit a broader audience.

**A couple of spelling issues:**
1. Line 153, 'programme' should be 'program'.
2. Line 164, 'Mesoscalle' should be 'Mesoscale'.
3. Line 265, 'view' should be 'few'.
4. Line 304, 'frequenlty' should be 'frequently'.
5. Line 436, 'approx.' can just be 'approximately'.

**References:**

M. Serra, P. Sathe, I. Rypina, A. Kirincich, S.D. Ross, P. Lermusiaux, A. Allen, T. Peacock, G. Haller, Search and rescue at sea aided by hidden flow structures, Nature Communications, 11, (2020)

J. J. Early, R. M. Samelson, and D. B. Chelton, The evolution and propagation of quasigeostrophic ocean eddies, Journal of Physical Oceanography, 41, 1535 (2011).

G. Froyland, C. Horenkamp, V. Rossi, and E. van Sebille, Studying an Agulhas ring's long-term pathway and decay with finite-time coherent sets, Chaos 25, 083119 (2015).

M.C. Denes, G. Froyland, S.R. Keating, Persistence and material coherence of a mesoscale ocean eddy, Physical Review Fluids 7, 034501 (2022).

---

## Author Comment (AC1)

**Replies to referee 1: Transient Attracting Profiles in the Great Pacific Garbage Patch**

Luca Kunz[1], Alexa Griesel[1], Carsten Eden[1], Rodrigo Duran[2,3], and Bruno Sainte-Rose[4]

[1]Institute of Oceanography, Universität Hamburg, Hamburg, Germany
[2]National Energy Technology Laboratory, U.S. Department of Energy, Albany OR, USA
[3]Theiss Research, La Jolla CA, USA
[4]The Ocean Cleanup, Rotterdam, The Netherlands

**Correspondence:** Luca Kunz

We thank the referee for their careful summary of our research project and for their helpful comments on our paper.

**Summary by the referee**

The paper identifies TRAPs – objectively defined (frame-independent) analogs of hyperbolic points, in the eastern N. Pacific using a 20-year long record of mesoscale surface currents derived from SSH altimetry and wind-induced Ekman estimates. The authors present statistics of TRAPS – lifetime, propagation speed, strength of attraction, probability of occurrence etc., investigate the relative vorticity patterns around TRAPs, and look for the influence of TRAPs on real drifters passing nearby by constructing composite maps of drifter velocities in the vicinity of TRAPs.

The results suggest that TRAPs are most commonly located in regions between mesoscale eddies, have similar westward propagation speeds to mesoscale eddies, and are associated with the hyperbolic flow geometry in their vicinity. The most typical vorticity patterns around TRAPs correspond to a quadrupole – 2 cyclonic and 2 anticyclonic vortices, with a TRAP at the center between them.

The paper is easy to read and the statistics is carefully estimated and clearly presented.

My main comments are:

**1 Mesoscale velocity data**

**The mesoscale currents used for the analysis are likely only marginally applicable for predicting the small-scale distribution of garbage on a day-to-day basis and thus would only marginally help in any real cleanup efforts, which is the main motivation that the authors give for this work. I am not convinced that TRAPs and their statistics would be unchanged if the submesoscale flow features were resolved. This should be clearly explained, so as not to overstate the usability of TRAPs for real garbage cleanup.**

*reply*

We appreciate the comment and agree that in the manuscript, we should highlight that different statistics may result for TRAPs computed from submesoscale velocities. It is also true that we do not obtain any fine details from the altimetry data used here, but if submesoscale flow feautures would alter the motion of a drifter over a few days, we would see it in our drifter experiment. We could think of it as retaining an aggregate effect of submesoscale motion, which is what is relevant for our purposes and the time scales we are interested in.

We propose to replace the paragraph starting line 450 by this new version:

Even though our study cannot resolve important submesoscale processes like e.g. filamentation, Langmuir cells or submesoscale vortices, it demonstrates the capability of TRAPs to predict material transport from first-order mesoscale observations. We observed the effect of TRAPs on drifters, both drogued and undrogued, that sample any oceanic structures found along their path, including submesoscales. Thus, our results include the aggregate effect of such flow components on the daily drifter position that we use in the analysis, even though it is not resolved by the satellite altimetry itself. Our algorithms offer a great chance to reapply the TRAPs concept to future high-resolution observations that will be provided by the current SWOT mission (International Altimetry Team, 2021). TRAPs computed from submesoscale velocities will probably show different characteristics than their mesoscale counterparts but are crucial to an application of the concept during offshore cleanups. An interesting approach would be to study the flow around drifter-TRAP pairs with long retention times. These long retention times might be due to drifter trapping within submesoscale vortices and filaments that result from instabilities at mesoscale fronts (van Sebille et al., 2020; Zhang et al., 2019). Mesoscale TRAPs indicate these mesoscale fronts and might provide a window to enhanced material clustering at the submesoscale where we also expect higher numbers of small-scale TRAPs. However, it remains to be investigated whether mesoscale TRAPs from large-scale observations of sea surface height $\mathcal{O}(10-100\text{km})$ can be used as a proxy for material accumulation at operational scales $\mathcal{O}(1-10\text{km})$. Subject to current research is the recast of our TRAPs record (Kunz, 2024b) to periods with available SWOT measurements with resolution about an order of magnitude higher than the data used here. Since the geostrophic assumption is needed to obtain a sea-surface velocity from SWOT measurements, such exploration can be complemented with additional observations in coastal regions from high-frequency radar which

gives the full and not only the geostrophic sea-surface velocity. These surveys can further improve the applicability of TRAPs to offshore cleanups and their integration within a cleanup system.

**2  Parameter choices**

55  **There are several subjective choices that likely affect the statistical results. First, limiting traps to 1 deg arclength seems both arbitrary and unnecessary. I don't think it is necessary to put any limit on TRAPs lengths. It would be better to identify the full extent of a TRAP line length and analyze its statistics. Second, a 75 km radius was used for the pairing of drifters and TRAPs. Again, this seems like an arbitrary and unnecessary choice. It would be better to define the area of influence for each TRAP around the local minimum, perform statistical analysis of the basin of influence, and use**

60  **the basin of influence for pairing with drifters.**

*reply to 1° TRAP length*

Higher thresholds for TRAP lengths automatically lead to less distinction between nearby structures. A stop condition for the integration of a TRAP curve like the 30% attraction strength criterion introduced by Serra et al. (2020) is necessary to obtain

65  distinguishable TRAPs instead of infinitely long tangents to a subset of the eigenvector field $e_2(\boldsymbol{x}, t)$. This cutoff criterion makes sense physically because attraction of nearby parcels becomes negligible as distance increases away from the core. We illustrate this in Fig. 1 below where TRAPs are cutoff wherever the attraction strength falls below 30% of the core attraction, in panel (a), and where no such stop condition applies, in panel (b). Considering the profiles in panel (a), we acknowledge that a second length condition like the 1 degree arclength might then be redundant. However, our statistics refer to the position

70  and attraction of the TRAP *core* and the main diagnostics of the paper do not depend on TRAP length. There are only two definitions that are dependent on TRAP length:

1. the radius of the vorticity curve parameterisation: without a maximal arclength of 1 degree, one would have to define an upper limit for the radius to capture the vorticity field sufficiently close to the TRAP core

2. the rotation angle for the composite maps of drifter velocities in the vicinity of TRAPs: without a maximal arclength
75  of 1 degree, one would have to define for every drifter-TRAP pair the part of the TRAP curve that needs to align with the zonal axis, such that the rotation of all drifter trajectories allows to identify patterns in a composite map (or apply another mapping)

We consider the choice of 1 degree arclength acceptable for our purposes since our conclusions are insensitive to this choice. But we also recommend to drop this second condition in future studies.

80

We now highlight this in the Methods section by replacing line 137 with this new version:

We truncate TRAP curves wherever the attraction rate along the curve falls below 30% of the attraction at the respective core. This cutoff criterion makes sense physically because the attraction of nearby parcels becomes negligible as distance increases away from the core. Without cutoff, TRAPs can become indefinitely long and

85  merge with nearby structures which makes them hard to distinguish. Their converging ends then put wrong emphasis on regions between TRAP cores where the attraction rate is comparably low, see Fig. S1 for details. In

addition, we limit TRAPs to a maximal arclength of 1° since TRAP lengths are of minor importance to our analysis and the statistics will refer to the position and attraction of the TRAP *core*. This choice does not affect our main diagnostics but is optional and can be neglected in future studies.

90 We also propose to insert Fig. 1 as new Fig. S1 in the Supplementary Material.

[Figure]

**Figure 1.** TRAPs with different lengths. (a) TRAPs with a maximal arclength of 100 degrees and cutoff at 30% of core attraction strength. (b) TRAPs with a maximal arclength of 100 degrees and no cutoff.

*reply to 75 km search radius*

In the readme 'Drifter-TRAP pair detection' within the repository of the pair algorithm we state:

> 75 kilometres is roughly about 1.5 times the average speed radius of mesoscale eddy detections in this region which we consult from The altimetric Mesoscale Eddy Trajectory Atlas (META3.2 DT) provided by AVISO+ et al. (2022).

This seemed a reasonable search radius to us considering that we look for drifter movement in the periphery of eddies. We acknowledge that this choice seems arbitrary and explain our further motivation for it. We have run this algorithm for different values of the search radius $r$ ranging from 50 km to 300 km. 86% of all drifter days occur within 75 km distance to their closest TRAP core. The mean distance and its standard deviation of drogued drifters to their closest TRAP core is $\overline{d} \approx (51 \pm 25)$ km, for undrogued ones it results in $\overline{d} \approx (49 \pm 24)$ km. The mean distance between TRAP cores is around $\overline{d} \approx (78 \pm 29)$ km. Figure 2a shows the respective distributions of distances between drifter positions and their closest TRAP core as well as between TRAP core positions and their closest, neighbouring TRAP core. Panel (b) emphasises that the vast majority of drifter positions is within a radius of 75 km. Beyond this limit, data is insufficient to derive more conclusions than the ones shown in Figure 10 of the manuscript.

For the 14% of drifter days beyond the 75 km limit, we observe a significant increase in the number of one-day pairings. If drifters are really attracted in a far distance, it is likely that they meet other structures in the way which terminates the pairing algorithm. Such a one-day 'retention' becomes arguable considering the temporal resolution of the data as well as the distance to the actual structure. Short-term pairings with distant drifters can also occur when a TRAP attracts a distant drifter but dissipates during the drifter's approach. To address this, we also tested a search radius based on the remaining lifetime of a TRAP which reduced the amount of short-term pairings but did not lead to any remarkable insights. With these aspects and the coarse resolution of our data in mind, it seemed straight-forward and reasonable to us to apply a 75 km threshold from the beginning.

Our pair algorithm searches for the closest TRAP around a drifter and therefore, its detection is insensitive to the individual attraction strength or impact range of surrounding TRAPs. As a consequence, drifters may be located within the impact range of one TRAP but will be assigned to another TRAP, if the latter is closer, even if the drifter is beyond its impact range. This could lead to a short-term pairing with the closer TRAP and an underestimated retention time with the impacting TRAP, towards which the drifter will be eventually attracted. Considering the actual impact range of a TRAP could therefore improve the accuracy of retention times and motion patterns of drifters around TRAPs. The definition of a dynamic impact range would be a valuable contribution to the TRAPs concept and would also benefit its operational application. We are currently not aware of a mathematical definition for an objective 'basin of attraction' that is compatible with the concept. However, basins of attraction are a known concept from dynamical systems theory, e.g. see Strogatz (2014); Heitzig et al. (2016); Menck et al. (2013). We also think that the eigenvalue fields $s_i(\boldsymbol{x}, t)$ and eigenvector fields $\boldsymbol{e}_i(\boldsymbol{x}, t)$ contain more information that can be used for such

a definition.

This endeavour opens a variety of theoretical questions which would be better placed in a separate research project. Our observational study can serve here as an example of motivation. The approach we use is sufficient to show the aggregate effect of TRAPs on drifters and to provide a first estimate of the retention times that drifters can spend around a TRAP.

We propose to modify the paragraph starting in line 213 as follows:

We want to see how drifters behave in the surroundings of a TRAP and therefore detect pairs of drifters and nearby TRAPs. In Kunz (2024a) we provide a comprehensive description of our pair algorithm that works from a drifter's perspective and searches for the closest TRAP within a distance of 75 kilometres. We choose this limit since it represents the average distance $\overline{d}$ plus one standard deviation between a drifter and its closest TRAP core, i.e. $\overline{d} \approx (51 \pm 25)$ km for drogued and $\overline{d} \approx (49 \pm 24)$ km for undrogued drifters. 86% of all drifter days occur within 75 km distance to their closest TRAP core, see Fig. S3 in the Supplementary Material for the respective distributions. For every instance of a drifter-TRAP pair, the algorithm records the drifter's distance to the closest TRAP. It also saves attributes like e.g. the TRAP age $\tau$ at first encounter, the TRAP lifetime $\Lambda$ and its attraction rate $s_1$. Moreover, we will know the daily vorticity pattern in which a drifter-TRAP pair is embedded, and we measure the pair's duration, i.e. the retention time $\varphi$ of a drifter around its closest TRAP. A lot of pairings will only last for a day due to ephemeral TRAPs with lifetimes of $\Lambda = 1$ day, due to drifters passing by in the periphery of a TRAP or due to a drifter meeting another structure in the way. We exclude these one-day pairs from our analysis since we cannot infer any motion statistics from them. Especially, we observe a lot of one-day pairings for drifters beyond the 75 km limit. We note that our pair algorithm searches for the closest TRAP around a drifter and therefore, its detection is insensitive to the individual attraction strength or impact range of surrounding TRAPs. The definition of a dynamic impact range would, however, be a valuable contribution to the TRAPs concept that we propose for future research. Our approach here will allow us to show the aggregate effect of TRAPs on drifters and to provide a first estimate of the retention times that drifters can spend around a TRAP.

With this we propose to include Fig. 2 as new Fig. S3 within the Supplementary Material.

[Figure]

**Figure 2.** Drifter distances to TRAPs. (a) Distribution of distances between drifter positions and their closest TRAP core (red lines) as well as between TRAP core positions and their closest, neighbouring TRAP core (black line). (b) Spatial histogram of drifter position counts around their closest TRAP. Similar to Fig. 10 within the manuscript, drifter positions are rotated towards the zonal axis and counted within hexagonal bins.

**3  Drifter retention**

**The authors seem to suggest that drifters (and other floating objects like garbage) are more likely to be found near TRAPs than in other regions of the flow. However, I don't think there is any confirmation of this claim in the paper. It would be interesting to compare the statistics for the drifters occurrences and retention times within TRAPs to that within the mesoscale eddies and maybe even for a random subset of regions of comparable size and spatial distribution.**

*reply*

We would like to clarify that we are not looking for encounter probabilities of drifters around TRAPs in comparison to other regions of the flow. Rather, we are looking for regions of confluence and separation where drifters and floating objects will aggregate, even if temporarily, due to the hyperbolic structures that TRAPs identify.

We propose to correct an unfortunate word choice in the paragraph starting line 358:

> We identify 33,878 drifter-TRAP pairs with retention times of $\varphi > 1$ day. These pairs cover 73% of all drifter days and exhibit a mean retention time of $\overline{\varphi} \approx (4.8 \pm 3.7)$ days which reflects the transient impact of TRAPs, i.e. drifters are  attracted and dispersed again within a few days.

We now highlight this in the Conclusions by editing line 406ff as follows:

> Therefore, the life cycle of TRAPs can explain why we observe hyperbolic drifter transport primarily throughout the mature phase of long-living TRAPs. At this stage, the surrounding flow is particularly organised and generates high strain. We find that hyperbolic transport around TRAPs takes on average $\overline{\varphi} \approx (5.3 \pm 3.8)$ days. In general, retention times can be very short and especially strong TRAPs quickly attract and disperse material. But we also detect a few drifters that spend multiple weeks around a TRAP. The highest retention time we measure counts $\varphi = 46$ days.

It would be worthwhile to study the likelihood of drifters to be found near TRAPs. We did investigate drifter times spent around TRAPs and within mesoscale eddies. In line 434 we mention this aspect and point to Fig. S3 in the Supplementary Material (Fig. S5 in the revised Supplementary Material), Fig. 3 below, where we present the time series of drifter days identified around both structures. We acknowledge that this part needs more clarification and propose to replace the description in lines 434 - 438 by this new version:

> We derived a 20-year time series of the number of daily drifter positions spent around TRAPs and within mesoscale eddies detected by AVISO+ et al. (2022) to compare the retention of drifters by both features, see Fig. S5 for details. We find that on average, the share of drifter days around TRAPs or within eddies approximately equals the proportion of surface area covered by these structures. This suggests no preference of drifter positions for any of these features. However, the number of drifters within our domain barely supports the analysis and at times with few drifters, we observe high standard deviations in the respective shares of drifter days. Drifters become more

abundant towards the end of the time series where the share of drifter days around TRAPs tends to surpass the respective proportion of covered surface area, in contrary to eddy detections. More studies are needed to clarify if a preference of drifter positions could be observed in experiments with more drifters, a greater domain, and on what time scale this might occur.

185

[Figure]

**Figure 3.** Time series of the surface area enclosed by, and drifter positions detected within TRAP and eddy contours. (a) Time series of the daily number of drifter positions within the study domain. (b) The black line indicates the proportion of domain area covered by daily eddy speed contours. The coloured line represents the ratio of daily drifter positions within these eddy contours. (c) and (d) as (b) but for contours around TRAP cores and TRAP curves, respectively. Mesoscale eddies as detected by AVISO+ et al. (2022).

**4 Other revisions to be mentioned for disclosure**

We want to correct an unfortunate word choice in line 384:

> The situation appears rather chaotic with no specific motion pattern to detect.

190 Chaotic here is probably not a good choice because in the literature of trajectories and hyperbolic motion it has a specific meaning, a meaning that happens to be opposite to what we wish to convey. We propose to change this to:

> The situation appears rather disorganised with no specific motion pattern to detect.

We want to correct a typo in line 358 to keep it consistent with other numbers using a comma instead of a dot:

> We identify 33.878 drifter-TRAP pairs ...

195 We propose to change this to:

> We identify 33,878 drifter-TRAP pairs . . .

We will edit the the short summary in the manuscript as follows:

> TRansient Attracting Profiles (TRAPs) indicate the most attracting regions of the flow and have the potential to facilitate offshore cleanup operations in the Great Pacific Garbage Patch. We study the characteristics of TRAPs
200 and the prospects for predicting debris transport from a mesoscale permitting dataset. Our findings provide an advanced understanding of TRAPs in this particular region and demonstrate the importance of TRAP lifetime estimations to an operational application. Our TRAPs tracking algorithm complements the recently published TRAPs concept and prepares its use with high-resolution observations from the SWOT mission. Our findings may also benefit research in other fields like e.g. optimal drifter deployment, sargassum removal, the identification of
205 foraging hotspots or search and rescue.

And we apply the same corrections for the version on the article page:

> TRansient Attracting Profiles (TRAPs) indicate the most attracting regions of the flow and have the potential to facilitate offshore cleanups in the Great Pacific Garbage Patch. We study the characteristics of TRAPs and the prospects for predicting debris transport from a mesoscale permitting dataset. Our findings show the relevance of
210 TRAP lifetime estimations to an operational application and our TRAPs tracking algorithm may benefit even more challenges that are related to the search at sea.

**References**

AVISO+, CNES, SSALTO/DUACS, and IMEDEA: Mesoscale Eddy Trajectories Atlas 3.2 Delayed-Time (META3.2 DT) Allsat Version, https://doi.org/10.24400/527896/a01-2022.005.220209, 2022.

215    Heitzig, J., Kittel, T., Donges, J. F., and Molkenthin, N.: Topology of sustainable management of dynamical systems with desirable states: from defining planetary boundaries to safe operating spaces in the Earth system, Earth System Dynamics, 7, 21–50, https://doi.org/10.5194/esd-7-21-2016, 2016.

International Altimetry Team: Altimetry for the future: Building on 25 years of progress, Advances in Space Research, 68, 319–363, https://doi.org/10.1016/j.asr.2021.01.022, 25 Years of Progress in Radar Altimetry, 2021.

220    Kunz, L.: Track and analyse Transient Attracting Profiles in the Great Pacific Garbage Patch, https://github.com/kunzluca/trapsgpgp, 2024a.

Kunz, L.: Datasets to Transient Attracting Profiles in the Great Pacific Garbage Patch, https://doi.org/10.5281/zenodo.10993736, 2024b.

Menck, P., Heitzig, J., , and Kurths, J.: How basin stability complements the linear-stability paradigm, Nature Physics, 9, 89–92, https://doi.org/10.1038/nphys2516, 2013.

Serra, M., Sathe, P., Rypina, I., Kirincich, A., Ross, S. D., Lermusiaux, P., Allen, A., Peacock, T., and Haller, G.: Search and rescue at sea

225    aided by hidden flow structures, Nature Communications, 11, https://doi.org/10.1038/s41467-020-16281-x, 2020.

Strogatz, S.: Nonlinear Dynamics and Chaos: With Applications to Physics, Biology, Chemistry, and Engineering, Studies in Nonlinearity, Avalon Publishing, 2014.

van Sebille, E., Aliani, S., Law, K. L., Maximenko, N., Alsina, J. M., Bagaev, A., Bergmann, M., Chapron, B., Chubarenko, I., Cózar, A., et al.: The physical oceanography of the transport of floating marine debris, Environmental Research Letters, 15, 023 003,

230    https://doi.org/10.1088/1748-9326/ab6d7d, 2020.

Zhang, Z., Qiu, B., Klein, P., and Travis, S.: The influence of geostrophic strain on oceanic ageostrophic motion and surface chlorophyll, Nature Communications, 10, 2838, https://doi.org/10.1038/s41467-019-10883-w, 2019.

---

## Author Comment (AC2)

**Replies to referee 2: Transient Attracting Profiles in the Great Pacific Garbage Patch**

Luca Kunz[1], Alexa Griesel[1], Carsten Eden[1], Rodrigo Duran[2,3], and Bruno Sainte-Rose[4]

[1]Institute of Oceanography, Universität Hamburg, Hamburg, Germany
[2]National Energy Technology Laboratory, U.S. Department of Energy, Albany OR, USA
[3]Theiss Research, La Jolla CA, USA
[4]The Ocean Cleanup, Rotterdam, The Netherlands

**Correspondence:** Luca Kunz

We thank the referee for their careful summary of our research project and for their helpful comments on our paper.

**Summary by the referee**

This manuscript applies the theory of Transient Attracting Profiles (TRAPs, defined as regions of strong attraction identified from the instantaneous velocity field) to identify regions of attraction in the Great Pacific Garbage Patch. Using a 20-year long dataset, the authors track TRAPs through time, identifying regions in the garbage patch that exhibit large numbers of TRAP trajectories, regions that exhibit the longest-lived TRAP trajectories, and regions with the highest average attraction rates. They correlate the location of TRAPs to the edges of mesoscale eddies, identifying a typical quadrupole pattern of eddies around a given TRAP. They also show that drifters are typically attracted to TRAPs, with shorter retention times on average compared to TRAP lifetimes.

Overall, the manuscript provides a novel analysis and is a nice contribution to the field. Below I provide some major and minor comments that I think would help to improve the manuscript.

**Major comments:**

**1 Temporal continuity of TRAPs**

**The most important issue to address is the temporal continuity of TRAPs. TRAPs by definition, are features that arise in the instantaneous velocity field, and the Serra et al. (2020) paper describes TRAPs as *'short-term attractors'*, and that *'TRAPs necessarily persist over short times'*, with an example of a TRAP existing for several hours. They used a high-spatial resolution HF Radar dataset, along with a high-resolution MIT-MSEAS forecast model with hourly output. Their focus was, of course, on the timescale of hours due to the search and rescue nature of their paper.**

**Lifetimes of TRAPs in this manuscript are in the timescale of days to almost a year long, and it's not clear to me how spatially proximate detections of TRAPs at consecutive timesteps necessarily determine that these TRAPs are the same object. TRAPs are, by definition, instantaneous features that *'necessarily persist over short times'* (Serra et al. (2020)). They could emerge, persist for hours, and later die, all within a day.**

**Can the authors provide more evidence on why TRAPs can be tracked on timescales of days (and months), when they may not exist for more than, as I understand, a few hours? Could successive TRAP identifications simply be older TRAPs decaying and newer TRAPs emerging? The comparison with drifter-TRAP pairs shows typical retention times of just a few days, with the largest retention time being 46 days, far shorter than the longest lifetime of a tracked TRAP. As it stands, I don't think there is enough in the manuscript to make that connection, and additional justification is needed.**

*reply*

We appreciate the comment and agree that this is an important aspect that needs clarification. The theory behind TRAPs guarantees their existence for short periods but says nothing about their existence at larger timescales. When Serra et al. (2020) mention that TRAPs 'necessarily persist over short times' they are not implying they cannot persist for longer periods. Serra et al. (2020) chose a period of 6 hours, which is a reasonable choice for a "short" timescale (relative to typical oceanic timescales), and, importantly, a critical timescale for search and rescue operations. However, the lifespan of a TRAP depends on the oceanic structures that give rise to the hyperbolic-type Lagrangian motion that TRAPs are designed to identify. Indeed, Serra and Haller (2016) show different types of OECSs, including TRAPs, computed from altimetry data, that last at least six days. In our paper, we show that TRAPs are closely related to vorticity patterns in general, and eddy-like features in particular. Thus, we do not find it surprising that mesoscale TRAPs would have lifetimes comparable to those mesoscale features, typically measured in months and not days.

We note that our drifter-TRAP pair results are observations of the hyperbolic-type Lagrangian motion induced by TRAPs, and therefore a confirmation of the persistence of TRAPs over periods considerably longer than a few hours. We know from our drifter-TRAP statistics that, over about a week, drifters are attracted normally to a TRAP, to then accelerate and leave the TRAP

50  in a tangential direction. Given that the drifter and altimetry datasets are independent oceanic observations, we have shown that TRAPs often persist for at least a week. However, due to the relatively quick transport time of a drifter in the vicinity of a TRAP, it should be expected that a TRAP's lifetime can be considerably longer than a week. Hence, the fact that we observe small drifter retention times is not at odds with long TRAP lifetimes. Importantly, we note that the behavior of drifters in the vicinity of TRAPs that are forming or decaying is clearly distinct from the hyperbolic behavior that is observed in drifters

55  when TRAPs neither form nor decay. Thus, the trajectories of drifters in the vicinity of TRAPs proves that we are following the same TRAP and that we are not following different TRAPs that form and decay quickly at similar locations.

An independent example of satellite-observed TRAPs that persist for at least a week while inducing independently-observed tracer deformation can be found in Duran et al. (2021).

60

We will address the temporal continuity of TRAPs with a new paragraph in the Introduction between the ones starting in line 85 and line 91:

[revised manuscript text omitted]

**2  Additional mathematical rigour**

120 **Section 2.1 Transient Attracting Profiles. This section would benefit from a more thorough description of the theory of TRAPs. In particular, additional rigour in the mathematics is required to make the method more readable to users. As I understand, $s_i = s_i(x, t)$ are, in fact, eigenvalue fields, and $e_i = e_i(x, t)$ are eigenvector fields. The manuscript then describes $e_1$-lines and $e_2$-lines, along with local minima of $s_1$ and local maxima of $s_2$ , which from the current description of $s_i$ and $e_i$ don't make sense. This section (and later sections) would benefit from more careful notation and rigour.**

125

*reply*

We acknowledge that our mathematical description of the concept requires more clarification. We propose to edit the description starting in line 107 as follows:

Serra et al. (2020) derive TRAPs from the instantaneous strain field of the ocean surface using snapshots of the
130 two-dimensional surface velocity field $\boldsymbol{u}(\boldsymbol{x}, t)$. The symmetric part of the velocity gradient represents the time-dependent strain tensor $\mathbf{S}(\boldsymbol{x}, t) = \frac{1}{2}(\boldsymbol{\nabla}\boldsymbol{u}(\boldsymbol{x}, t) + [\boldsymbol{\nabla}\boldsymbol{u}(\boldsymbol{x}, t)]^{\top})$ with the eigenvalue fields $s_i(\boldsymbol{x}, t)$ and eigenvector fields $\boldsymbol{e}_i(\boldsymbol{x}, t)$. $\mathbf{S}$, $s_i$ and $\boldsymbol{e}_i$ denote the respective quantities at a fixed position $\boldsymbol{x}_0$ and time $t_0$ and we apply the notation for the diagonal form of $\mathbf{S}$ from Serra and Haller (2016):

$$\mathbf{S}\boldsymbol{e}_i = s_i\boldsymbol{e}_i, \quad |\boldsymbol{e}_i| = 1, \quad i = 1, 2; \quad s_1 \leq s_2, \quad \boldsymbol{e}_2 = \mathbf{R}\boldsymbol{e}_1, \quad \mathbf{R} := \begin{pmatrix} 0 & -1 \\ 1 & 0 \end{pmatrix} \tag{1}$$

135 The deformation of any fluid's surface element $A$ is determined by the local strain rates $s_i$ which specify the rates of stretching $(s_i > 0)$ or compression $(s_i < 0)$ of $A$ along the principle axes indicated by the local eigenvectors $\boldsymbol{e}_i$, see Olbers et al. (2012) for details. Due to the condition $s_1 \leq s_2$, the local eigenvector $\boldsymbol{e}_1$ describes the direction of minimal and $\boldsymbol{e}_2$ the direction of maximal stretching for a non-uniform deformation. The compression and stretching of surface elements translates into the attraction and repulsion of material and negative local minima
140 of $s_1(\boldsymbol{x}, t)$ therefore describe the most attracting regions of the flow, maximising attraction normal to $\boldsymbol{e}_2$ at the respective position. For incompressible conditions, $s_1 = -s_2$ further holds and local minima of $s_1(\boldsymbol{x}, t)$ simultaneously indicate local maxima of $s_2(\boldsymbol{x}, t)$. The strongest attraction and strongest repulsion then occur at the same position and in orthogonal directions.

145 TRAPs indicate the most attracting regions of the flow as they start at negative local minima of the $s_1(\boldsymbol{x}, t)$ strain field and extend tangent to the local eigenvectors $\boldsymbol{e}_2$ until the strain rate $s_1$ along the tangent ceases to be monotonically increasing. Consequently, TRAPs contain one minimum value of $s_1(\boldsymbol{x}, t)$, i.e. the point of strongest attraction perpendicular towards the TRAP. The position of this local minimum is called the TRAP *core* which represents an objective saddle-type stagnation point of the unsteady flow (Serra and Haller, 2016). The TRAP

150      itself, as it is at every point tangent to the unit eigenvectors $e_2$ describes the direction of maximal stretching and will in the following also be referred to as TRAP *curve*.

We will also differentiate more carefully between field quantities $\mathbf{S}(\boldsymbol{x},t)$, $s_i(\boldsymbol{x},t)$ and $\boldsymbol{e}_i(\boldsymbol{x},t)$ and local quantities $\mathbf{S}$, $s_i$ and $\boldsymbol{e}_i$ throughout the rest of the manuscript:

     line 128: underlying $s_1(\boldsymbol{x},t)$ strain field

155      line 130: of the $s_1(\boldsymbol{x},t)$ field

     caption Fig. 2: the colourmap indicates the $s_1(\boldsymbol{x},t)$ strain field

     row 3 Table B1: while other structures like e.g. $s_1(\boldsymbol{x},t)$ minima

**3 Spatial analysis of TRAP trajectories**

**Section 3.1 Spatial distribution of TRAPs. I like the spatial analysis, however I think it is hampered by the same prob-**
**lem that spatial analyses using Lagrangian approaches have. Specifically, that trajectories of TRAPs (like Lagrangian**
**particles) that start outside of the domain and later enter the domain (or TRAPs that start in the domain and shortly**
**exit the domain), will be undersampled throughout their true lifetimes, and necessarily have shorter lifetimes (on aver-**
**age) than those that start and remain in the domain throughout their entire lifetimes. Given the size of the domain, the**
**timescales of the largest TRAP lifetimes, and the 20-year duration of the dataset, can the authors comment on any bias**
**this might have on the analysis?**

*reply*

We acknowledge that this effect occurs within our tracking procedure. We also consider it important to clarify any bias on our
lifetime statistics since our analysis depends on TRAP lifetime. We propose to address this topic in the paragraph starting line
152 as follows:

> Our tracking algorithm runs on the full TRAPs record and finds spatially proximate detections at consecutive
> timestamps which can be identified as one single feature of the flow. The only free parameter $\epsilon$ defines the size
> of the search area around a current TRAP to look for a detection in the next snapshot and is set to $\epsilon = 0.25°$, see
> Kunz (2024) for more details. The algorithm assigns a unique label to each TRAP trajectory and its associated
> instances and it derives metrics like e.g. the lifetime $\Lambda$ of TRAPs and their age $\tau$ at a particular snapshot. The
> programme only captures the time spent *inside* the study domain and period and therefore gives rise to potential
> bias in the lifetime estimation of TRAPs that reach beyond the tempo-spatial limits of the domain. However,
> we find that only 5.4% of all TRAP trajectories are adjacent to these limits and might not entirely occur within
> the study domain, but our conclusions and particularly the TRAP lifetime distributions don't change if those
> biased trajectories are excluded, see Section 3 in the Supplementary Material where we analyse this in detail.
> With the trajectory estimation, we can now derive the zonal and meridional translation speeds $c_x$ and $c_y$ for every
> instance of a TRAP trajectory. Therefore we choose all TRAPs that persist for at least three days and average
> the forward and backward shifted velocity at a current timestamp. The forward/backward shifted velocity is the
> distance to its succeeding/preceding position divided by the time lapsed between both instances, respectively. This
> way we deliberately create no velocities at the start and end of a trajectory and do not gain propagation speeds
> for trajectories of two days lifetime. In turn, we obtain translation speeds of individual TRAP instances which we
> consider more accurate than taking the full distance travelled by a TRAP and dividing it by the respective lifetime.

Then we will include the following explanation together with Fig. 1 and Table 1 as a new Section 3 in the Supplementary
Material:

> In Section 2 of the Supplementary Material and in the documentation of our tracking algorithm (Kunz, 2024), we
> define the search area around a TRAP to look for future detections by a box reaching $\pm\epsilon$ in zonal and meridional

direction around the position of the current TRAP core. There, we also motivate our choice of $\epsilon = 0.25°$ and we use this parameter in Kunz (2024) to define a smaller $\epsilon$-domain with the new boundaries displaced by $\epsilon$ from the original domain boundaries. The boundaries of the $\epsilon$-domain are exclusive.

When a TRAP core is located outside the $\epsilon$-domain, its past/future position within the search box may be on or beyond the boundaries of the original study domain - but we do not detect TRAPs there. As a consequence, TRAP trajectories that reach beyond the $\epsilon$-domain at least once have the potential to be shortened because they might originate from outside the domain or continue there. Their lifetime would then be underestimated. But lifetimes can also be overestimated beyond the $\epsilon$-domain due to a wrong association of two close trajectories for which additional data is hidden behind the boundaries of the study domain. The algorithm corrects to some extent for the second case, see *bias circles* in Kunz (2024), while it is agnostic to the first case. Because the second case presupposes the first one, we expect that lifetimes are generally underestimated for detections beyond the $\epsilon$-domain.

A similar boundary error may occur at the temporal limits of our dataset. TRAP trajectories that start on the first or end on the last day of our record might have existed before or might continue to exist after the study period, respectively.

To estimate the bias that might result from these boundary effects, we filter the TRAPs record for trajectories that reach outside the $\epsilon$-domain or that occur on the first or the last snapshot of our record. We find that 5.4% of all TRAP trajectories fulfil one of these conditions and are therefore susceptible to a spurious lifetime estimation (with 5.2 percentage points being attributed to the spatial limits only). We flag these trajectories as potential bias and define four groups of trajectories:

1. the *biased* dataset, i.e. the original TRAPs record

2. the *bias* subset of trajectories with potentially spurious lifetime estimation

3. the *debiased* subset which only consists of trajectories that stay within the tempo-spatial limits of the experiment, i.e. the biased set excluding the bias set

4. the *corrected* dataset with lifetimes increased by 13 days for all trajectories that are part of the bias subset, 13 days is one standard deviation of the lifetime distribution within the debiased set

In Fig. 1 we present the distribution of TRAP lifetime $\Lambda$ within each of these groups. It illustrates that the bias potentially introduced by the tempo-spatial limits of the experiment, i.e. trajectories entering or leaving the domain and period, is negligible since the biased and debiased lifetime distributions in panel (a) as well as the biased and corrected distributions in panel (b) almost perfectly coincide. In panel (b), we try to compensate for lifetime underestimation by simply increasing spurious lifetimes by one standard deviation of the debiased distribution.

In Table 1 we compare a few statistics for the biased, the debiased and the corrected datasets. The subtle differences between the subsets confirm that these boundary effects are negligible and that the 5.4% potentially spurious

225     lifetime estimations do not affect the main findings of our paper. We note that our approach here is conservative and therefore will produce false positives which require an individual examination. Future studies might however try to further reduce the impact of these boundary effects by e.g. choosing a larger domain.

[Figure]

**Figure 1.** Distribution of TRAP lifetime Λ within different subsets of the TRAPs record. (a) The red line indicates the distribution within the *biased* dataset, i.e. the original TRAPs record as shown in Fig. 5 of the manuscript, the blue line represents the distribution of the *bias* subset and the black line indicates the *debiased* subset. (b) as (a) but with the black line indicating the distribution for the *corrected* dataset.

**Table 1.** Comparison between TRAP lifetime statistics from the biased (original), the debiased and the corrected datasets. Trajectories that might enter and/or leave the study domain and/or period represent the potential bias. Mean values are given together with one standard deviation $\sigma$.

| position in manuscript | metric | biased dataset | debiased dataset | corrected dataset |
|---|---|---|---|---|
| line 266 | average TRAP lifetime $\overline{\Lambda}$ | $(5.66 \pm 12.38)$ days | $(5.74 \pm 12.48)$ days | $(6.36 \pm 12.65)$ days |
| line 268 | share of long-living TRAP trajectories with $\Lambda > 30$ days | 4.3 % | 4.4 % | 4.5 % |
| line 268 | share of TRAP instances associated with long-living TRAP trajectories with $\Lambda > 30$ days | 40.5 % | 40.8 % | 41.1 % |
| line 253 | mean attraction strength $\overline{s}_1$ of instances associated with long-living TRAPs with $\Lambda > 30$ days | $(-0.283 \pm 0.111)\,\mathrm{s}^{-1}$ | $(-0.281 \pm 0.108)\,\mathrm{s}^{-1}$ | $(-0.283 \pm 0.111)\,\mathrm{s}^{-1}$ |
| line 253 | mean attraction strength $\overline{s}_1$ of instances associated with 'short'-living TRAPs with $\Lambda \leq 30$ days | $(-0.198 \pm 0.087)\,\mathrm{s}^{-1}$ | $(-0.197 \pm 0.084)\,\mathrm{s}^{-1}$ | $(-0.197 \pm 0.085)\,\mathrm{s}^{-1}$ |

**Minor comments:**

230   1. **Overall, the manuscript would benefit from additional editing, for language and grammar, as some parts of the manuscript are a little hard to follow.**

Our replies to referee 1 and referee 2 involve many editions to the original manuscript. Therefore, we will check the revised manuscript for correct spelling, grammar and language and apply necessary corrections before submitting the revised manuscript for peer-review completion and potential final publication in OS.

235   2. **The abstract contains notation ($\overline{\Lambda}$, and $\overline{\varphi}$) which are somewhat confusing if not explained, I would stick to more plain language.**

We will apply the following editions:

line 14: on average lasts for six days

line 15: TRAPs with lifetimes greater than 30 days

240   line 18: a streamlined bypass takes on average five days

3. **Throughout the manuscript, the authors describe 'attractive regions' in the flow. To be consistent with typical dynamical systems literature, these should be described as 'attracting regions', 'attractors', or 'regions of high attraction'.**

245   We will apply the following editions:

line 7: the most attracting regions of the flow

line 8: TRAPs are the attracting form of

line 26: indicate the most attracting regions of the flow

line 61: it allows to detect the most attracting regions of the flow

250   line 62: TRAPs are the attracting form of

line 116: describe the most attracting regions of the flow

line 201: highlight the most attracting regions

But we propose to remain with the following terms:

line 11: about the persistence and attractive properties

255   line 75: since the persistence and attractive properties

line 241: the most attractive TRAP

line 355: for increasingly attractive TRAPs

4. **Line 73, the authors mention 'inevitable errors'. Can they expand on what these errors are? Inevitable in the sense of those that Serra et al. (2020) mention with numerical integration schemes, or other errors?**

We mean inevitable errors in the sense of those described in Serra et al. (2020) and propose to replace the respective phrase starting line 70 as follows:

> Moreover, Serra and Haller (2016), Serra et al. (2020) and Duran et al. (2021) argue that TRAPs are more robust to moderate errors in the underlying velocity field while trajectory-based methods are susceptible to error accumulation during the velocity integration, see Table B1 for more details and benefits of the TRAPs method.

5. **In Section 2.2-2.6, the authors use a maximal arclength of 1°, a search area of $\epsilon = 0.25°$ (which corresponds to the model resolution), and a maximal drifter-TRAP pair distance of 75km. These choices seem a little arbitrary, can the authors comment on why they chose these parameters? Could one, for instance, choose a drifter-TRAP pair distance that is related to the TRAP attraction rate? Weak TRAPs may not influence debris 75km away, but strong TRAPs can?**

*1° TRAP length and 75 km search radius*

We have received similar comments from referee 1. For a statement on the 1° maximal arclength and the 75 km search radius, we would like to point referee 2 to *Section 2: Parameter choices* in our response to referee 1. The idea to dynamically relate the search distance for drifter-TRAP pairs to the TRAP attraction rate is similar to the other referee's suggestion of defining a 'basin of influence' around a TRAP. We also discuss this point in our reply to referee 1.

*choice of $\epsilon$*

We already motivate our choice of $\epsilon = 0.25°$ in the documentation of our tracking algorithm (Kunz, 2024). We now also include a new section in the Supplementary Material where the choice is motivated and explained in detail. We added in line 153ff of the paper:

> The only free parameter $\epsilon$ defines the size of the search area around a current TRAP to look for a detection in the next snapshot and is set to $\epsilon = 0.25°$. A larger value for the algorithm creates 'jumps' from a current to an unrealistically far future TRAP detection and overestimates trajectory lengths, see Section 2 in the Supplementary Material for a detailed explanation and motivation of this choice.

And we will include the following description as new Section 2 in the Supplementary Material:

> We define the search area around a TRAP to look for future detections by a box reaching $\pm\epsilon$ in zonal and meridional direction around the position of the current TRAP core. We have tested the tracking algorithm for different values of $\epsilon$ and find that the distribution of TRAP lifetimes $\Lambda$ broadens with increasing $\epsilon$ until it remains practically constant for $\epsilon \geq 0.75°$. The longest TRAP lifetime $\Lambda_{max}$ likewise plateaus for $\epsilon \geq 0.75°$. Table 2 lists the tested values of $\epsilon$ together with the respective value of $\Lambda_{max}$. The broadening of the lifetime

distribution from $\epsilon = 0.1°$ to $\epsilon = 0.75°$ occurs because small values of $\epsilon$ will lead to underestimated while large values of $\epsilon$ to overestimated TRAP trajectory lengths. In the first case, the search box is too small to capture the future position of a TRAP while in the last case, the algorithm creates 'jumps' from a current to an unrealistically far future TRAP detection. To choose a sensible $\epsilon$-value from this range, we can derive the highest possible absolute TRAP propagation speed $c_{max}(\epsilon)$ in each realisation and compare it to propagation speeds of mesoscale eddies, as e.g. given in Abernathey and Haller (2018); Chelton et al. (2011), because we expect a relation between these mesoscale flow features.

$\epsilon$ defines the maximal distance which can be tracked between a current and a future TRAP position. This distance limit ranges between $\epsilon$ in purely zonal or meridional direction and $\sqrt{2}\epsilon$ in purely northwest, southwest, southeast or northeast direction, i.e. into each corner of the box. The upper threshold for the absolute TRAP propagation speed $c_{max}(\epsilon)$ consequently depends on direction and ranges between $c_{max}^{+}(\epsilon)$ in purely zonal or meridional direction and $c_{max}^{\times}(\epsilon)$ in purely northeast, northwest, southwest or southeast direction. Practically, the algorithm allows higher propagation speeds towards intercardinal directions. The limits of this range can be approximated as follows:

$$c_{max}^{+}(\epsilon) \approx \frac{111120 \text{ m}}{1 \text{ degree arclength}} \cdot \frac{\epsilon}{86400 \text{ s}}$$

$$c_{max}^{\times}(\epsilon) = \sqrt{2} \cdot c_{max}^{+}(\epsilon)$$

A future version of the algorithm should use a search circle to remove this sensitivity to direction. Table 2 presents the values of $c_{max}^{+}$ and $c_{max}^{\times}$ for each test run. Propagation speeds of mesoscale eddies typically range below $0.2 \text{ m s}^{-1}$ and even less for the latitudes that we study (Abernathey and Haller, 2018; Chelton et al., 2011). Therefore it seemed reasonable to discard test runs with $\epsilon \geq 0.5°$ because they certainly include TRAPs with propagation speeds above $0.32 \text{ m s}^{-1}$ which is revealed by the increase of $\Lambda_{max}$ when switching from $\epsilon = 0.25°$ to $\epsilon = 0.5°$. On the other hand, $\epsilon = 0.1°$ could be too restrictive on TRAP propagation speeds because mesoscale features might, even if rarely, move with speeds above $0.13 \text{ m s}^{-1}$. Moreover, $\epsilon = 0.1°$ is below the technical resolution of our velocity data. For these reasons, we considered $\epsilon = 0.25°$ as a reasonable choice for the analysis.

**Table 2.** Values of the search box parameter $\epsilon$ for which the tracking algorithm has been tested together with the longest TRAP lifetime measured $\Lambda_{max}$ and upper threshold for absolute TRAP propagation speed $c_{max}$.

| $\epsilon$ [degree arclength] | 0.10 | 0.25 | 0.50 | 0.75 | 1.00 | 1.25 | 1.50 |
|---|---|---|---|---|---|---|---|
| $\Lambda_{max}$ [day] | 197 | 294 | 302 | 321 | 321 | 321 | 321 |
| $c_{max}^{+}$ [m s$^{-1}$] | 0,13 | 0,32 | 0,64 | 0.97 | 1,29 | 1,60 | 1,93 |
| $c_{max}^{\times}$ [m s$^{-1}$] | 0,18 | 0,46 | 0,91 | 1.36 | 1,82 | 2,27 | 2,73 |

After the tracking procedure, we computed the zonal and meridional propagation speed of individual TRAP detections which allows us to compare the distribution of absolute TRAP translation speeds $c$ with the estimated thresholds $c_{max}^{+}$ and $c_{max}^{\times}$ from Table 2 and to evaluate our choice of $\epsilon = 0.25°$. We further measured the surface geostrophic + Ekman current velocity at the position of every TRAP core which provides an additional distribution of flow velocities around these features. In Fig. 2 we show these two distributions of absolute TRAP and absolute surface velocities.

First, we see a clear difference between both distributions which indicates that TRAPs are not advected by the flow. TRAP propagation speeds are generally smaller than geostrophic + Ekman currents. Next, we see that a choice of $\epsilon = 0.1°$ would have caused an underestimation of TRAP trajectories since the respective limits of $c_{max}^{+}(0.1°)$ and $c_{max}^{\times}(0.1°)$ would cut-off the smooth tail of the distribution of TRAP propagation speeds. This is different for the choice of $\epsilon = 0.25°$ where most of the tail is preserved by $c_{max}^{+}(0.25°)$ and the distribution ends well before $c_{max}^{\times}(0.25°)$. It suggests that the search box is large enough to capture the majority of TRAP propagation speeds, i.e. future TRAP positions, in any direction. And it is small enough to prevent 'jumps' to unrealistically far future TRAP detections, in intercardinal directions, that would artificially extend the distribution up to the limit of $c_{max}^{\times}(0.25°)$. Since the highest measured TRAP translation speed lies clearly within the range between $c_{max}^{+}(0.25°)$ and $c_{max}^{\times}(0.25°)$, we expect the optimal value of $\epsilon$ within this range. Future studies could fine-tune this parameter using a search circle instead of a box. Optimising $\epsilon$ for TRAPs at different scales would be another valuable contribution since it makes the algorithm applicable to different kinds of velocity sources.

[Figure]

**Figure 2.** Distribution of absolute TRAP and surface water velocities. The red line illustrates the distribution of all measured absolute TRAP translation speeds $c$. The blue line presents the distribution of absolute geostrophic + Ekmann current surface velocities measured at *all* TRAP positions. Filled arrows indicate the maximum value of each distribution, empty arrows the upper thresholds $c_{max}^{+}(\epsilon)$ and $c_{max}^{\times}(\epsilon)$ of absolute TRAP propagation speed, displayed for three values of the search box parameter $\epsilon$. The shaded bands illustrate the range between these upper thresholds which results from the dependence of $c_{max}(\epsilon)$ on direction due to the geometry of the search box.

6. **The authors use a 0.25° spatial resolution velocity dataset, which is quite coarse for operational purposes. Would the authors expect similar results (and similar statistics) when using a higher resolution velocity field (e.g. 0.1° eddy-resolving, or even higher submesoscale resolving velocity fields more commonly used for operational purposes)?**

   We acknowledge that in the manuscript, we should highlight that different statistics can be expected for TRAPs computed from submesoscale velocities. We have received similar comments from referee 1 and we would like to point referee 2 to *Section 1: Mesoscale velocity data* in our response to referee 1. There we propose important editions to the manuscript.

7. **In the discussion around Figure 4, can the authors give further explanation for why the locations of the strongest average attraction rate, number of TRAP trajectories, and largest average TRAP lifetimes don't correlate well? Could this be hampered by the major point above (point 3)?**

   We acknowledge that this result requires more explanation. We propose to complement line 261 as follows:

   > We summarise that TRAP trajectories are very abundant but only remain for a few days around the eddy desert while they become less abundant but more persistent towards the equator and the eastern boundary. It suggests that the underlying oceanic structures that create TRAPs show different characteristics for these two regions. Our observations are therefore consistent with the sparse occurrence of weak mesoscale eddies in the eddy desert around the northern domain boundary (Chelton et al., 2011) and with the generation of energetic mesoscale eddies around the CALUS and the NHRC (Pegliasco et al., 2015; Lindo-Atichati et al., 2020), which eventually propagate through the southeastern part of the domain.

   The different spatial distributions we see in Fig. 4 of the manuscript do not effect from TRAPs that enter or leave the domain. We have demonstrated in point 3 that the 5.4% potentially spurious trajectories do not affect the main findings of our paper and therefore are not expected to have a visible impact on our spatial histograms. Moreover, panel (a) in Fig. 4 cannot be affected by this problem since it is based on TRAP instances. Panel (c) can neither be affected because it counts the number of TRAP trajectories which would remain constant for a truncation of trajectories at the domain boundaries. If panel (c) was hampered by the underestimation of lifetimes for entering or leaving TRAPs, we should see some signal of high average TRAP lifetimes $\overline{\Lambda}$ around the northeast-southwest diagonal of the domain since TRAPs are propagating westward and towards the equator. $\overline{\Lambda}$ would then decrease on both sides of this diagonal, but we do not observe such a pattern.

8. **On lines 425-426, the authors say the computations of OECSs and TRAPs are 'instantaneous'. Do the authors mean these computations are on 'instantaneous datasets'?**

   We appreciate the comment and will change line 424ff to:

   > Serra and Haller (2016) introduce elliptic OECSs which can be derived from singularities of $\mathbf{S}(\boldsymbol{x}, t)$ and build a complement to the strain-dominated regions uncovered by TRAPs, with the additional benefit of both methods being applicable to Eulerian snapshots of velocity.

9. **The paragraph on line 428 describes a debate in the community on whether mesoscale eddies accumulate and transport material, whether the transport by an eddy is largely outside of the eddy core, and whether objective methods exist that identify the periphery of an eddy. This discussion point is missing some references, and would be further enhanced with comments on the following articles which describe the transport by both the eddy core and the periphery of an eddy core:**

**Early et al. (2011) (using relative vorticity in an idealised flow),**
**Froyland et al. (2015) (using finite-time coherent sets from a transfer operator),**
**Denes et al. (2022) (using finite-time coherent sets from a dynamic Laplace operator).**

We appreciate the suggestion of these articles and will include respective comments in the paragraph starting line 428 as follows (note that the subsequent paragraph lines 434ff will also be modified as described in *Section 3: Drifter retention* of our reply to referee 1):

There is an ongoing debate on whether mesoscale eddies accumulate and transport floating material. van Sebille et al. (2020) discuss confirming examples such as Brach et al. (2018), Budyansky et al. (2015) and Dong et al. (2014). Early et al. (2011) use the zero contour of relative vorticity to define eddy boundaries in an idealised flow. They show how an anticyclonic eddy core perfectly transports floats and tracers over large distances but they also explain why fluid from the outside cannot be entrained by the core. The authors illustrate that the ring of fluid around an eddy core both entrains and sheds fluid from and into the environment and can disperse material over different scales. Abernathey and Haller (2018), however, argue that transport by coherent eddies is negligible and that material transport is caused by stirring and filamentation at the periphery of strictly coherent eddies, rather than by the coherent motion within eddy cores. They emphasise the need for objective methods to identify such peripheral regions. Froyland et al. (2015) demonstrate such an objective approach using finite-time coherent sets from a transfer operator which minimises mass loss from eddy boundaries. They are able to track the long-term decay of an observed Agulhas ring and to estimate the proportion of surface water that has leaked from this coherent structure. Using their theory, Denes et al. (2022) derive finite-time coherent sets from a dynamic Laplace operator and estimate the material transport that is provided by the periphery of a modelled Agulhas ring. They show that the quasi-coherent outer ring of this eddy significantly contributes to the entrainment and retention of fluid. TRAPs are intrinsic to these peripheral regions and the concept should facilitate further understanding of these processes.

10. **The manuscript suggests that the TRAP approach is useful to marine debris cleanup operators, but the analysis is mostly statistical, analysing a large set of TRAP trajectories. A nice-to-have would be a description of how operators may use the TRAP approach in their cleanup operations.**

11. **The manuscript would benefit from more discussion around the potential applications of the TRAP-tracking approach, mentioned in the very last line of the manuscript (lines 470-471). The current main application mentioned is marine pollution cleanup, but a broader description of the applications (by expanding the very last line of the conclusion) may benefit a broader audience.**

We will address both questions 10 and 11 with this updated final paragraph starting line 463ff:

Our results can already support offshore cleanup operations since they reveal which TRAPs are most likely to indicate the large-scale confluence of drifting objects. Operators should search for long-living TRAPs that are at an advanced stage of their life cycle. These TRAPs streamline floating objects into hyperbolic pathways. Such a streamlined bypass involves a short but strong attraction which could be exploited to *filter* the flow around a TRAP. The state-of-the-art cleanup system which operates in the Great Pacific Garbage Patch tows a two-kilometres-long surface barrier behind two vessels (The Ocean Cleanup, 2023). The system could move along TRAPs in order to act like a filter on the through-flowing water. Apparently, the scales of this system will better correspond to TRAPs computed from submesoscale observations and therefore, more research is needed to characterise TRAPs at different scales. But if enhanced submesoscale clustering can be confirmed around mesoscale TRAPs, the latter will point operations in the right direction. Moreover, this research is not limited to the subject of marine debris and various offshore applications can benefit from the detection and tracking of these hyperbolic structures. For instance, authorities might use TRAPs to mitigate sargassum transport towards ports and coastal areas where beaching events cause limited accessibility. Oceanographers can apply the TRAPs concept to optimise drifter deployments in case drifter trajectories should separate fast or remain within a specific region. If mesoscale TRAPs can indicate clustering at the submesoscale, we can expect elevated levels of organic compounds around these structures. This might help biologists to monitor and protect the foraging of pelagic species. Moreover, TRAPs make it possible to estimate oil transport at the ocean surface and might be considered in the emergency response to oil spills. Finally, and importantly, a better understanding of TRAPs will help to establish their use in the essential search and rescue operations that are constantly carried out at sea.

**A couple of spelling issues:**

1. **Line 153, 'programme' should be 'program'.**

2. **Line 164, 'Mesoscalle' should be 'Mesoscale'.**

3. **Line 265, 'view' should be 'few'.**

4. **Line 304, 'frequenlty' should be 'frequently'.**

5. **Line 436, 'approx.' can just be 'approximately'.**

We appreciate the advice and will correct the spelling issues 2 - 5 in the manuscript. The first issue arises from the fact that we write in British English.

**4 Other revisions to be mentioned for disclosure**

We want to correct an unfortunate word choice in line 384:

The situation appears rather chaotic with no specific motion pattern to detect.

Chaotic here is probably not a good choice because in the literature of trajectories and hyperbolic motion it has a specific meaning, a meaning that happens to be opposite to what we wish to convey. We propose to change this to:

The situation appears rather disorganised with no specific motion pattern to detect.

We want to correct a typo in line 358 to keep it consistent with other numbers using a comma instead of a dot:

We identify 33.878 drifter-TRAP pairs ...

We propose to change this to:

We identify 33,878 drifter-TRAP pairs . . .

We will correct another unfortunate word choice in the paragraph starting line 358:

[revised manuscript text omitted]

---

## Author Response (AR1)

**Author's response for* Transient Attracting Profiles in the Great Pacific Garbage Patch**

Luca Kunz[1], Alexa Griesel[1], Carsten Eden[1], Rodrigo Duran[2,3], and Bruno Sainte-Rose[4]

[1]Institute of Oceanography, Universität Hamburg, Hamburg, Germany
[2]National Energy Technology Laboratory, U.S. Department of Energy, Albany OR, USA
[3]Planetary Science Institute, 1700 East Fort Lowell, Tucson, AZ 85719, USA
[4]The Ocean Cleanup, Rotterdam, The Netherlands

**Correspondence:** Luca Kunz (luca.kunz@orac.earth)

**Preface**

We thank the referees for their careful summary of our research project and for their helpful comments on our paper. In the following, we provide a detailed point-by-point response to all referee comments and specify all changes in the revised manuscript. Our response to a referee comment is structured in three steps: (1) *comment* from the referee, (2) author's *reply*, (3) author's *changes* in the manuscript. In addition, we provide a marked-up manuscript version showing the changes made using the latexdiff command. This version is the reference for line numbers in the following replies.

**1 Replies to referee 1**

**1.1 Mesoscale velocity data**

*comment*
**The mesoscale currents used for the analysis are likely only marginally applicable for predicting the small-scale distribution of garbage on a day-to-day basis and thus would only marginally help in any real cleanup efforts, which is the main motivation that the authors give for this work. I am not convinced that TRAPs and their statistics would be unchanged if the submesoscale flow features were resolved. This should be clearly explained, so as not to overstate the usability of TRAPs for real garbage cleanup.**

*reply*
We appreciate the comment and agree that in the manuscript, we should highlight that different statistics may result for TRAPs computed from submesoscale velocities. It is also true that we do not obtain any fine details from the altimetry data used here, but if submesoscale flow features would alter the motion of a drifter over a few days, we would see it in our drifter experiment. We could think of it as retaining an aggregate effect of submesoscale motion, which is the relevant effect for our purposes and

the time scales we are interested in. Further, we will briefly describe a potential application of mesoscale TRAPs in combination with vessel-based methods to detect debris and submesoscale flow features in the surroundings of a cleanup operation.

*changes*
We address this with the changes made in lines 560 - 580, 582 - 595 and line 18.

**1.2 Parameter choices**

*comment*
**There are several subjective choices that likely affect the statistical results. First, limiting traps to 1 deg arclength seems both arbitrary and unnecessary. I don't think it is necessary to put any limit on TRAPs lengths. It would be better to identify the full extent of a TRAP line length and analyze its statistics.**

*reply*
Higher thresholds for TRAP lengths automatically lead to less distinction between nearby structures. A stop condition for the integration of a TRAP curve like the 30 % attraction strength criterion introduced by Serra et al. (2020) is necessary to obtain distinguishable TRAPs instead of infinitely long tangents to a subset of the eigenvector field $e_2(x,t)$. This cutoff criterion makes sense physically because the attraction of nearby parcels becomes negligible as distance increases away from the core. We will illustrate this in a new Fig. S1 in the Supplementary Material where TRAPs are cut off wherever the attraction strength falls below 30 % of the core attraction, in panel (a), and where no such stop condition applies, in panel (b). Considering the profiles in panel (a), we acknowledge that a second length condition like the 1 degree arclength might then be redundant. However, our statistics refer to the position and attraction of the TRAP *core*, and the main diagnostics of the paper do not depend on TRAP length. There are only two definitions that are dependent on TRAP length:

1. the radius of the vorticity curve parameterisation: without a maximal arclength of 1 degree, one would have to define an upper limit for the radius to capture the vorticity field sufficiently close to the TRAP core

2. the rotation angle for the composite maps of drifter velocities in the vicinity of TRAPs: without a maximal arclength of 1 degree, one would have to define for every drifter-TRAP pair the part of the TRAP curve that needs to align with the zonal axis, such that the rotation of all drifter trajectories allows to identify patterns in a composite map (or apply another mapping)

We consider the choice of 1 degree arclength acceptable for our purposes since our conclusions are insensitive to this choice. But we also recommend dropping this second condition in future studies.

*changes*
We address this with the changes made in lines 162 - 170.

We complement this by inserting the new Fig. S1 in the Supplementary Material.

*comment*

**Second, a 75 km radius was used for the pairing of drifters and TRAPs. Again, this seems like an arbitrary and unnecessary choice. It would be better to define the area of influence for each TRAP around the local minimum, perform statistical analysis of the basin of influence, and use the basin of influence for pairing with drifters.**

*reply*

In the readme 'Drifter-TRAP pair detection' within the repository of the pair algorithm we state:

> 75 kilometres is roughly about 1.5 times the average speed radius of mesoscale eddy detections in this region which we consult from The altimetric Mesoscale Eddy Trajectory Atlas (META3.2 DT) provided by AVISO+ et al. (2022).

This seemed a reasonable search radius to us, considering that we look for drifter movement in the periphery of eddies. We acknowledge that this choice seems arbitrary and explain our further motivation for it. We have run this algorithm for different values of the search radius $r$ ranging from 50 km to 300 km. 86 % of all drifter days occur within 75 km distance to their closest TRAP core. The mean distance and its standard deviation of drogued drifters to their closest TRAP core are $\overline{d} \approx (51 \pm 25)$ km, for undrogued ones it results in $\overline{d} \approx (49 \pm 24)$ km. The mean distance between TRAP cores is around $\overline{d} \approx (78 \pm 29)$ km. We will provide a new Fig. S3 in the Supplementary Material that shows the respective distributions of distances between drifter positions and their closest TRAP core, as well as between TRAP core positions and their closest, neighbouring TRAP core. Panel (b) emphasises that the vast majority of drifter positions is within a radius of 75 km. Beyond this limit, data is insufficient to derive more conclusions than the ones shown in Figure 10 of the manuscript.

For the 14 % of drifter days beyond the 75 km limit, we observe a significant increase in the number of one-day pairings. If drifters are really attracted in a far distance, it is likely that they meet other structures along the way, which terminates the pairing algorithm. Such a one-day 'retention' becomes arguable considering the temporal resolution of the data as well as the distance to the actual structure. Short-term pairings with distant drifters can also occur when a TRAP attracts a distant drifter but dissipates during the drifter's approach. To address this, we also tested a search radius based on the remaining lifetime of a TRAP, which reduced the number of short-term pairings but did not lead to any remarkable insights. With these aspects and the coarse resolution of our data in mind, it seemed straightforward and reasonable to us to apply a 75 km threshold from the beginning.

Our pair algorithm searches for the closest TRAP around a drifter, and therefore, its detection is insensitive to the individual attraction strength or impact range of surrounding TRAPs. As a consequence, drifters may be located within the impact range of one TRAP but will be assigned to another TRAP if the latter is closer, even if the drifter is beyond its impact range. This could

lead to a short-term pairing with the closer TRAP and an underestimated retention time with the impacting TRAP, towards which the drifter will be eventually attracted. Therefore, considering the actual impact range of a TRAP could improve the accuracy of retention times and motion patterns of drifters around TRAPs. The definition of a dynamic impact range would be a valuable contribution to the TRAPs concept and would also benefit its operational application. We are currently not aware of a mathematical definition for an objective 'basin of attraction' that is compatible with the concept. However, basins of attraction are a known concept from dynamical systems theory, e.g. see Strogatz (2014); Heitzig et al. (2016); Menck et al. (2013). We also think that the eigenvalue fields $s_i(\boldsymbol{x},t)$ and eigenvector fields $\boldsymbol{e}_i(\boldsymbol{x},t)$ contain more information that can be used for such a definition.

This endeavour opens a variety of theoretical questions which would be better placed in a separate research project. Our observational study can serve as an example of motivation. The approach we use is sufficient to show the aggregate effect of TRAPs on drifters and to provide a first estimate of the retention times that drifters can spend around a TRAP.

*changes*
We address this with the changes made in lines 256 - 275.
We complement this by inserting the new Fig. S3 in the Supplementary Material.

**1.3   Drifter retention**

*comment*
**The authors seem to suggest that drifters (and other floating objects like garbage) are more likely to be found near TRAPs than in other regions of the flow. However, I don't think there is any confirmation of this claim in the paper. It would be interesting to compare the statistics for the drifters occurrences and retention times within TRAPs to that within the mesoscale eddies and maybe even for a random subset of regions of comparable size and spatial distribution.**

*reply*
We would like to clarify that we are not looking for encounter probabilities of drifters around TRAPs in comparison to other regions of the flow. Rather, we are looking for regions of confluence and separation where drifters and floating objects will aggregate, even if temporarily, due to the hyperbolic structures that TRAPs identify. It would be worthwhile to study the likelihood of drifters to be found near TRAPs. We did investigate drifter times spent around TRAPs and within mesoscale eddies. In line 434 of the initial version of the manuscript, we mention this aspect and point to Fig. S3 in the initial Supplementary Material, where we present the time series of drifter days identified around both structures. We acknowledge that this part needs more clarification and will improve the description in the conclusions of the revised version.

*changes*

We address this by correcting an unfortunate word choice in line 427.

We address this with the changes made in lines 481 - 482 and 523 - 532.

Fig. S3 in the Supplementary Material of the initial manuscript becomes the new Fig. S5 in the Supplementary Material of the revised version.

**2 Replies to referee 2**

**2.1 Temporal continuity of TRAPs**

*comment*

**The most important issue to address is the temporal continuity of TRAPs. TRAPs by definition, are features that arise in the instantaneous velocity field, and the Serra et al. (2020) paper describes TRAPs as *'short-term attractors'*, and that *'TRAPs necessarily persist over short times'*, with an example of a TRAP existing for several hours. They used a high-spatial resolution HF Radar dataset, along with a high-resolution MIT-MSEAS forecast model with hourly output. Their focus was, of course, on the timescale of hours due to the search and rescue nature of their paper.**

**Lifetimes of TRAPs in this manuscript are in the timescale of days to almost a year long, and it's not clear to me how spatially proximate detections of TRAPs at consecutive timesteps necessarily determine that these TRAPs are the same object. TRAPs are, by definition, instantaneous features that *'necessarily persist over short times'* (Serra et al. (2020)). They could emerge, persist for hours, and later die, all within a day.**

**Can the authors provide more evidence on why TRAPs can be tracked on timescales of days (and months), when they may not exist for more than, as I understand, a few hours? Could successive TRAP identifications simply be older TRAPs decaying and newer TRAPs emerging? The comparison with drifter-TRAP pairs shows typical retention times of just a few days, with the largest retention time being 46 days, far shorter than the longest lifetime of a tracked TRAP. As it stands, I don't think there is enough in the manuscript to make that connection, and additional justification is needed.**

*reply*

We appreciate the comment and agree that this is an important aspect that needs clarification. The theory behind TRAPs guarantees their existence for short periods but says nothing about their existence at larger timescales. When Serra et al. (2020) mention that TRAPs 'necessarily persist over short times', they are not implying they cannot persist for longer periods. Serra et al. (2020) chose a period of six hours, which is a reasonable choice for a "short" timescale (relative to typical oceanic timescales) and, importantly, a critical timescale for search and rescue operations. However, the lifespan of a TRAP depends on the oceanic structures that give rise to the hyperbolic-type Lagrangian motion that TRAPs are designed to identify. Indeed, Serra and Haller (2016) show different types of OECSs, including TRAPs, computed from altimetry data, that last at least six days. In our paper, we show that TRAPs are closely related to vorticity patterns in general and eddy-like features in particular. Thus, we do not find it surprising that mesoscale TRAPs would have lifetimes comparable to those mesoscale features, typically measured in months and not days.

We note that our drifter-TRAP pair results are observations of the hyperbolic-type Lagrangian motion induced by TRAPs, and therefore, a confirmation of the persistence of TRAPs over periods considerably longer than a few hours. We know from our drifter-TRAP statistics that, over about a week, drifters are attracted normally to a TRAP, to then accelerate and leave the TRAP in a tangential direction. Given that the drifter and altimetry datasets are independent oceanic observations, we have shown that TRAPs often persist for at least a week. However, due to the relatively quick transport time of a drifter in the vicinity of a TRAP, it should be expected that a TRAP's lifetime can be considerably longer than a week. Hence, the fact that we observe small drifter retention times is not at odds with long TRAP lifetimes. Importantly, we note that the behaviour of drifters in the vicinity of TRAPs that are forming or decaying is clearly distinct from the hyperbolic behaviour that is observed in drifters when TRAPs neither form nor decay. Thus, the trajectories of drifters in the vicinity of TRAPs prove that we are following the same TRAP and that we are not following different TRAPs that form and decay quickly at similar locations.

An independent example of satellite-observed TRAPs that persist for at least a week while inducing independently-observed tracer deformation can be found in Duran et al. (2021).

Moreover, our study shows from large datasets that TRAPs persist over periods considerably longer than the short period for which they are mathematically guaranteed to exist. We explicitly state this result in a new paragraph of the revised manuscript.

*changes*
We address this with the changes made in lines 98 - 108 and 545 - 558.

**2.2 Additional mathematical rigour**

*comment*
**Section 2.1 Transient Attracting Profiles. This section would benefit from a more thorough description of the theory of TRAPs. In particular, additional rigour in the mathematics is required to make the method more readable to users. As I understand, $s_i = s_i(x,t)$ are, in fact, eigenvalue fields, and $e_i = e_i(x,t)$ are eigenvector fields. The manuscript then describes $e_1$-lines and $e_2$-lines, along with local minima of $s_1$ and local maxima of $s_2$ , which from the current description of $s_i$ and $e_i$ don't make sense. This section (and later sections) would benefit from more careful notation and rigour.**

*reply*
We acknowledge that our mathematical description of the concept requires more clarification. We will also differentiate more carefully between field quantities $\mathbf{S}(x,t)$, $s_i(x,t)$ and $e_i(x,t)$ and local quantities $\mathbf{S}$, $s_i$ and $e_i$ throughout the rest of the manuscript.

We address this with the changes made in lines 126 - 156, in the caption of Fig. 2 line 158, in the caption of Fig. 3 line 242, in the caption of Fig. 6 in line 377, in the caption of Fig. A2 line 606, in row 3 of Table B1 line 607 and in Table B2 line 607.

**2.3 Spatial analysis of TRAP trajectories**

*comment*

**Section 3.1 Spatial distribution of TRAPs. I like the spatial analysis, however I think it is hampered by the same problem that spatial analyses using Lagrangian approaches have. Specifically, that trajectories of TRAPs (like Lagrangian particles) that start outside of the domain and later enter the domain (or TRAPs that start in the domain and shortly exit the domain), will be undersampled throughout their true lifetimes, and necessarily have shorter lifetimes (on average) than those that start and remain in the domain throughout their entire lifetimes. Given the size of the domain, the timescales of the largest TRAP lifetimes, and the 20-year duration of the dataset, can the authors comment on any bias this might have on the analysis?**

*reply*

We acknowledge that this effect occurs within our tracking procedure. We also consider it important to clarify any bias on our lifetime statistics since our analysis depends on TRAP lifetime. We find that only 5.4 % of detected trajectories are adjacent to the limits of our study domain and period, and therefore, could be susceptible to an underestimation of trajectory length. However, this set of potentially biased trajectories has no impact on our lifetime statistics. We highlight this in the revised description of our tracking algorithm, and we point the reader to a detailed explanation together with a new Fig. S8 and Table S2 within the new Section S3 of the Supplementary Material.

*changes*

We address this with the changes made in lines 190 - 195.

We complement this by inserting the new Section S3, including a new Fig. S8 and a new Table S2, in the Supplementary Material.

**2.4 Minor comments**

*comment*

**Overall, the manuscript would benefit from additional editing, for language and grammar, as some parts of the manuscript are a little hard to follow.**

*reply*

We apply many minor revisions to improve spelling and grammar or to meet the quality standards of OS. We call these minor revisions because they aim to enhance the easiness of reading and understanding our paper without changing the meaning of the original content. Some of these revisions imply a rewording of sentences or paragraphs.

*changes*

The correction for spelling and grammar involves a large number of minor changes throughout the entire manuscript, which we don't list here for brevity.

*comment*

**The abstract contains notation ($\overline{\Lambda}$, and $\overline{\varphi}$) which are somewhat confusing if not explained, I would stick to more plain language.**

*reply*

We agree and will use plain language for the abstract.

*changes*

We address this with the changes made in lines 15 - 16 and 20 - 21.

*comment*

**Throughout the manuscript, the authors describe 'attractive regions' in the flow. To be consistent with typical dynamical systems literature, these should be described as 'attracting regions', 'attractors', or 'regions of high attraction'.**

*reply*

We agree and will change the term "attractive" to "attracting" except for the terms "attractive properties of TRAPs" and "increasingly attractive TRAPs", where "attractive" seems to be the less ambiguous choice.

*changes*

We address this with the changes made in lines 7 - 8, 12, 29, 66 - 67, 138, 244 and 294.

*comment*

**Line 73, the authors mention 'inevitable errors'. Can they expand on what these errors are? Inevitable in the sense of those that Serra et al. (2020) mention with numerical integration schemes, or other errors?**

*reply*

We mean inevitable errors in the sense of those described in Serra et al. (2020).

*changes*

We address this with the changes made in lines 78 - 79.

*comment*

**In Section 2.2-2.6, the authors use a maximal arclength of 1°, a search area of $\epsilon$ = 0.25° (which corresponds to the model resolution), and a maximal drifter-TRAP pair distance of 75km. These choices seem a little arbitrary, can the authors comment on why they chose these parameters? Could one, for instance, choose a drifter-TRAP pair distance that is related to the TRAP attraction rate? Weak TRAPs may not influence debris 75km away, but strong TRAPs can?**

*reply regarding 1° TRAP length and 75 km search radius*

We have received similar comments from referee 1. For a statement on the 1° maximal arclength and the 75 km search radius, we would like to point referee 2 to *Section 1.2: Parameter choices* in our response to referee 1. The idea to dynamically relate the search distance for drifter-TRAP pairs to the TRAP attraction rate is similar to the other referee's suggestion of defining a 'basin of influence' around a TRAP. We also discuss this point in our replies to referee 1.

*reply regarding the choice of $\epsilon$*

We already motivated our choice of $\epsilon = 0.25°$ in the documentation of our tracking algorithm (Kunz, 2024). We now also include a new section in the Supplementary Material where the choice is motivated and explained in detail. We will refer to this new section in the description of our tracking algorithm within the revised manuscript.

*changes regarding 1° TRAP length and 75 km search radius*

We address this with the changes made in lines 162 - 170 and 256 - 275.

We complement this by inserting the new Figs. S1 and S3 in the Supplementary Material.

*changes regarding the choice of $\epsilon$*

We address this with the changes made in lines 186 - 189.

We complement this by inserting the new Section S2, including the new Fig. S7 and Table S1, in the Supplementary Material.

*comment*

**The authors use a 0.25° spatial resolution velocity dataset, which is quite coarse for operational purposes. Would the authors expect similar results (and similar statistics) when using a higher resolution velocity field (e.g. 0.1° eddy-resolving,**

**or even higher submesoscale resolving velocity fields more commonly used for operational purposes)?**

*reply*

Within the manuscript, we should highlight that different statistics can be expected for TRAPs computed from submesoscale velocities. We have received similar comments from referee 1, and we would like to point referee 2 to *Section 1.1: Mesoscale velocity data* in our response to referee 1. There, we describe important editions to the manuscript.

*changes*

We address this with the changes made in lines 560 - 580, 582 - 595 and line 18.

*comment*

**In the discussion around Figure 4, can the authors give further explanation for why the locations of the strongest average attraction rate, number of TRAP trajectories, and largest average TRAP lifetimes don't correlate well? Could this be hampered by the major point above (point 3)?**

*reply*

At first sight, the different patterns in Fig. 4 do not correlate well, but they all seem connected by the generation and propagation of mesoscale eddies. We acknowledge that this result requires more explanation, which we will provide in the revised manuscript. In addition, we note that the different spatial distributions we see in Fig. 4 do not result from TRAPs that enter or leave the domain. We demonstrate in Section S3 of the Supplementary Material that the 5.4 % potentially spurious trajectories do not affect the main findings of our paper and, therefore, are not expected to have a visible impact on our spatial histograms. Moreover, panel (a) in Fig. 4 cannot be affected by this problem since it is only based on instantaneous TRAP detections. Panel (c) can neither be affected because it counts the number of TRAP trajectories, which would remain constant for a truncation of trajectories at the domain boundaries. If panel (c) was hampered by the underestimation of lifetimes for entering or leaving TRAPs, we should see some signal of high average TRAP lifetimes $\overline{\Lambda}$ around the northeast-southwest diagonal of the domain since TRAPs are propagating westward and towards the equator. $\overline{\Lambda}$ would then decrease on both sides of this diagonal, but we do not observe any sign of such a pattern.

*changes*

We address this with the changes made in lines 317 - 322.

*comment*

**On lines 425-426, the authors say the computations of OECSs and TRAPs are 'instantaneous'. Do the authors mean these computations are on 'instantaneous datasets'?**

*reply*

We confirm that we mean these computations are made on Eulerian snapshots of velocity.

*changes*

We address this with the changes made in lines 499 - 500.

*comment*

**The paragraph on line 428 describes a debate in the community on whether mesoscale eddies accumulate and transport material, whether the transport by an eddy is largely outside of the eddy core, and whether objective methods exist that identify the periphery of an eddy. This discussion point is missing some references, and would be further enhanced with comments on the following articles which describe the transport by both the eddy core and the periphery of an eddy core:**

**Early et al. (2011) (using relative vorticity in an idealised flow),**
**Froyland et al. (2015) (using finite-time coherent sets from a transfer operator),**
**Denes et al. (2022) (using finite-time coherent sets from a dynamic Laplace operator).**

*reply*

We appreciate the suggestion of these articles and will include them in our paragraph on material accumulation and transport by mesoscale eddies (note that the subsequent paragraph will also be modified as described in *Section 1.3: Drifter retention* of our reply to referee 1).

*changes*

We address this with the changes made in lines 504 - 521.

*comment*

**The manuscript suggests that the TRAP approach is useful to marine debris cleanup operators, but the analysis is mostly statistical, analysing a large set of TRAP trajectories. A nice-to-have would be a description of how operators may use the TRAP approach in their cleanup operations.**

*reply*

We will use the second last paragraph to briefly describe a potential application of mesoscale TRAPs in combination with vessel-based methods that can help to detect debris and submesoscale flow features in the surroundings of a cleanup operation.

*changes*

We address this with the changes made in lines 582 - 595.

*comment*

**The manuscript would benefit from more discussion around the potential applications of the TRAP-tracking approach, mentioned in the very last line of the manuscript (lines 470-471). The current main application mentioned is marine pollution cleanup, but a broader description of the applications (by expanding the very last line of the conclusion) may benefit a broader audience.**

*reply*

We agree and will provide a broader description of potential applications in the final paragraph of the revised manuscript.

*changes*

We address this with the changes made in lines 597 - 605.

*comment*

**A couple of spelling issues:**

1. **Line 153, 'programme' should be 'program'.**

2. **Line 164, 'Mesoscalle' should be 'Mesoscale'.**

3. **Line 265, 'view' should be 'few'.**

4. **Line 304, 'frequenlty' should be 'frequently'.**

5. **Line 436, 'approx.' can just be 'approximately'.**

*reply*

We appreciate the advice and will correct the spelling issues 2 - 5 in the manuscript. The first issue arises from our writing in British English.

*changes*

We address this with the changes made in lines 205 - 206, 325, 366, 337, 380, 392 - 393 and 525.

**3   Other revisions to be mentioned for disclosure**

*changes*

We replace the application example "identification of foraging hotspots" with "oil spill containment" in lines 25 - 26 and 35 since it might benefit a larger community.

*changes*

We apply the same corrections as for the short summary in lines 29 - 36 for the version on the article page:

> TRansient Attracting Profiles (TRAPs) indicate the most attracting regions of the flow and have the potential to facilitate offshore cleanups in the Great Pacific Garbage Patch. We study the characteristics of TRAPs and the prospects for predicting debris transport from a mesoscale permitting dataset. Our findings show the relevance of TRAP lifetime estimations to an operational application and our TRAPs tracking algorithm may benefit even more challenges that are related to the search at sea.

*changes*

To strengthen the point we make, we add one more reference in line 62.

*changes*

We abbreviate the description of the computation of TRAP propagation speeds by simply using the common term "centred difference" in lines 196 - 199.

*changes*

We considered the comparison to propagation speeds computed by "taking the full distance travelled by a TRAP and dividing it by the respective lifetime" in lines 199 - 203 somewhat misplaced after reading it another time. Instead, we decided to briefly emphasise that velocities cannot be computed at the formation and decay of a TRAP and how this can be estimated alternatively.

*changes*

We remove the sentence in lines 252 - 253 since it repeats the start of the section.

*changes*

In the caption of Fig. 8 in line 423, we clarify from which reference group we derive the distributions in panel (b).

*changes*

To keep it consistent with other numbers using a comma instead of a dot we correct the numbers in line 425 and column 7 of Table 1.

*changes*

We correct an unfortunate word choice in lines 452 - 453. "Chaotic" here is probably not a good choice because in the literature of trajectories and hyperbolic motion it has a specific meaning, a meaning that happens to be opposite to what we wish to convey.

*changes*

In lines 616 - 617, we apply the same terms for the individual contributions of each author to prevent a biased impression by different wording. We added a last phrase for the contributions to the revision of the manuscript in line 618.

*changes*

In lines 622 - 623, we thank the referees for their careful review and comments, and we state the usage of the AI tools that LK has used for assistance in English writing, especially for improving the spelling and grammar of the revised manuscript.

*changes*

We note that in the revised manuscript, all references now comply with the formal requirements given by OS.

**References**

AVISO+, CNES, SSALTO/DUACS, and IMEDEA: The altimetric Mesoscale Eddy Trajectories Atlas 3.2 Delayed-Time (META3.2 DT) Allsat Version, AVISO+ [dataset], https://doi.org/10.24400/527896/a01-2022.005.220209, 2022.

Duran, R., Nordam, T., Serra, M., and H. Barker, C.: Chapter 3 - Horizontal transport in oil-spill modeling, in: Marine Hydrocarbon Spill Assessments, edited by Makarynskyy, O., pp. 59–96, Elsevier, https://doi.org/10.1016/B978-0-12-819354-9.00004-1, https://doi.org/10.48550/arXiv.2009.12954, 2021.

Heitzig, J., Kittel, T., Donges, J. F., and Molkenthin, N.: Topology of sustainable management of dynamical systems with desirable states: from defining planetary boundaries to safe operating spaces in the Earth system, Earth System Dynamics, 7, 21–50, https://doi.org/10.5194/esd-7-21-2016, 2016.

Kunz, L.: Track and analyse Transient Attracting Profiles in the Great Pacific Garbage Patch, Zenodo [code], https://doi.org/10.5281/zenodo.12761097, 2024.

Menck, P., Heitzig, J., , and Kurths, J.: How basin stability complements the linear-stability paradigm, Nature Physics, 9, 89–92, https://doi.org/10.1038/nphys2516, 2013.

Serra, M. and Haller, G.: Objective Eulerian coherent structures, Chaos: An Interdisciplinary Journal of Nonlinear Science, 26, 053 110, https://doi.org/10.1063/1.4951720, 2016.

Serra, M., Sathe, P., Rypina, I., Kirincich, A., Ross, S. D., Lermusiaux, P., Allen, A., Peacock, T., and Haller, G.: Search and rescue at sea aided by hidden flow structures, Nature Communications, 11, 2525, https://doi.org/10.1038/s41467-020-16281-x, 2020.

Strogatz, S.: Nonlinear Dynamics and Chaos: With Applications to Physics, Biology, Chemistry, and Engineering, Studies in Nonlinearity, Avalon Publishing, 2014.

---

## Referee Report (RR1)

**Review of resubmitted manuscript:** *Transient Attracting Profiles in the Great Pacific Garbage Patch*, by Kunz et al.

The authors have done a good job at responding to earlier comments, however, I feel the manuscript requires more polish regarding the sentence structure and grammar. I am happy to recommend the manuscript for publication provided additional editing for grammar and sentence structure is completed, and a few comments on word choice are addressed. I provide a *non-exhaustive* list of my concerns below. I feel an exhaustive list of my concerns is unhelpful, rather, I suggest the authors go through the manuscript thoroughly to polish the presentation quality.

**Minor editorial comments:**
1. Line 6 – "we here take …" to "here, we take …" (and on line 59).
2. Line 21 – "… and can benefit even more offshore operations, …", this is a little ambiguous. Do you mean to say that offshore operations are benefitted by a better understanding of TRAPs? Or, that a more (in the numerical sense) offshore operations are benefitted?
3. Line 37 – "that exhibit" to "which exhibit" since it's a non-restrictive sentence.
4. Line 39 – "at [a] global scale".
5. Line 40 – "has been" to "have been" since "experiments" is plural.
6. Line 48 – "…, which eventually allow to derive …" to "from which … can be derived".
7. Line 61 – " … provides answers to this since it …" is unclear. Is 'it' refereeing to the "concept of [TRAPs]" or simply the "[TRAPs]"?
8. Line 64 – "the ocean surface" to "a two-dimensional surface, such as the ocean surface," since, as I understand, TRAPs are not confined to just the ocean surface, but could be computed at depth or along a density surface.
9. Line 71 – "more" can be removed as no direct comparison on robustness is being made.
10. Line 79 (and elsewhere) – "geostrophic + Ekman current velocities", I would refrain from using "+" in a sentence like this, rather, "the combined near-surface geostrophic and Ekman current velocities".
11. Line 85 – I'm not sure I understand what you mean by "altimetry acts like a filter". This can be clearer.
12. Line 87 – "We investigate how these coherent structures relate", unless you define a coherent structure, I would refrain from calling TRAPs "coherent", not to conflate with the typical Lagrangian coherent structures. Additionally, "relate" to what? Perhaps "We investigate the relation between TRAPs and mesoscale eddies in order to…"?

---

## Author Response (AR3)

**Authors' second response for* Transient Attracting Profiles in the Great Pacific Garbage Patch**

Luca Kunz1, Alexa Griesel1, Carsten Eden1, Rodrigo Duran2,3, and Bruno Sainte-Rose4

1Institute of Oceanography, Universität Hamburg, Hamburg, Germany
 2National Energy Technology Laboratory, U.S. Department of Energy, Albany OR, USA
 3Planetary Science Institute, 1700 East Fort Lowell, Tucson, AZ 85719, USA
 4The Ocean Cleanup, Rotterdam, The Netherlands

Correspondence: Luca Kunz (luca.kunz@orac.earth)

**Preface**

We thank the referees for their careful review and helpful comments on our revised manuscript. In the following, we provide a detailed point-by-point response to all referee comments from the second review iteration and specify all changes made to the revised manuscript. Our response to a referee comment is structured in three steps: (1) **comment** from the referee, (2) author's

5 *reply* and (3) author's *changes* in the manuscript. In addition, we provide a marked-up manuscript showing the changes made to the last version of the manuscript using the latexdiff command. Unless otherwise stated, this marked-up manuscript is the reference for line numbers in the following replies.

**1** Replies to referee 1 (report 2)**

**Statement by the referee**

10 I find the revised paper improved but still have some follow-up questions on all my original comments.

**1.1 Mesoscale vs. submesoscale**

I am still confused about the authors' overall take on this question. The authors seem to admit that 1) the statistics of TRAPS in a submesoscale-resolving flow (let's term these submesoscale TRAPS) is likely to be different than that of mesoscale TRAPS, and that 2) submesoscale TRAPS will likely influence the movement of floating objects over short

- 15 timescales (say, a day or less). However, they also claim that 3) it is the mesoscale TRAPS that are most important for the purposes of cleanup operations (on spatial scales of 1-10 km, i.e., in the submesoscale range). I am having a hard time reconciling (3) with (1) and (2), as well as finding support for (3) without actually computing TRAPS in a submesoscale-resolving flow and looking at their statistics and behavior of drifters in their vicinity. I am not implying that the authors necessarily compute submesoscale TRAPS (although they could do it, for example, in a numerical
- 20 model) and their statistics, but at least they should further clarify their overall view on the importance and interaction

reply

We appreciate the comment and acknowledge that this point needs further clarification. We do not claim that (3) mesoscale TRAPs are more important for cleanup operations than submesoscale TRAPs. In lines 513 - 542 of the previous manuscript

- version, we dedicate two paragraphs to the topic of mesoscale vs. submesoscale TRAPs. Contrary to point (3), we emphasise in lines 519 - 520 of the previous manuscript version that "TRAPs computed from submesoscale velocities [...] are crucial to an application of the concept during offshore cleanups.". In lines 530 - 531 of the previous manuscript version, we further explain that surveys about TRAPs computed from SWOT measurements "can further improve the applicability of TRAPs to offshore
- 30 cleanups". This statement implies that we do not expect mesoscale TRAPs to be the best predictor for cleanup operations. However, unless comparable research about submesoscale TRAPs becomes available, we can only recommend mesoscale TRAPs derived from satellite altimetry as an additional tool for the cleanup community. We describe a potential application of mesoscale TRAPs where we explicitly state in lines 538 - 541 of the previous manuscript version that "The large-scale navigation along mesoscale TRAPs could then be complemented by [...] the detection of [...] attracting flow features on the
- 35 small scales [...] ". In lines 541 542 of the previous manuscript version, we communicate the need for more research about the relation between mesoscale and submesoscale TRAPs. These examples disprove point (3). However, we acknowledge that we should explain our view more explicitly throughout the introduction and the conclusion of the new manuscript version. We think both mesoscale and submesoscale TRAPs are important for detecting debris hotspots. We note that computing submesoscale TRAPs from numerical models would go beyond the scope of our study and be better placed in a follow-up project.

40

We will also emphasise our point in lines 521 - 525 of the previous manuscript version that large-scale strain generates/intensifies submesoscale fronts and, therefore, mesoscale TRAPs could be useful as a proxy for intensified submesoscale features while at the same time supplying these submesoscales with material.

45 In the new manuscript version, we will highlight one additional motivation for computing TRAPs from mesoscale-permitting satellite observations, which has not been communicated clearly within previous versions: TRAPs can predict material aggregation. At the mesoscale and daily frequency, TRAPs computed from near-real-time observations of the flow, such as altimetry measurements, should be able to indicate where drifting objects will aggregate within a few days.

**50 changes**

We address these points with the changes made in the following lines:

- 1. mesoscale and submesoscale TRAPs: 84 85, 91 96, 615 619, 621 622, 650 652
- 2. large-scale strain and submesoscale features: 629, 631 633
- 3. prediction of aggregation: 75 80, 91 96, 615 619, 642 644, 663 row 1 of Table B1

**55 1.2 Parameter choices**

**1.2.1 1 degree**

The authors admit that the 1-degree criterion is not necessary and can be dropped in future studies. If so, it might be better to just drop it in this study and do the analysis without it. It is not a requirement for publication, just a suggest tion. Instead of the 1 deg criterion, they suggest using a 30% strength criterion for defining the lengths of TRAP lines.

60 I understand that this criterion has been used in prior work, but the choice of 30% still seems arbitrary to me – why not 25 or 35%? I am suggesting that the authors more clearly acknowledge that TRAP algorithm has several subjective parameter choices

**reply**

65 We appreciate the comment, and we refine our statement about TRAP lengths. It will explain why we keep the 1° limit on TRAP length. As a consequence of this statement, we refrain from recomputing our dataset for different TRAP lengths unless required for publication since it will barely generate new insights for our study purposes:

We truncate TRAP curves wherever the attraction rate along the curve falls below 30 % of the attraction at the respective

- 70 core. Such a cutoff criterion makes sense physically because the attraction of nearby parcels becomes negligible as distance increases away from the core. Without cutoff, TRAPs can become indefinitely long and merge with nearby structures, which makes them hard to distinguish. In addition, their converging ends put wrong emphasis on regions between TRAP cores where the attraction rate is comparably low. Moving away from a TRAP core, the local eigenvectors *e*2 also start pointing in arbitrary directions and are no longer representative of the TRAP. The attraction strength criterion does not necessarily prevent such an excessive integration of the eigenvectors. To obtain an accurate TRAP that indicates hyperbolic flow, one has to define an upper
- limit for TRAP length in addition to the cutoff by attraction strength, see Fig. S1 in the Supplementary Material for details.

The TRAPs algorithm (Serra, 2020) provides default values of 1° for the maximum arclength of a TRAP and 30 % for the attraction strength cutoff. Together, both parameters determine the length of a TRAP curve and must be set thoughtfully before

- computation. With the mesoscale velocity data we use, the preset values provide a clear saddle-type representation of TRAPs. Also, a maximal arclength of 1° limits each TRAP branch to a maximal arclength of 0.5°, which roughly corresponds to the average radius  $\bar{r}_e$  of mesoscale eddies in our domain. We consult an eddy census product by AVISO+ et al. (2022) and derive an average eddy radius and its standard deviation of  $\bar{r}_e \approx (53 \pm 20)$  km. We expect TRAPs to highlight strain between mesoscale eddies, and therefore, it is helpful to study TRAPs and eddies on comparable length scales. For these reasons, we
- 85 keep the preset parameter values. However, this choice does not affect our main diagnostics, and future studies should adjust these settings according to the applied velocity data. Modified TRAP lengths do not change our analysis since our statistics refer to the position and attraction of the TRAP *core*.

changes

90 We implement such a statement with the changes made in lines 170 - 189 and 191 - 193.

**1.2.2 75 km radius**

I appreciate the additional statistics on the distance of drifters to the nearest TRAP and the explanation that this is 1.5 times the eddy radius, but it doesn't fully address my question about why one should care more about the statistics of drifters within 75 km from TRAPS and not, say, within 50 or 25 km?

**reply**

95

We refine our statement on the 75 km search radius. When defining the search zone, our main question was 'Up to what distance can we observe hyperbolic drifter motion around a TRAP?'. We wanted to maximise the search zone in order to capture

- 100 all surrounding hyperbolic drifter motion, regardless of whether it occurs within a distance of, e.g., 25 km or 70 km to a TRAP. We aimed to record as many hyperbolic drifter trajectories as possible since they occur within an abundance of motion patterns, and we needed a large dataset to develop robust statistics.
- Since a quadrupole represents the average flow around mesoscale TRAPs, we assume that the position and size of surrounding
  mesoscale eddies determine the limit to which we can observe hyperbolic drifter motion. AVISO+ et al. (2022) find an average radius of re ≈ 53 km for mesoscale eddies in our domain, and we use it in Fig. 1 to sketch an idealised quadrupole situation. We show that within a search radius of rs = √2re, we can capture the eddy regions that constitute the hyperbolic flow around a TRAP. Smaller radii rs will also allow the detection of hyperbolic drifter motion. However, they will not provide a complete picture up to the centre of an eddy, and they may not suffice for larger eddies, for eddies that are less adjacent than illustrated
  here, or for TRAPs that are up to 25 km off their estimated position due to the coarse resolution of our velocity data. For these reasons, we apply a search radius of rs = √2re = √2 · 53 km ≈ 75 km. We acknowledge that Fig. 1 only illustrates an idealised situation and that rs is constant for all TRAPs. Nevertheless, the agreement between the observed hyperbolic drifter
- 115 The statistics for drifter-TRAP distances additionally support our choice for  $r_s$  since the vast majority of drifter positions is within a radius of 75 km. Drifter detections beyond this limit do not provide more insights than the ones shown in Fig. 10 of the manuscript. Instead, we observe a significant increase in the number of one-day pairings for the 14 % of drifter days beyond this radius. For these reasons, we set  $r_s = 75$  km for our analysis. We will present such a refined statement in the new version of the manuscript.

120

**changes**

We address this with the changes made in lines 273 - 277 and 283 - 302.

motion around mature TRAPs and this scaling of a quadrupole confirms our approach.

We complement this by inserting Fig.1 as the new Fig. S4 in the Supplementary Material. The figure number of all subsequent figures in the Supplementary Material will shift by 1, and we correct the respective references in the new version of the

**125 manuscript.**

Figure 1. Drifter motion around TRAPs and eddies. We embed Fig. 10d of the article in a scheme of four idealised mesoscale eddies. Blue and red circles represent cyclonic and anticyclonic mesoscale eddies, respectively, with a radius equal to the mean radius  $\bar{r}_e \approx 53$  km that we find for mesoscale eddies in our domain, using eddy detections from AVISO+ et al. (2022).  $\bar{r}_e$  determines the search radius  $r_s = \sqrt{2}\bar{r}_e$ of our drifter-TRAP pair algorithm. A transparent TRAP in the middle represents a generic profile, the black circle draws the limits of the drifter search zone around it. The rotation of the idealised eddies aligns with the hyperbolic drifter motion we observe around mature TRAPs.

**1.3 Implications for cleanup**

If a drifter is equally likely to be found near a TRAP as within an eddy, and assuming that statistics of drifters is similar to that for floating debris, then I don't see why one should preferentially look for debris near TRAPS (rather than in the eddies) and concentrate the cleanup efforts there.

**reply**

We acknowledge that our description in lines 478 - 486 of the previous manuscript version raises this question. We clarify our

view in the following statement, which we will communicate more clearly within the new manuscript. We will only mention Fig. S6, former Fig. S5, to illustrate the low number of drifters and as a motivation for future research: 135

In our study domain, the average surface area within a mesoscale eddy is  $\overline{A}_e \approx 8361 \text{ km}^2$ , as derived from eddy detections by AVISO+ et al. (2022). Even if mesoscale eddies accumulate floating material in their interior, there is no preferential region within that area expediting debris collection. For a similar area, however, TRAPs maximise normal attraction of nearby trajectories, which then move tangentially to a TRAP. Surrounding material will move towards the TRAP, aggregate and then

- 140 move along the TRAP towards its ends. Because the circulation aggregates material from both sides of a TRAP into a smaller subarea, we expect the density of debris along a TRAP, i.e. the number of debris per unit area, to be considerably greater than in its periphery. Eventually, the hyperbolic flow would convey this aggregated debris into a strategically placed cleanup system. For these reasons, navigating a cleanup system along mesoscale TRAPs could be more productive than navigating it through 145
- mesoscale eddies.

We demonstrate this aggregation with Fig. 10d of the manuscript, where the hyperbolic flow transports drifters into a smaller subarea. As illustrated in these rotated scenes, there are two pathways for searching debris, i.e. the western and eastern branches of the TRAP, each supplied with material from the north and south. However, we can only show this effect using a composite of

- many drifter trajectories in Fig. 10d. Individual examples of a TRAP attracting multiple drifters are rare due to the low number 150 of drifters in the domain. Fig. 3b presents one of the few observations with two drifters. Although the number of drifters allows us to show the impact of TRAPs on individual, nearby drifters, it is insufficient to quantify the likelihood of drifters aggregating around mesoscale TRAPs and other regions of the flow, such as mesoscale eddies. We illustrate this deficiency with a time series for the number of daily drifter positions spent around TRAPs and within mesoscale eddies in Fig. S6, former Fig. S5, of
- 155 the Supplementary Material. The high standard deviations for respective drifter counts result from the low number of drifters and prevent an accurate comparison. We leave the time series as a motivation for future studies. Prospective research could investigate the likelihood of aggregation using a significantly higher number of drifters or suitable measurements of debris concentrations, which are being collected during current missions.

**changes 160**

We address this with the changes made in lines 552 - 581.

**2 Replies to referee 2 (report 1)**

**Statement by the referee**

165 The authors have done a good job at responding to earlier comments, however, I feel the manuscript requires more polish regarding the sentence structure and grammar. I am happy to recommend the manuscript for publication provided additional editing for grammar and sentence structure is completed, and a few comments on word choice are addressed. I provide a *non-exhaustive* list of my concerns below. I feel an exhaustive list of my concerns is unhelpful, rather, I suggest the authors go through the manuscript thoroughly to polish the presentation quality.

**170 2.1 Minor editorial comments**

1. Line 6 – "we here take ..." to "here, we take ..." (and on line 59).

**reply**

We appreciate the comment and change the wording accordingly.

**175**

**changes**

Respective changes are made in lines 6 and 61.

180

185

2. Line 21 – "... and can benefit even more offshore operations, ...", this is a little ambiguous. Do you mean to say that offshore operations are benefitted by a better understanding of TRAPs? Or, that a more (in the numerical sense) offshore operations are benefitted?

reply

We mean "more" in the numerical sense of "more offshore applications in addition to ocean cleanups". We change the wording accordingly.

**changes**

Respective changes are made in lines 22 - 23.

**190 3. Line 37 – "that exhibit" to "which exhibit" since it's a non-restrictive sentence.**

**reply**

We agree and change the wording accordingly. We will search for similar examples of incorrectly used relative clauses

|    | throughout the manuscript and correct them accordingly.                                                               |
|----|-----------------------------------------------------------------------------------------------------------------------|
|    | changes                                                                                                               |
|    | Respective changes are made in lines 38, 97, 306, 314, 353 and 611.                                                   |
| 4. | Line 39 – "at [a] global scale".                                                                                      |
|    | reply                                                                                                                 |
|    | We agree and change the wording accordingly.                                                                          |
|    | changes                                                                                                               |
|    | Respective changes are made in line 40.                                                                               |
| 5. | Line 40 – "has been" to "have been" since "experiments" is plural.                                                    |
|    | reply                                                                                                                 |
|    | We agree and change the wording accordingly. We will search for similar examples of incorrect conjugations throughout |
|    | the manuscript and correct them accordingly.                                                                          |
|    | changes                                                                                                               |
|    | Respective changes are made in line 41, 57, 403, 429 and 454.                                                         |
| 6. | Line 48 – ", which eventually allow to derive" to "from which can be derived".                                        |
|    | reply                                                                                                                 |
|    | We agree and change the wording accordingly.                                                                          |
|    | changes                                                                                                               |
|    | Respective changes are made in lines 49 - 50.                                                                         |
|    |                                                                                                                       |

**7. Line 61 – "... provides answers to this since it ..." is unclear. Is 'it' refereeing to the "concept of [TRAPs]" or simply the "[TRAPs]"?**

**reply**

225

230

Here, "it" refers to the concept since its numerus is singular and the numerus of TRAPs is plural. We prefer to keep this version of line 61 in the previous manuscript version but we acknowledge that this is a delicate wording. Our idea is that "TRAPs *detect* to most attractive regions of the flow" whereas "the concept *allows to detect* the most attractive regions of the flow". Following this logic, we have to correct a similar expression in line 7 of the abstract. There, we replace "it" by "TRAPs". We prefer this over using "allow to" a second time after a first instance in line 2.

**changes**

**235 Respective changes are made in line 7.**

8. Line 64 – "the ocean surface" to "a two-dimensional surface, such as the ocean surface," since, as I understand, TRAPs are not confined to just the ocean surface, but could be computed at depth or along a density surface.

**240 *reply**

We agree and change the wording accordingly.

**changes**

Respective changes are made in line 66.

**245**

9. Line 71 - "more" can be removed as no direct comparison on robustness is being made.

**reply**

We agree and change the wording accordingly.

**250**

**changes**

Respective changes are made in line 74.

10. Line 79 (and elsewhere) – "geostrophic + Ekman current velocities", I would refrain from using "+" in a sentence
 like this, rather, "the combined near-surface geostrophic and Ekman current velocities".

9

We agree and change the wording accordingly for all instances of "geostrophic + Ekman".

**260 *changes**

Respective changes are made in lines 44 (caption of Fig. 1), 88, 157 - 158, 162 (caption of Fig. 2) and 200.

**11. Line 85 – I'm not sure I understand what you mean by "altimetry acts like a filter". This can be clearer.**

**265**

We mean that conventional altimetry measurements of the ocean surface filter out all small-scale, short-term features of the flow. We clarify the explanation accordingly.

**changes**

reply

270 Respective changes are made in lines 100 - 101.

12. Line 87 – "We investigate how these coherent structures relate", unless you define a coherent structure, I would refrain from calling TRAPs "coherent", not to conflate with the typical Lagrangian coherent structures. Additionally, "relate" to what? Perhaps "We investigate the relation between TRAPs and mesoscale eddies in order to..."?

**275**

**reply**

We acknowledge the comment and apply the suggested formulation. We note that within the manuscript, there are three more instances of the term "coherent structure", two of them being part of the term "Objective Eulerian Coherent Structure" (TRAPs). The third instance of "coherent structure" appears in our discussion about TRAPs and mesoscale eddies where we use it as a synonym for an Agulhas ring. We replace this instance by "coherent eddy" so as not to create the need to define a coherent structure.

changes

Respective changes are made in lines 103 and 547.

285

280

**3 Second revision on English language**

290

300

In response to referee 2 and in addition to our first revision on spelling, grammar, sentence structure and word choice, we thoroughly went through the manuscript and applied minor revisions to further improve these aspects. We also applied minor revisions to meet the quality standards of OS. We call these minor revisions because they aim to enhance the easiness of reading and understanding our paper without changing the meaning of the original content. Some of these revisions imply a rewording of sentences or rearranging of paragraphs. Since we apply a large number of minor changes throughout the entire manuscript,

we only list the most important ones for brevity:

- 1. We rearrange paragraphs by moving lines 189 191 and 193 to lines 166 168.
- 2. We adapt the tense to the narrative in line 251.
- 3. We move "however" from line 305 to line 306 to put the emphasis on our research.
  - 4. We rewrite a cumbersome sentence in lines 334 336.
  - 5. We clarify that the distributions in Fig.4 are spatial distributions with 2000-2019 being the reference period in line 355 (caption of Fig. 4).
  - 6. In line 371, we clarify that we make this conclusion based on the previous paragraph, not only based on the previous sentence.
    - 7. We clarify that water parcels within coherent eddies rotate within and not about closed transport barriers in line 394.
    - 8. We correct a preposition and specify that we mean *coherent* eddies in line 536.
    - 9. We remove a contestable and redundant expression in line 539.
  - 10. We correct the symbol for the mean retention time of hyperbolic drifter motion in line 589.
- 305 11. We remove the term "first-order" in line 610 since it incorrectly expresses that our velocity data would be a first-order choice compared to other mesoscale observations.
  - 12. In order to avoid repetitions, we shorten lines 634 639 and move them to 622 625.

**4 Other revisions to be mentioned for disclosure**

**changes**

310 In the previous versions, multiple references within one referencing command had no specific order. We now follow the OS requirements and, in such cases, order references by date. This causes the reference changes made in lines 44 (caption of Fig. 1), 47 - 48, 318, 353 and 525 - 526.

**changes**

315 We insert "of the study domain" in the caption of Fig. 1 in line 44 to clarify to what the term "boundaries" refers.

**changes**

We acknowledge that our previous descriptions of the vorticity curves  $\zeta(\alpha)$ , the removal of the average background vorticity from each curve, and the usage of the normalised vorticity curves  $\hat{\zeta}(\alpha)$  were somewhat confusing about when and for what purpose these are computed. An average background vorticity is removed from each vorticity curve  $\zeta(\alpha)$  for the detection of

320 purpose these are computed. An average background vorticity is removed from each vorticity curve  $\zeta(\alpha)$  for the detection of vorticity patterns. We use normalised vorticity curves  $\hat{\zeta}(\alpha)$  only for visualisation purposes. We clarify this and correct wrong variables with the changes made in lines 255 - 258, 406 - 409, 411 (caption Fig. 6), 419, 425 (caption Fig. 7) and 456 (caption Fig. 8).

**325 changes**

We insert "observed" in the caption of Fig. 3 in line 259 to clarify that we are demonstrating real observations of drifter movement in the ocean. We complete this by indicating the source for these observations a few lines later within the caption.

**changes**

The pair algorithm works from a drifter's perspective, but we show results that refer to the TRAP core. In lines 273, 274 and 277, we clarify why this is possible.

**changes**

One might actually create statistics from drifter-TRAP pairs that last for one day. However, we cannot derive any statistics from them that are useful for our study purposes. We clarify this by inserting "useful" in line 301.

**changes**

In line 375, we clarify that the article does not provide the mentioned histogram of the mean eddy contour speeds U.

**340 changes**

The changes in lines 476, 481 and 499 arise from changing internal labels within the latex version of the manuscript. However,

these table and figure numbers are already correct within the previous manuscript versions.

**changes**

345 We remove the term "however" in line 542 because the statement by Abernathey and Haller (2018) is actually consistent with the results by Early et al. (2011).

**changes**

In line 677 we now mention the identification number of the funding DFG project. In lines 678 - 679, we now also thank the handling editor for reviewing the new version of the manuscript.

**References**

Abernathey, R. and Haller, G.: Transport by Lagrangian Vortices in the Eastern Pacific, Journal of Physical Oceanography, 48, 667–685, https://doi.org/10.1175/JPO-D-17-0102.1, 2018.

- 355 AVISO+, CNES, SSALTO/DUACS, and IMEDEA: The altimetric Mesoscale Eddy Trajectories Atlas 3.2 Delayed-Time (META3.2 DT) Allsat Version, AVISO+ [dataset], https://doi.org/10.24400/527896/a01-2022.005.220209, 2022.
  - Early, J. J., Samelson, R. M., and Chelton, D. B.: The Evolution and Propagation of Quasigeostrophic Ocean Eddies, Journal of Physical Oceanography, 41, 1535–1555, https://doi.org/10.1175/2011JPO4601.1, 2011.

Serra, M.: Compute Transient Attracting Profiles (TRAPs), GitHub [code], https://github.com/MattiaSerra/TRAPs, 2020.